# 🔎 CATCH: CHANNEL-AWARE MULTIVARIATE TIME SERIES ANOMALY DETECTION VIA FREQUENCY PATCHING

**Xingjian Wu**[1], **Xiangfei Qiu**[1], **Zhengyu Li**[1], **Yihang Wang**[1], **Jilin Hu**[1], **Chenjuan Guo**[1],
**Hui Xiong**[2], **Bin Yang**[1]*

[1]East China Normal University
[2]The Hong Kong University of Science and Technology (Guangzhou)
{xjwu,xfqiu,lizhengyu,yhwang}@stu.ecnu.edu.cn,
{jlhu,cjguo,byang}@dase.ecnu.edu.cn, xionghui@ust.hk

## ABSTRACT

Anomaly detection in multivariate time series is challenging as heterogeneous subsequence anomalies may occur. Reconstruction-based methods, which focus on learning normal patterns in the frequency domain to detect diverse abnormal subsequences, achieve promising results, while still falling short on capturing fine-grained frequency characteristics and channel correlations. To contend with the limitations, we introduce CATCH, a framework based on frequency patching. We propose to patchify the frequency domain into frequency bands, which enhances its ability to capture fine-grained frequency characteristics. To perceive appropriate channel correlations, we propose a Channel Fusion Module (CFM), which features a patch-wise mask generator and a masked-attention mechanism. Driven by a bi-level multi-objective optimization algorithm, the CFM is encouraged to iteratively discover appropriate patch-wise channel correlations, and to cluster relevant channels while isolating adverse effects from irrelevant channels. Extensive experiments on 12 real-world datasets and 12 synthetic datasets demonstrate that CATCH achieves state-of-the-art performance. We make our code and datasets available at https://github.com/decisionintelligence/CATCH.

## 1 INTRODUCTION

Modern cyber physical systems are often monitored by multiple sensors, which produce sucessive multivariate time series data (Wang et al., 2024f; Dai et al., 2024; Sun et al., 2025; Tian et al., 2025; Guo et al., 2014). In recent years, there has been significant progress in multivariate time series analysis, with key tasks such as forecasting (Qiu et al., 2025c; Yu et al., 2023; Wu et al., 2024b; Liu et al., 2024a; Huang et al., 2024; Chen et al., 2024), classification (Yao et al., 2024b; Campos et al., 2023), and imputation (Gao et al., 2024; Wang et al., 2024c;b), among others (Huang et al., 2023; Yao et al., 2024a; Liu et al., 2025), gaining attention. Among these, Multivariate Time Series Anomaly Detection (MTSAD), which focuses on identifying abnormal data in multivariate time series, stands out as a critical studied task. It is applied widely including but not limited to financial fraud detection, medical disease identification, and cybersecurity threat detection (Wen et al., 2022; Yang et al., 2023a; Kieu et al., 2018; 2019; Wu et al., 2024c; Shentu et al., 2024; Yang et al., 2021).

Time series anomalies are typically classified into point anomalies and subsequence anomalies. The point anomalies can be further classified as *contextual* or *global* anomalies (Lai et al., 2021). Recent reconstruction-based methods show strong capability of detecting point anomalies, which are characterized by specific values that significantly deviate from the normal range of the probability distribution. However, subsequence anomalies consist of values that fall within the probability distribution, making them much harder to detect (Paparrizos et al., 2022b; Nam et al., 2024). According to the behavior-driven taxonomy (Lai et al., 2021), the subsequence anomalies can be further divided

---

*Corresponding author

into *seasonal*, *shapelet*, and *trend* anomalies–see Figure 1a. A promising approach is to transform the time series into the frequency domain to better derive the subsequence anomalies.

When transformed into the frequency domain, distinct subsequence anomalies also show prominent differences against the normal series in different frequency bands (Figure 1a). In this case, shapelet anomalies mainly affect the third frequency band while seasonal anomalies affect the first two frequency bands. However, the frequency domain features a long-tailed distribution that most information centralizes in the low frequency bands. Coarse-grained reconstruction-based methods may neglect the details in the high frequency bands (Guo et al., 2023; Piao et al., 2024; Park & Kim, 2022; Wang et al., 2022), thus failing to detect correspond anomalies, which calls for *fine-grained modeling in each frequency band* to precisely reconstruct the normal patterns, so that heterogeneous subsequence anomalies can be detected. Moreover, considering the relationships among channels also promotes better reconstruction for normal patterns.

Figure 1b shows a multivariate time series with three channels, and we observe the varying channel associations in different frequency bands, where Channel 1 and Channel 2 are similar in the third band but dissimilar to Channel 3, and all channels are similar in the fourth band but show dissimilarity in the fifth. However, the commonly-used Channel-Independent (CI) and Channel-Dependent (CD) strategies exhibit polarization effects, rendering them inadequate for this task. CI uses the same model across different channels and overlooks potential channel correlations, which offers robustness (Nie et al., 2023; Qiu et al., 2025a) but lacks generalizability and capacity. CD considers all channels simultaneously with larger capacity, but may be susceptible to noise from irrelevant channels, thus lacking robustness (Han et al., 2024; Qiu et al., 2025a). This calls for *flexibly adapting the distinct channel interrelationships in different frequency bands*.

Inspired by the above observations, we propose **CATCH**, a **C**hannel-**A**ware M**T**SAD framework via frequency pat**ch**ing. Technically, we utilize Fourier Transformation to stretch across time and frequency domains to facilitate the detection of both point and subsequence anomalies, of which the latter can be improved by patching in the frequency domain for fine-grained modeling. To flexibly utilize the channel correlations in frequency bands, we propose a Channel Fusion Module (CFM) that incorporates a Channel Correlation Discovering mechanism and utilizes masked attention through a bi-level multi-objective optimization process. Specifically, we utilize a patch-wise mask generator to adaptively discover channel correla-

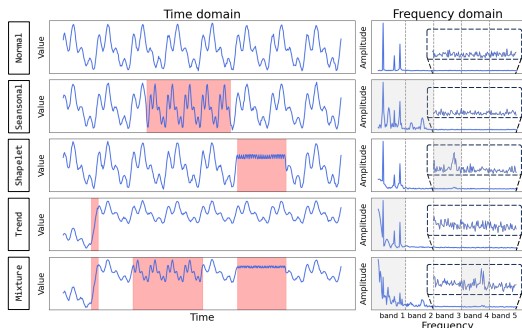

(a) Different subsequence anomalies.

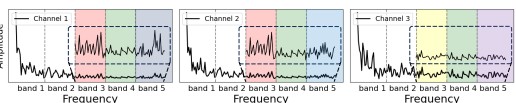

(b) Varying correlations among channels.

Figure 1: The frequency domain is partitioned into five frequency bands with the high-frequency bands zoomed in for clarity. (a) Shows a normal time series and its variations after being injected seasonal, shapelet, trend and mixture subsequence anomalies. In the time domain, anomalies are highlighted in red, while the affected frequency bands in the frequency domain are emphasized in gray. (b) Shows the frequency bands of a multivariate time series with three channels. These channels exhibit varying correlations across different frequency bands, where Channels 1 and 2 exhibit similar behavior in the third frequency band and are therefore marked in red, while Channel 3 exhibits distinct characteristics and is marked in yellow. In the fourth frequency band, all channels behave similarly and are marked in green, while in the fifth band, they exhibit distinct characteristics and are marked in different colors.

tion for each frequency band. The discovered channel correlation is between CI and CD, providing both the capacity and robustness by clustering relevant channels while isolating the adverse effects from irrelevant channels. The contributions are summarized as follows:

- We propose a general framework called CATCH, which enables simultaneous detection of heterogeneous point and subsequence anomalies via frequency patch learning. The framework enhances subsequence anomaly detection through frequency-domain patching and integrates fine-grained adaptive channel correlations across frequency bands.

- We design the CFM to fully utilize the fine-grained channel correlations. Driven by a bi-level multi-objective optimization algorithm, the CFM is able to iteratively discover appropriate channel correlations and facilitate the isolation of irrelevant channels and the clustering of relevant channels, which provides both the capacity and robustness.

- We conduct extensive experiments on 24 multivariate datasets. The results show that CATCH outperforms state-of-the-art baselines.

## 2 RELATED WORK

### 2.1 MULTIVARIATE TIME-SERIES ANOMALY DETECTION

Traditional MTSAD methods can be classified into non-learning (Breunig et al., 2000; Goldstein & Dengel, 2012; Yeh et al., 2016; Li et al., 2024a) and machine learning (Liu et al., 2008; Ramaswamy et al., 2000). Recently, deep learning has made impressive progress in natural language processing (He et al., 2025b; Wang et al., 2024e; Wu et al., 2024a; 2025a), computer vision (Wu et al., 2025b; Li et al., 2025a;b; Yu et al., 2024b), and other aspects (Yi et al., 2025; He et al., 2025a; Li et al., 2023). Studies have shown that learned features may perform better than human-designed features (Yu et al., 2025; 2024a). Therefore, many deep learning-based MTSAD methods have been proposed and have received substantial extensive attention (Liu et al., 2024b; Liu & Paparrizos, 2024; Hu et al., 2024; Wang et al., 2023; Li et al., 2021; Wang et al., 2024a;g). They can be classified into forecasting-based, reconstruction-based and contrastive-based methods. GDN (Deng & Hooi, 2021) is a forecasting-based model that uses a graph structure to learn topology and a graph attention network to encode input series, with anomaly detection based on the maximum forecast error among channel variables. Anomaly Transformer is a reconstructive approach that combines series and prior association to make anomalies distinctive (Xu et al., 2021). DCdetector uses contrastive learning in anomaly detection to create an embedding space where normal data samples are close together and anomalies are farther apart (Yang et al., 2023b).

### 2.2 CHANNEL STRATEGIES IN MTSAD

A channel refers to a variable in MTSAD, while a channel strategy refers to how these channel correlations are effectively considered during the modeling process. CATCH employs a reconstruction-based MTSAD algorithm, using reconstruction error as the anomaly score, making reconstruction quality crucial for anomaly detection accuracy. Since the channels in multivariate time series often exhibit complex dependencies, explicitly modeling these correlations enables a more comprehensive capture of global features, thereby improving reconstruction capabilities and anomaly detection performance. There are mainly two existing approaches that consider relationships among channels. Channel-Independent (CI) based methods such as: PatchTST (Nie et al., 2023) and SparseTSF (Lin et al., 2024) impose the constraint of using the same model across different channels. While it offers robustness, it overlooks potential interactions among channels and can be limited in generalizability and capacity for unseen channels (Han et al., 2024). Previous studies have shown that correlation discovery in data is crucial for time series anomaly detection (Song et al., 2018). Channel-Dependent (CD) based methods such as: MSCRED (Zhang et al., 2019) uses a convolutional-LSTM network with attention and a loss function to reconstruct correlation matrices among channels in multivariate time series input. MTAD-GAT (Zhao et al., 2020) treats each univariate time series as a feature and uses two parallel graph attention layers to capture dependencies across both temporal and channel dimensions. The existing methods could not adequately extract interrelationships, they may be susceptible to noise from irrelevant channels, reducing the model's robustness.

### 2.3 FREQUENCY DOMAIN ANALYSIS FOR TSAD

Frequency domain analysis can uncover subsequence anomalies that are challenging to detect in the time domain, such as anomalies in periodic fluctuations or oscillation patterns, significantly enhancing detection accuracy (Zhang et al., 2022). Conse-

Table 1: Comparison of existing TSAD methods.

| Property | Multivariate time serie anomaly detection | Time-frequency granularity alignment | Handle high-frequency information | Capture channel correlations |
|---|---|---|---|---|
| SR-CNN | ✗ | ✗ | ✗ | ✗ |
| PFT | ✗ | ✗ | ✗ | ✗ |
| TFAD | ✓ | ✗ | ✗ | ✗ |
| Dual-TF | ✓ | ✓ | ✗ | ✗ |
| **CATCH** | ✓ | ✓ | ✓ | ✓ |

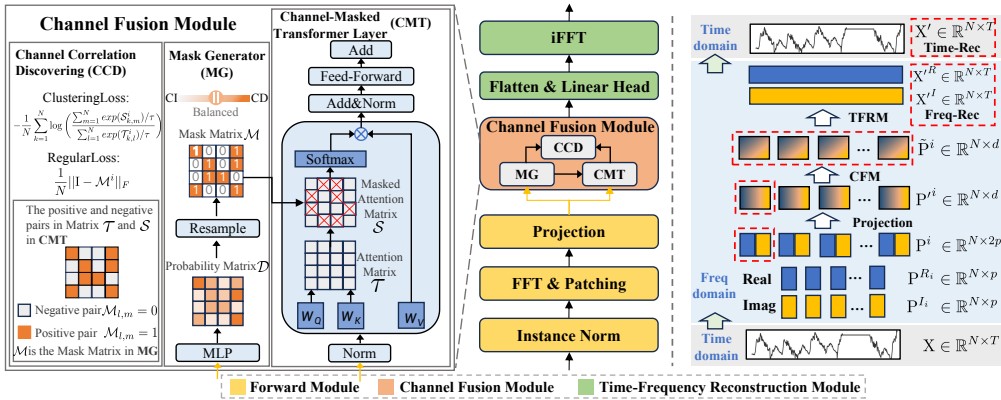

Figure 2: CATCH architecture. (1) Forward Module normalizes the input data, patchifies the frequency domain, and then projects it into the hidden space. (2) Channel Fusion Module captures channel interrelationships in each frequency band with a Channel-Masked Transformer (CMT) Layer, where the mask matrix (channel correlation) is generated by Mask Generator (MG). During training, MG and CMT are optimized by Channel Correlation Discovering (CCD) for more appropriate channel correlations. (3) Time-Frequency Reconstruction Module obtains the frequency reconstruction through Flatten & Linear Head Layer, and obtains the time reconstruction after iFFT.

quently, frequency-based time series anomaly detection models have garnered widespread attention in recent years. SR-CNN (Ren et al., 2019), the first method to leverage the frequency domain for TSAD, employs a frequency-based approach to generate saliency maps for identifying anomalies, and PFT (Zhang et al., 2022) built on this foundation by introducing partial Fourier transform to achieve substantial acceleration. However, both methods are confined to univariate time series and fail to address the complexities of multivariate scenarios. TFAD (Zhang et al., 2022) emerges as the first approach to integrate time-domain and frequency-domain analyses for MTSAD, yet it lacks time-frequency granularity alignment. Dual-TF (Nam et al., 2024), the most recent algorithm for MTSAD using time-frequency analysis, partially addresses the time-frequency granularity alignment issue but lacks a tailored backbone, instead relying directly on the Anomaly Transformer (Xu et al., 2021). While TFAD and Dual-TF represent progress in this area, they still exhibit the following limitations: i) Frequency-domain modeling methods have inherent biases, often overlooking high-frequency information; 2) Insufficient exploration and utilization of channel correlations in multivariate time series.

# 3  CATCH

In the context of time series anomaly detection, $X \in \mathbb{R}^{N \times T}$ denotes a time series with $N$ channels and length $T$. For clear delineation, we separate dimensions with commas and use this format throughout this paper. For example, we denote $X_{i,j}$ as the $i$-th channel at the $j$-th timestamp, $X_{n,:} \in \mathbb{R}^T$ as the time series of $n$-th channel, where $n = 1, 2, \cdots, N$. The multivariate time series anomaly detection problem is to determine whether $X_{:,t}$ is anomaly or not.

## 3.1  STRUCTURE OVERVIEW

Figure 2 shows the overall architecture of the **CATCH**, which consists of three main modules: 1) the *Forward Module*, 2) the *Channel Fusion Module* (CFM), and 3) the *Time-Frequency Reconstruction Module* (TFRM). Specifically, the input multivariate time series is firstly processed via the *Forward Module*, consisting of the *Instance Norm Layer*, *FFT&Patching Layer*, and *Projection Layer*.

The *Instance Norm Layer* is firstly to mitigate the distributional shifts between the training and testing data caused by varying statistical properties, enhancing the model's generalization in reconstructing the testing data. Then, the *FFT&Patching Layer* is to perform fine-grained modeling in each frequency band (patch). Specifically, we utilize the efficient FFT (Brigham & Morrow, 1967) to transform time series into orthogonal trigonometric signals in the frequency domain, where we keep

both the real and imaginary parts through $X^R, X^I = \text{FFT}(X)$ for maximum information retention, where $X, X^R, X^I \in \mathbb{R}^{N \times T}$. Next, we apply the frequency patching operation to create fine-grained frequency bands (patches), which can be formalized as follows.

$$\{P^{R_1}, P^{R_2}, \cdots, P^{R_L}\} = \text{Patching}(X^R), \{P^{I_1}, P^{I_2}, \cdots, P^{I_L}\} = \text{Patching}(X^I), \quad (1)$$

where $P^{R_i}, P^{I_i} \in \mathbb{R}^{N \times p}$ denote the $i$-th patch of $X^R$ and $X^I$. $L = [T - p]/s + 1$ is the total patch number, where $p$ is the patch size and $s$ is the patch stride.

We then concat each pair of $P^{R_i}$ and $P^{I_i}$ into $P^i \in \mathbb{R}^{N \times 2p}$, as the $i$-th frequency patch. After patching in the frequency domain, the frequency patches are then projected into the high-dimensional hidden space through the *Projection Layer*: $P'^i = \text{Projection}(P^i)$.

After the *Forward Module*, the processed time series is fed into the *Channel Fusion Module* (CFM) to dynamically model the channel correlations in each fine-grained frequency band.

$$\{\tilde{P}^1, \tilde{P}^2, \cdots, \tilde{P}^L\} = \text{CFM}(\{P'^1, P'^2, \cdots, P'^L\}), \quad (2)$$

where $P'^i, \tilde{P}^i \in \mathbb{R}^{N \times d}$, $N$ is the number of channels and $d$ is the hidden dimension in attention blocks. The CFM parallels in a patch-wise way to model the frequency patches simultaneously. We further introduce the details of CFM in Section 3.2.

Next, we utilize the *Time-Frequency Reconstruction Module* (TFRM) to reconstruct all frequency spectrums for real and imaginary patches and simultaneously obtain their temporal reconstruction:

$$X', X'^R, X'^I = \text{TFRM}(\{\tilde{P}^{R_1}, \tilde{P}^{R_2}, \cdots, \tilde{P}^{R_L}\}, \{\tilde{P}^{I_1}, \tilde{P}^{I_2}, \cdots, \tilde{P}^{I_L}\}), \quad (3)$$

where $X'$ is the temporal reconstruction, and $X'^R, X'^I \in \mathbb{R}^{N \times T}$ are the frequency reconstruction. Finally, we integrate the reconstruction error in both time and frequency domains as anomaly score.

## 3.2 CHANNEL FUSION MODULE

The *Channel Fusion Module* (CFM) contains the following three components: 1) the *Mask Generator* (MG), 2) the *Channel-Masked Transformer Layer* (CMT), and 3) the *Channel Correlation Discovering* (CCD) mechanism. Specifically, the MG is to perceive and generate the mask matrices (channel correlations) in different frequency bands and guide the masked attention in CMT. The CMT aims to model the appropriate patch-wise channel correlations. And the CCD is to guide the MG and CMT to explore better channel correlations during optimization.

**Mask Generator.** Inspired by Selective State Space Models such as Mamba (Dao & Gu, 2024), which utilizes Linear projections to flexibly update the hidden states based on the current data for larger capacity, the patch-wise channel associations can also be treated as a changing hidden state strongly associated with the current patch. Therefore, we devise a Linear-based mask generator to perceive the suitable channel associations for each frequency band by generating binary mask matrices to isolate the adverse effects from irrelevant channels. Note that the binary mask is an intermediate state between CI (identity matrix) and CD (all-ones matrix) strategies. We take the $i$-th frequency patch as an example:

$$\mathcal{D}^i = \sigma(\text{Linear}(P'^i)), \mathcal{M}^i = \text{Resample}(\mathcal{D}^i), \quad (4)$$

where $P'^i \in \mathbb{R}^{N \times d}$, $\mathcal{D}^i \in \mathbb{R}^{N \times N}$, and $\mathcal{M}^i \in \mathbb{R}^{N \times N}$ are the hidden representation, the probability matrix, and the binary mask matrix of $i$-th patch, respectively. $\sigma$ projects the values to probabilities.

Since our goal is to filter out the adverse effects of irrelevant channels, we further perform Bernoulli resampling on the probability matrices to obtain binary mask matrix $\mathcal{M}^i$ with the same shape. Higher probability $\mathcal{D}^i_{l,m}$ results in $\mathcal{M}^i_{l,m}$ closer to 1, indicating a relationship between channel $l$ and channel $m$. And we manually keep the diagonal items to 1. To ensure the propagation of gradients, we use the Gumbel Softmax reparameterization trick (Jang et al., 2017) during Bernoulli resampling.

**Channel-Masked Transformer Layer.** After the *Mask Generator* outputs the mask matrices for frequency bands, we utilize the transformer layer to further capture the fine-grained channel correlations. The Layer Normalization is applied before each attention block to mitigate the over-focusing

phenomenon on frequency components with larger amplitudes (Piao et al., 2024):

$$P^{*i} = \text{LayerNorm}(P'^{i}) = (P'^{i} - \text{Mean}_{n=1}^{N}(P'^{i}_{n,:}))/\sqrt{\text{Var}_{n=1}^{N}(P'^{i}_{n,:})}, \quad (5)$$

where $P'^{i} \in \mathbb{R}^{N \times d}$ and $P^{*i} \in \mathbb{R}^{N \times d}$ are the hidden representation and the normalized representation of $i$-th patch, respectively. Empirically, we utilize the masked attention mechanism to further model the fine-grained interrelationships among relevant channels and integrate the mask matrix in a calculated way to keep the propagation of gradients:

$$Q^{i} = P^{*i} \cdot W^{Q}, K^{i} = P^{*i} \cdot W^{K}, V^{i} = P^{*i} \cdot W^{V}, \quad (6)$$

$$\mathcal{T}^{i} = Q^{i} \cdot (K^{i})^{T}, \mathcal{S}^{i} = \mathcal{T}^{i} \odot \mathcal{M}^{i} + (1 - \mathcal{M}^{i}) \odot (-\infty), \quad (7)$$

$$\text{MaskedScores}^{i} = \mathcal{S}^{i}/\sqrt{d}, \tilde{P}^{i} = \text{Softmax}(\text{MaskedScores}^{i}) \cdot V^{i}, \quad (8)$$

where $W^{Q}, W^{K}, W^{V} \in \mathbb{R}^{d \times d}$. $\mathcal{M}^{i} \in \mathbb{R}^{N \times N}, \mathcal{T}^{i} \in \mathbb{R}^{N \times N}, \mathcal{S}^{i} \in \mathbb{R}^{N \times N}$, and $\tilde{P}^{i} \in \mathbb{R}^{N \times d}$ are the binary mask matrix, the attention matrix, the masked attention matrix, and the hidden representation processed by CMT of $i$-th patch, respectively.

We utilize the same Feed-Forward networks and skip connections as the classical transformers (Vaswani et al., 2017). We also apply multi-head mechanism to jointly attend to information from different representational subspaces and the CMT can be stacked multiple times.

**Channel Correlation Discovering.** From an optimization perspective, it is essential to design appropriate optimization objectives to enhance the effectiveness of generated masks. A direct motivation is to explicitly enhance the attention scores between relevant channels defined by the mask, thus aligning the attention mechanism with the currently discovered optimal channel correlation, which helps isloate the adverse effects from irrelevant channels and provides robustness for the attention mechanism. Then we iteratively optimize the *Mask Generator* to refine the channel correlations, tuning the capacity of attention mechanism in the *Channel-Masked Transformer Layer* to fully capture the interrelationships between channels. Intuitively, we devise two loss functions to guide the *Mask Generator* exploring the space of channel correlations (from CI to CD). The proposed loss functions are formalized as:

$$\text{ClusteringLoss} = -\frac{1}{N} \sum_{k=1}^{N} \log \left( \frac{\sum_{m=1}^{N} exp(\mathcal{S}_{k,m}^{i})/\tau}{\sum_{l=1}^{N} exp(\mathcal{T}_{k,l}^{i})/\tau} \right), \quad (9) \quad \text{RegularLoss} = \frac{1}{N}||I - \mathcal{M}^{i}||_{F}, \quad (10)$$

where $\tau$ is the temperature coefficient, $N$ is the number of channels, I is the identity matrix, and $|| \cdot ||_{F}$ is the Frobenius norm. $\mathcal{T}^{i} \in \mathbb{R}^{N \times N}, \mathcal{S}^{i} \in \mathbb{R}^{N \times N}$, and $\mathcal{M}^{i} \in \mathbb{R}^{N \times N}$ are the attention matrix, the masked attention matrix, and the binary mask matrix for $i$-th patch, respectively.

The ClusteringLoss is similar to the InfoNCE (He et al., 2020) in form but does not fix the number of "positive" pairs. In contrast, it changes its "positive" pairs for different patches based on the current discovered channel correlation $\mathcal{M}^{i}$. Theoretically, the similarities between "positive" or "negative" pairs are also calculated through inner product, so we share the calculating results of $\mathcal{S}^{i}$ and $\mathcal{T}^{i}$ from the attention mechanism to save the computational cost. As shown in Figure 2, we exhibit the "positive" and "negative" pairs in the attention matrix $\mathcal{T}$ and masked attention matrix $\mathcal{S}$. Specifically, it sets the Query and Key views of relevant channels in a frequency band ($\mathcal{M}_{l,m}^{i} = 1$) as the "positive" pairs, otherwise ($\mathcal{M}_{l,m}^{i} = 0$) "negative" pairs, thus encouraging the $W^{Q}$ and $W^{K}$ to cluster the patch-wise relevant channels in the hidden spaces and lead to higher attention scores. However, only a single ClusteringLoss may cause some adverse effects by urging the *Mask Generator* to output constant ones matrix, so that we add a RegularLoss to mitigate this risk by restricting the number of relevant channels. Equipped with the two optimization objectives, the *Mask Generator* is encouraged to discover appropriate patch-wise channel correlations between CI and CD, and the attention mechanism is also enchanced by optimizing the $W^{Q}$ and $W^{K}$ to learn fine-grained channel representations in the hidden spaces.

### 3.3 TIME FREQUENCY RECONSTRUCTION MODULE

The *Time-Frequency Reconstruction Module* (TFRM) contains the following two components: 1) the *Flatten & Linear Head Layer* and 2) the *iFFT Layer*. Specifically, after the CFM fully extract

the fine-grained channel correlations, we utilize the TFRM to flatten the patch-wise representations and reconstruct all frequency spectrums with MLP projections separately for real and imaginary patches, and obtain temporal reconstruction through iFFT:

$$\mathrm{X}'^{R} = \text{Projection}_{R}(\text{FlattenHead}(\{\tilde{\mathrm{P}}^{R_1}, \tilde{\mathrm{P}}^{R_2}, \cdots, \tilde{\mathrm{P}}^{R_L}\})), \tag{11}$$

$$\mathrm{X}'^{I} = \text{Projection}_{I}(\text{FlattenHead}(\{\tilde{\mathrm{P}}^{I_1}, \tilde{\mathrm{P}}^{I_2}, \cdots, \tilde{\mathrm{P}}^{I_L}\})), \tag{12}$$

$$\mathrm{X}' = \text{iFFT}(\mathrm{X}'^{R}, \mathrm{X}'^{I}), \tag{13}$$

where $\mathrm{X}'^{R}$, $\mathrm{X}'^{I}$, and $\mathrm{X}' \in \mathbb{R}^{N \times T}$. We then adopt the reconstruction loss functions both in time and frequency domains to separately enhance the ability of point-to-point and subsequence modeling. The reconstruction functions in time and frequency domains are formalized as:

$$\text{RecLoss}^{time} = ||\mathrm{X} - \mathrm{X}'||_F^2 \quad (14) \qquad \text{RecLoss}^{freq} = ||\mathrm{X}^R - \mathrm{X}'^{R}||_1 + ||\mathrm{X}^I - \mathrm{X}'^{I}||_1 \quad (15)$$

We utilize 2-norm in the time domain and 1-norm in the frequency domain due to the distinct numerical characteristics of time and frequency domains (Wang et al., 2024d).

### 3.4 Joint Bi-level optimization

We then design a novel joint bi-level training process to enhance the model's ability to detect both point anomalies and subsequence anomalies. Our TotalLoss $\mathcal{L}$ mainly consists of reconstruction loss functions in the time ($\text{RecLoss}^{time}$) and frequency ($\text{RecLoss}^{freq}$) domains, ClusteringLoss and RegularLoss from the *Channel Correlation Discovering* mechanism. The reconstruction loss functions are used to enhance model's ability in both time and frequency domains to detect point and subsequence anomlies. The ClusteringLoss and RegularLoss are used to guide the discovering for fine-grained channel correlations. Specifically, we weightsum these four optimization objectives:

$$\mathcal{L} = \text{RecLoss}^{time} + \lambda_1 \cdot \text{RecLoss}^{freq} + \lambda_2 \cdot \text{ClusteringLoss} + \lambda_3 \cdot \text{RegularLoss}, \tag{16}$$

where $\lambda_1$, $\lambda_2$, and $\lambda_3$ are empircal coefficients. We then utilize a bi-level optimization Algorithm 1 to iteratively update the *Mask Generator* and other model parameters. Intuitively, the process optimizes model parameters for current channel correlations and then discovers better channel correlations for the optimized model parameters, which facilitates the refinement of channel correlations in a continuous way.

### 3.5 Anomaly scoring

When calculating the anomaly score, the convention is to obey the point-to-point manner by assigning an anomaly score for each timestamp, thus mainly reflecting the point anomlies in the time domain. To better quantify subsequence anomalies, existing methods often consider coarse-grained window-granularity scoring, which adds a frequency anomaly score to each point in the whole input window (Ren et al., 2019; Park et al., 2021; Zhang et al., 2022). However, they fail to know the actual boundaries of subsequence anomalies, thus causing misjudgment or omission. During scoring, we perform the patching operation in the input window with the stride length equal to 1. In Figure 3, the shadow in the Time Series indicates a series of subsequence anomalies. Take the calculation of the

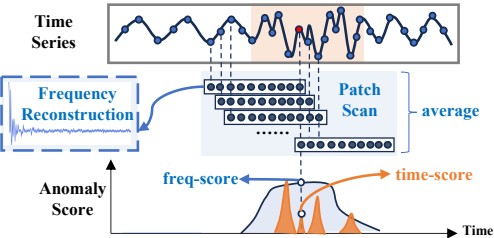

Figure 3: Anomaly Scoring.

anomaly score for the red time point in the Time Series as an example, we first calculate the time domain reconstruction error ($\text{RecLoss}^{time}$) between its reconstructed value in the time domain and the true value using Equation 14, and denote the result as the time domain anomaly score (time-score). At the same time, we collect all patches containing the red time point, transform these patches into the frequency domain, and compute the frequency reconstruction error ($\text{RecLoss}^{freq}$) between the reconstructed value and the true frequency-domain value using Equation 15. The average frequency reconstruction error of all patches is then taken as the frequency domain anomaly score (freq-score)

for this red time point, as the average performs better than the minimum or maximum value (Schmidl et al., 2022). Finally, we weightsum the time-score and freq-score point-to-point to obtain the final anomaly score for the Time Series.

$$\text{AnomalyScore} = \text{time-score} + \lambda_{score} \cdot \text{freq-score} \qquad (17)$$

For ease of understanding, we provide an efficient implementation version of the calculation of freq-score in Appendix A.6. Obviously, our method can reflect the real surroundings of each point by considering all possible subsequence anomalies around this point, thus achieving the point-granularity alignment and showing strong sensitivity (Nam et al., 2024).

## 4 EXPERIMENTS

**Datasets** We conduct experiments using 12 real-world datasets and 12 synthetic datasets (TODS datasets) to assess the performance of CATCH, more details of the benchmark datasets are included in Appendix A.1. The synthetic datasets are generated using the method reported in (Lai et al., 2021). Please refer to Appendix A.7 for specific implementation. We report the results on 12 real-world MTSAD datasets, including MSL, SMAP, PSM, SMD, SWAT, CICIDS, CalIt2, NYC, Creditcard, GECCO, Genesis, and ASD in the main text. We also report the mean results of the 6 types of synthetic anomalies. The complete results can be found in the Appendix C.

**Baselines** We comprehensively compare our model against 15 baselines, including the latest state-of-the-art (SOTA) models. These baselines feature the 2024 SOTA iTransformer (iTrans) (Liu et al., 2024c), ModernTCN (Modern) (Luo & Wang, 2024), and DualTF (Nam et al., 2024), along with the 2023 SOTA Anomaly Transformer (ATrans) (Xu et al., 2021), DCdetector (DC) (Yang et al., 2023b), TimesNet (TsNet) (Wu et al., 2023), PatchTST (Patch) (Nie et al., 2023), DLinear (DLin) (Zeng et al., 2023), NLinear (NLin) (Zeng et al., 2023), TFAD (Zhang et al., 2022), and AutoEncoder (AE) (Sakurada & Yairi, 2014). Additionally, we include non-learning methods such as One-Class SVM (OCSVM)(Schölkopf et al., 1999), Isolation Forest (IF)(Liu et al., 2008), Principal Component Analysis (PCA) (Shyu et al., 2003), and HBOS (Goldstein & Dengel, 2012).

**Setup** To keep consistent with previous works, we adopt Label-based metric: Affiliated-F1-score ($Aff\text{-}F$) (Huet et al., 2022) and Score-based metric: Area under the Receiver Operating Characteristics Curve ($ROC$) (Fawcett, 2006) as evaluation metrics. We report the algorithm performance under a total of 16 evaluation metrics in the Appendix C, and the details of the 16 metrics can be found in Appendix A.2. More implementation details are presented in the Appendix A.3.

### 4.1 MAIN RESULTS

We first evaluate CATCH with 15 competitive baselines on 12 real-world multivariate and 6 types of synthetic multivariate datasets generated by the methods reported in TODS (Lai et al., 2021) as shown in Table 2. It can be seen that our proposed CATCH achieves SOTA results under the widely used Affiliated-F1-score metric in most benchmark datasets. Besides, CATCH has the highest AUC-ROC values on most datasets. It means that our model performs well in the false-positive and true-positive rates under various pre-selected thresholds, which is important for real-world applications. Since the continuous abnormal segments in SWAT exceed the look-back window supported by the reconstruction model, CATCH cannot effectively capture this kind of abnormality. In summary, CATCH effectively handles both point (Contextual, Global) and subsequence (Seasonal, Shapelet, Trend, Mixture) anomalies while showing greater improvement in detecting the subsequence anomalies.

### 4.2 MODEL ANALYSIS

**Ablation study** To ascertain the impact of different modules within CATCH, we perform ablation studies focusing on the following components: (1) Substitute the channel correlation discovering mechanism with fixed Channel Strategies. (2) Delete one of the four optimization objectives separately. (3) Remove the patching operation during training process. (4) Replace the Scoring technique with others. (5) Replace the bi-level optimization process with a normal process to optimize the *Mask Generator* and model simultaneously.

Table 2: Average A-R (AUC-ROC) and Aff-F (Affiliated-F1) accuracy measures for 12 real-world datasets and 6 synthetic datasets of different types of anomalies. The best results are highlighted in bold, and the second-best results are underlined.

| Dataset | Metric | CATCH | Modern | iTrans | DualTF | ATrans | DC | TsNet | Patch | DLin | NLin | AE | Ocsvm | IF | PCA | HBOS | TFAD |
|---|---|---|---|---|---|---|---|---|---|---|---|---|---|---|---|---|---|
| CICIDS | Aff-F | **0.787** | 0.654 | 0.708 | 0.692 | 0.560 | 0.664 | 0.657 | 0.660 | 0.669 | 0.669 | 0.243 | 0.693 | 0.604 | 0.619 | 0.542 | 0.579 |
| | A-R | **0.795** | 0.697 | 0.692 | 0.603 | 0.528 | 0.638 | 0.732 | 0.716 | 0.751 | 0.691 | 0.629 | 0.537 | 0.787 | 0.601 | 0.760 | 0.504 |
| Callt2 | Aff-F | **0.835** | 0.780 | 0.812 | 0.751 | 0.729 | 0.697 | 0.794 | 0.793 | 0.793 | 0.757 | 0.587 | 0.783 | 0.402 | 0.768 | 0.756 | 0.744 |
| | A-R | **0.838** | 0.676 | 0.791 | 0.574 | 0.533 | 0.527 | 0.771 | 0.808 | 0.752 | 0.695 | 0.767 | 0.804 | 0.775 | 0.790 | 0.798 | 0.504 |
| SWAT | Aff-F | 0.755 | 0.728 | 0.718 | 0.695 | 0.696 | 0.696 | **0.793** | 0.716 | 0.736 | 0.715 | 0.738 | 0.703 | 0.681 | 0.708 | 0.650 | 0.686 |
| | A-R | 0.345 | 0.244 | 0.242 | 0.567 | 0.405 | 0.501 | 0.288 | 0.242 | 0.521 | 0.231 | 0.817 | 0.657 | 0.346 | 0.819 | **0.831** | 0.500 |
| Credit | Aff-F | **0.750** | 0.744 | 0.713 | 0.663 | 0.650 | 0.632 | 0.744 | 0.746 | 0.738 | 0.742 | 0.561 | 0.714 | 0.634 | 0.710 | 0.695 | 0.600 |
| | A-R | **0.958** | 0.957 | 0.934 | 0.703 | 0.552 | 0.504 | 0.957 | 0.957 | 0.954 | 0.948 | 0.909 | 0.953 | 0.860 | 0.871 | 0.951 | 0.500 |
| GECCO | Aff-F | **0.908** | 0.893 | 0.839 | 0.701 | 0.782 | 0.687 | 0.894 | 0.906 | 0.893 | 0.882 | 0.823 | 0.666 | 0.424 | 0.785 | 0.708 | 0.627 |
| | A-R | **0.970** | 0.952 | 0.795 | 0.714 | 0.516 | 0.555 | 0.954 | 0.949 | 0.947 | 0.936 | 0.769 | 0.804 | 0.619 | 0.711 | 0.557 | 0.499 |
| Genesis | Aff-F | **0.896** | 0.833 | 0.891 | 0.810 | 0.856 | 0.776 | 0.864 | 0.856 | 0.856 | 0.829 | 0.854 | 0.677 | 0.788 | 0.814 | 0.721 | 0.535 |
| | A-R | **0.974** | 0.676 | 0.690 | 0.937 | 0.947 | 0.659 | 0.913 | 0.685 | 0.696 | 0.755 | 0.931 | 0.733 | 0.549 | 0.815 | 0.897 | 0.497 |
| MSL | Aff-F | **0.740** | 0.726 | 0.710 | 0.588 | 0.692 | 0.694 | 0.734 | 0.724 | 0.725 | 0.723 | 0.625 | 0.641 | 0.584 | 0.678 | 0.680 | 0.665 |
| | A-R | **0.664** | 0.633 | 0.611 | 0.576 | 0.508 | 0.507 | 0.637 | 0.624 | 0.592 | 0.562 | 0.524 | 0.524 | 0.552 | 0.574 | 0.500 | |
| NYC | Aff-F | **0.994** | 0.769 | 0.684 | 0.708 | 0.853 | 0.862 | 0.794 | 0.776 | 0.828 | 0.819 | 0.689 | 0.667 | 0.648 | 0.680 | 0.675 | 0.689 |
| | A-R | **0.816** | 0.466 | 0.640 | 0.633 | 0.671 | 0.549 | 0.791 | 0.709 | 0.768 | 0.671 | 0.504 | 0.456 | 0.446 | 0.502 | | |
| PSM | Aff-F | **0.859** | 0.825 | 0.854 | 0.725 | 0.710 | 0.682 | 0.842 | 0.831 | 0.831 | 0.843 | 0.707 | 0.531 | 0.620 | 0.702 | 0.658 | 0.628 |
| | A-R | **0.652** | 0.593 | 0.592 | 0.600 | 0.514 | 0.501 | 0.592 | 0.586 | 0.580 | 0.585 | 0.650 | 0.619 | 0.542 | 0.648 | 0.620 | 0.500 |
| SMD | Aff-F | **0.847** | 0.840 | 0.827 | 0.679 | 0.724 | 0.675 | 0.831 | 0.845 | 0.841 | 0.844 | 0.439 | 0.742 | 0.626 | 0.738 | 0.629 | 0.660 |
| | A-R | **0.811** | 0.722 | 0.745 | 0.631 | 0.508 | 0.502 | 0.727 | 0.736 | 0.728 | 0.738 | 0.774 | 0.602 | 0.664 | 0.679 | 0.626 | 0.500 |
| SMAP | Aff-F | 0.699 | 0.635 | 0.587 | 0.674 | **0.703** | 0.701 | 0.638 | 0.606 | 0.616 | 0.601 | 0.463 | 0.503 | 0.512 | 0.505 | 0.509 | 0.675 |
| | A-R | 0.504 | 0.455 | 0.409 | 0.478 | 0.511 | 0.522 | 0.453 | 0.448 | 0.397 | 0.434 | 0.522 | 0.393 | 0.487 | 0.396 | **0.585** | 0.500 |
| ASD | Aff-F | **0.804** | 0.782 | 0.780 | 0.605 | 0.674 | 0.702 | 0.800 | 0.777 | 0.782 | 0.766 | 0.731 | 0.617 | 0.781 | 0.656 | 0.669 | 0.630 |
| | A-R | **0.824** | 0.692 | 0.759 | 0.579 | 0.506 | 0.520 | 0.805 | 0.760 | 0.739 | 0.690 | 0.704 | 0.588 | 0.618 | 0.656 | 0.603 | 0.502 |
| Contextual | Aff-F | **0.823** | 0.619 | 0.802 | 0.635 | 0.601 | 0.597 | 0.666 | 0.766 | 0.780 | 0.700 | 0.755 | 0.696 | 0.679 | 0.475 | 0.481 | 0.569 |
| | A-R | **0.910** | 0.562 | 0.905 | 0.598 | 0.546 | 0.525 | 0.908 | 0.854 | 0.700 | 0.530 | 0.896 | 0.711 | 0.821 | 0.538 | 0.464 | 0.504 |
| Global | Aff-F | **0.949** | 0.748 | 0.922 | 0.649 | 0.656 | 0.567 | 0.910 | 0.940 | 0.928 | 0.808 | 0.919 | 0.849 | 0.912 | 0.704 | 0.528 | 0.566 |
| | A-R | **0.997** | 0.873 | 0.976 | 0.595 | 0.564 | 0.514 | 0.989 | 0.992 | 0.979 | 0.675 | 0.996 | 0.996 | 0.938 | 0.758 | 0.608 | 0.500 |
| Seasonal | Aff-F | **0.997** | 0.681 | 0.992 | 0.776 | 0.788 | 0.859 | 0.992 | 0.989 | 0.993 | 0.951 | 0.927 | 0.805 | 0.938 | 0.637 | 0.673 | 0.686 |
| | A-R | **0.998** | 0.512 | 0.946 | 0.701 | 0.584 | 0.644 | 0.958 | 0.922 | 0.823 | 0.623 | 0.949 | 0.829 | 0.918 | 0.437 | 0.516 | 0.502 |
| Shapelet | Aff-F | **0.985** | 0.675 | 0.961 | 0.692 | 0.699 | 0.737 | 0.941 | 0.933 | 0.961 | 0.759 | 0.871 | 0.771 | 0.887 | 0.683 | 0.640 | 0.684 |
| | A-R | **0.970** | 0.522 | 0.864 | 0.573 | 0.519 | 0.597 | 0.877 | 0.818 | 0.684 | 0.563 | 0.865 | 0.655 | 0.748 | 0.517 | 0.337 | 0.503 |
| Trend | Aff-F | **0.916** | 0.734 | 0.901 | 0.677 | 0.584 | 0.765 | 0.897 | 0.888 | 0.721 | 0.830 | 0.699 | 0.691 | 0.914 | 0.693 | 0.669 | 0.642 |
| | A-R | **0.892** | 0.612 | 0.847 | 0.524 | 0.500 | 0.569 | 0.858 | 0.835 | 0.671 | 0.642 | 0.482 | 0.471 | 0.878 | 0.484 | 0.468 | 0.502 |
| Mixture | Aff-F | **0.892** | 0.856 | 0.862 | 0.652 | 0.641 | 0.709 | 0.863 | 0.879 | 0.727 | 0.839 | 0.673 | 0.676 | 0.881 | 0.676 | 0.667 | 0.710 |
| | A-R | **0.931** | 0.763 | 0.854 | 0.570 | 0.522 | 0.516 | 0.861 | 0.863 | 0.767 | 0.749 | 0.493 | 0.475 | 0.911 | 0.517 | 0.531 | 0.501 |

Table 3 illustrates the unique impact of each module. We have the following observations: 1) Compared to the Channel-Independent (CI) Strategy, considering the relationships between variables using Channel Dependent (CD) Strategy or random masking yields better results, with the random masking performing worse than the CD method. Ours outperforms the CD method, further demonstrating the effectiveness of the channel correlation discovering mechanism. 2) Removing any of the four optimization objectives leads to

Table 3: Ablation studies for CATCH in terms of the highest AUC-ROC highlighted in bold.

| Variations | | CICIDS | Callt2 | GECCO | MSL | SMD | Avg |
|---|---|---|---|---|---|---|---|
| Channel Correlation | CI | 0.649 | 0.806 | 0.912 | 0.625 | 0.782 | 0.755 |
| | CD | 0.735 | 0.818 | 0.955 | 0.657 | 0.787 | 0.790 |
| | Random | 0.742 | 0.807 | 0.945 | 0.631 | 0.784 | 0.782 |
| Optimization Objectives | w/o RecLoss$^{time}$ | 0.784 | 0.827 | 0.953 | 0.652 | 0.766 | 0.796 |
| | w/o RecLoss$^{freq}$ | 0.663 | 0.825 | 0.96 | 0.608 | 0.745 | 0.760 |
| | w/o ClusteringLoss | 0.775 | 0.822 | 0.958 | 0.644 | 0.791 | 0.798 |
| | w/o RegularLoss | 0.788 | 0.830 | 0.966 | 0.657 | 0.802 | 0.809 |
| Training w/o Patching | | 0.747 | 0.802 | 0.947 | 0.632 | 0.777 | 0.781 |
| Scoring Technique | point + window score | 0.751 | 0.743 | 0.952 | 0.653 | 0.794 | 0.775 |
| | w/o point score | 0.688 | 0.808 | 0.960 | 0.622 | 0.781 | 0.770 |
| | w/o patch score | 0.763 | 0.780 | 0.956 | 0.648 | 0.785 | 0.787 |
| w/o bi-level optimization | | 0.791 | 0.833 | 0.965 | 0.658 | 0.809 | 0.811 |
| **CATCH (ours)** | | **0.795** | **0.838** | **0.970** | **0.664** | **0.811** | **0.816** |

a decline in model performance, with the most significant drop occurring when the frequency loss is removed. This fully demonstrates the rationality and effectiveness of the four optimization objectives. 3) When the patching operation is removed during training and replaced with a window-based approach to model the relationships between variables, the model performance significantly decreases. This indicates that the patching operation captures fine-grained frequency information, which is more conducive to anomaly detection. 4) When replacing the combination of point-granularity temporal anomaly scores and patch-wise point-aligned frequency anomaly scores with the combination of point-granularity temporal anomaly scores and window-granularity frequency anomaly scores, or when using only one of them, the model performance decreases in both cases. This indicates our Scoring technique shows stronger sensitivity in detecting anomalies. 5) When using a normal optimization process, the model performance also decreases consistently, which provides empircal evdience for the bi-level optimization.

**Parameter Sensitivity** We also study the parameter sensitivity of the CATCH. Figure 4a shows the performance under different input window sizes. As discussed, a single point can not be taken as an instance in time series. Window segmentation is widely used in the analysis, and window size is a significant parameter. For our primary evaluation, the window size is usually set as 96 or 192. Besides, we adopt the score weight in section 3.5 to trade off the temporal score and the

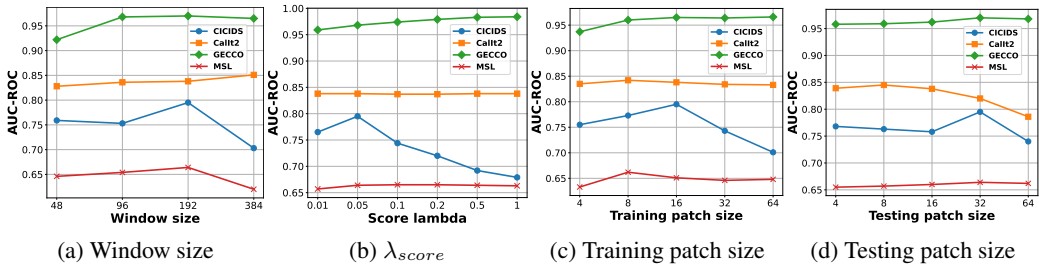

(a) Window size  (b) $\lambda_{score}$  (c) Training patch size  (d) Testing patch size

Figure 4: Parameter sensitivity studies of main hyper-parameters in CATCH.

frequency score–see Figure 4b. We find that score weight is mostly stable and easy to tune in the range of 0.01 to 0.1. Figure 4c and Figure 4d present that our model is stable to the Traing patch size and Testing patch size respectively over extensive datasets. Note that a small patch size indicates a larger memory cost and a larger patch number. Especially, only considering the performance, its relationship to the patch size can be determined by the data pattern. For example, our model performs better when the traing patch size is 8 for MSL, the testing patch size is 32 for CICIDS.

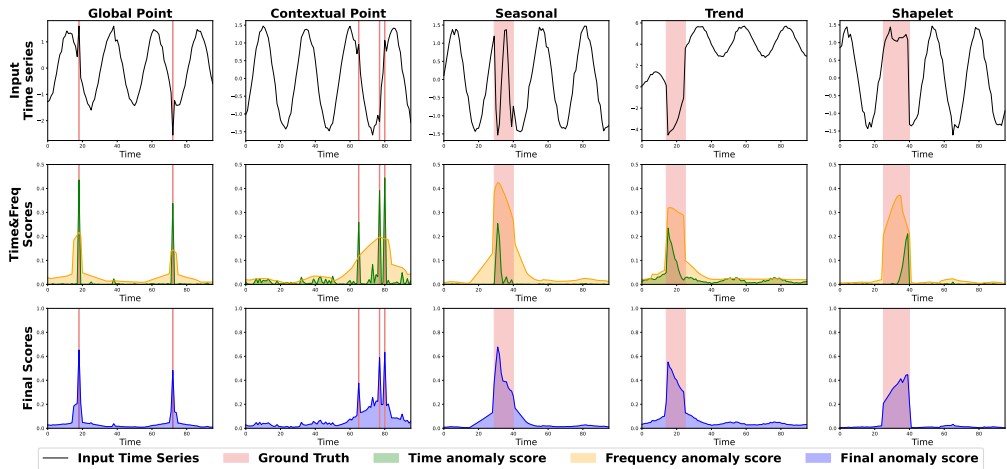

Figure 5: Visualization of dual-domain anomaly scores from CATCH for different categories of point and subsequence anomalies using the TODS dataset.

**Anomaly criterion visualization** We show how CATCH works by visualizing different anomalies in Figure 5. This figure showcases CATCH's performance across five anomaly categories (Lai et al., 2021) on the TODS dataset, with temporal and frequency scores displayed in the second row, and the final anomaly scores in the third row. For point anomalies (first and second columns), the temporal scores exhibit sharp increases at the true anomaly locations, dominating the total scores. In contrast, for subsequence anomalies (third, fourth, and fifth columns), frequency scores remain elevated across the entire anomaly interval, show strong sensitivity to the actual boundaries of subsequence anomalies and compensate for the insensitivity of the temporal scores. Consequently, each domain contributes uniquely, allowing the final anomaly scores to accurately capture both point and subsequence anomalies.

## 5 CONCLUSION

In this paper, we propose a novel framework, CATCH, capable of simultaneously detecting both point and subsequence anomalies. To sum up, it patchifys the frequency domain for fine-grained insights into frequency bands, flexibly perceives and discovers appropriate channel correlations, optimizes the attention mechanism for both robustness and capacity with a bi-level optimization algorithm. These innovative mechanisms collectively empower CATCH to precisely detect both point and subsequence anomalies. Comprehensive experiments on real-world and synthetic datasets demonstrate that CATCH achieves state-of-the-art performance.

ACKNOWLEDGEMENTS

This work was partially supported by National Natural Science Foundation of China (62472174, 62372179). Bin Yang is the corresponding author of the work.

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

## A  EXPERIMENTAL DETAILS

### A.1  DATASETS

Table 4: Statistics of multivariate datasets (AR: anomaly ratio).

| Dataset | Domain | Dim | AR (%) | Avg Total Length | Avg Test Length | Description |
|---|---|---|---|---|---|---|
| MSL | Spacecraft | 55 | 5.88 | 132,046 | 73,729 | Spacecraft incident and anomaly data from the MSL Curiosity rover |
| SMAP | Spacecraft | 25 | 9.72 | 562,800 | 427,617 | Presents the soil samples and telemetry information used by the Mars rover |
| PSM | Server Machine | 25 | 11.07 | 220,322 | 87,841 | A dataset collected from multiple application server nodes at eBay |
| SMD | Server Machine | 38 | 2.08 | 1,416,825 | 708,420 | Telemetry dataset collected from 28 different server machines at a large Internet company |
| Creditcard | Finance | 29 | 0.17 | 284,807 | 142,404 | Credit card dataset includes a subset of online transactions that occurred over two days |
| GECCO | Water treatment | 9 | 1.25 | 138,521 | 69,261 | Water quality dataset published in the GECCO Industrial Challenge |
| SWAT | Water treatment | 51 | 5.78 | 944,919 | 449,919 | Obtained from 51 sensors of the critical infrastructure system under continuous operations |
| CICIDS | Web | 72 | 1.28 | 170,231 | 85,116 | Network traffic data from CICFlowMeter with 80+ features and attack labels |
| CalIt2 | Visitors flowrate | 2 | 4.09 | 5,040 | 2,520 | Person flow dataset records the movement of people in and out of a building over 15 days |
| Genesis | Machinery | 18 | 0.31 | 16,220 | 12,616 | A sensor and control signals dataset collected from cyber-physical production systems |
| NYC | Transport | 3 | 0.57 | 17,520 | 4,416 | Transportation dataset provides information on taxi and ride-hailing trips in New York |
| ASD | Application Server | 19 | 1.55 | 12,848 | 4,320 | Application Server Dataset collected from a large Internet company |
| TODS | Synthetic | 5 | 6.35 | 20,000 | 5,000 | Including 6 anomaly types: global, contextual, shapelet, seasonal, trend, and mix anomalies |

In order to comprehensively evaluate the performance of CATCH, we evaluate 12 real-world datasets and 12 synthetic datasets which cover 9 domains. The anomaly ratio vary from 0.17% to 11.07%, the range of feature dimensions varies from 3 to 72, and the sequence length varies from 5,040 to 1,416,825. This substantial diversity of the datasets enables comprehensive studies of MTSAD methods. Table 4 lists statistics of the 24 multivariate time series.

### A.2  METRICS

The metrics we support can be divided into two categories: Score-based and Label-based. Label-based metrics includes Accuracy ($Acc$), Precision ($P$), Recall ($R$), F1-score ($F1$), Range-Precision ($R$-$P$), Range-Recall ($R$-$R$), Range-F1-score ($R$-$F$) (Tatbul et al., 2018), Precision@k, Affiliated-Precision ($Aff$-$P$), Affiliated-Recall ($Aff$-$R$), and Affiliated-F1-score ($Aff$-$F$) (Huet et al., 2022). Score-based metrics includes the Area Under the Precision-Recall Curve ($A$-$P$) (Davis & Goadrich, 2006), the Area under the Receiver Operating Characteristics Curve ($A$-$R$) (Fawcett, 2006), the Range Area Under the Precision-Recall Curve ($R$-$A$-$P$), the Range Area under the Receiver Operating Characteristics Curve ($R$-$A$-$R$) (Paparrizos et al., 2022a), the Volume Under the Surface of Precision-Recall ($V$-$PR$), and the Volume Under the Surface of Receiver Operating Characteristic ($V$-$ROC$) (Paparrizos et al., 2022a). As noted earlier, CATCH computes all metrics to provide a complete picture of each method. More implementation details are presented in the Appendix A.3.

### A.3  IMPLEMENTATION DETAILS

The "*Drop Last*" issue is reported by several researchers (Qiu et al., 2024; Li et al., 2024b; Qiu et al., 2025b). That is, in some previous works evaluating the model on test set with drop-last=True setting may cause additional errors related to test batch size. In our experiment, to ensure fair comparison in the future, we set the drop last to False for all baselines to avoid this issue.

All experiments are conducted using PyTorch (Paszke et al., 2019) in Python 3.8 and execute on an NVIDIA Tesla-A800 GPU. We employ the ADAM optimizer during training. Initially, the batch size is set to 32, with the option to reduce it by half (to a minimum of 8) in case of an Out-Of-Memory (OOM) situation. We do not use the "*Drop Last*" operation during testing. To ensure reproducibility and facilitate experimentation, datasets and code are available at: https://github.com/decisionintelligence/CATCH.

## A.4 MODEL HYPERPARAMETER SETTINGS

For each baseline method, we strictly follow the hyperparameter configurations recommended in their original papers. Additionally, we conduct hyperparameter searches on multiple sets and select the optimal configurations based on these evaluations to ensure a comprehensive and fair assessment of each method's performance.

The hyperparameters for the baseline methods are set as follows:

- **AutoEncoder:** The architecture of Autoencoder is selected from (10, 3), (25, 10, 5), (50, 20, 10).

- **One-Class SVM:** The RBF kernel is used. The upper bound on the fraction of training errors is set to be 0.5.

- **Isolaion Forest:** The number of estimators is selected from 3, 5, 25, 50, 60, 70, 80, 90, 100, 110.

- **Principal Component Analysis:** The number of principal components is the smaller of the number of samples and the number of features.

- **HBOS:** The number of bins is selected from 3, 5, 10, 20, 30, 40, 50

- **ModernTCN:** The dimension of hidden states is 256. The FFN ratio is 2. Patch size is set as 16 and patch stride is set as 8.

- **Anomaly Transformer, DualTF:** The number of layers is selected form 1, 2, 3, the channel number of hidden states dmodel is 512, and the number of heads is 8. The loss function hyperparameter $lambda$ for balancing two parts is set as 3.

- **iTransformer, TimesNet, DCdetector, PatchTST:** The number of blocks is selected form 1, 2, 3. The dmodel is selected form 8, 64, 128, 256, 512 and the number of heads is 8.

- **DLinear, NLinear:** The length of the reconstructed sequence is set to 100.

- **TFAD:** The number of layers in the TCN is selected from 5, 6, 7, 8, and the number of kernels is selected from 5, 7.

## A.5 BI-LEVEL GRADIENT DESCENT OPTIMIZATION

---
**Algorithm 1** Bi-level Gradient Descent Optimization

---
1: **Input:** Model parameters $\theta_{model}$, $\theta_{mask}$, learning rate $\eta_{model}$, $\eta_{mask}$, number of iterations $\mathcal{N}_O$, $\mathcal{N}_I$, loss function $\mathcal{L} = \text{RecLoss}^{time} + \lambda_1 \cdot \text{RecLoss}^{freq} + \lambda_2 \cdot \text{ClusteringLoss} + \lambda_3 \cdot \text{RegularLoss}$
2: **Initialize:** $\theta_{model} \leftarrow$ initial value, $\theta_{mask} \leftarrow$ initial value
3: **For** $i = 1$ to $\mathcal{N}_O$
4: **Outer Loop:** Update the mask generator parameters
5:     $\theta_{mask} \leftarrow \theta_{mask} - \eta_{mask} \cdot \nabla_{\theta_{mask}} \mathcal{L}(\theta_{model}, \theta_{mask})$     ▷ Update the Mask Generator
6:     **For** $j = 1$ to $\mathcal{N}_I$
7:     **Inner Loop:** Update the model parameters
8:         $\theta_{model} \leftarrow \theta_{model} - \eta_{model} \cdot \nabla_{\theta_{model}} \mathcal{L}(\theta_{model}, \theta_{mask})$     ▷ Update the model
9:     **EndFor**
10: **EndFor**
11: **Output:** Optimized parameters $\theta_{model}$, $\theta_{mask}$

---

## A.6 IMPLEMENTATION DETAILS OF SCORING

We provide an efficient implementation of Frequency-Enhanced Point-Granularity Scoring in Section 3.5. We present the pseudo-code in Algorithm 2. Specifically, we adopt the scatter operation in Pytorch to efficiently parallel the collection of patches to which each point belongs.

---

**Algorithm 2** Calculation of freq-score

---

```python
from einops import rearrange
import torch

class frequency_criterion(torch.nn.Module):
    def __init__(self, configs):
        super(frequency_criterion, self).__init__()
        # Define the frequency metric
        self.metric = frequency_loss(configs, dim=1, keep_dim=True)
        self.patch_size = configs.inference_patch_size
        self.patch_stride = configs.inference_patch_stride
        self.win_size = configs.seq_len
        self.patch_num = int((self.win_size - self.patch_size)
                    / self.patch_stride + 1)
        self.padding_length = self.win_size - (self.patch_size
                    + (self.patch_num - 1) * self.patch_stride)

    def forward(self, outputs, batch_y):

        output_patch = outputs.unfold(dimension=1,
                    size=self.patch_size, step=self.patch_stride)

        b, n, c, p = output_patch.shape
        output_patch = rearrange(output_patch, 'b n c p -> (b n) p c')
        y_patch = batch_y.unfold(dimension=1,
                    size=self.patch_size, step=self.patch_stride)
        y_patch = rearrange(y_patch, 'b n c p -> (b n) p c')

        main_part_loss = self.metric(output_patch, y_patch)

        # Create the patches
        main_part_loss = main_part_loss.repeat(1, self.patch_size, 1)
        main_part_loss = rearrange(main_part_loss,
                        '(b n) p c -> b n p c', b=b)

        # Calculate the overlapped indices
        end_point = self.patch_size + (self.patch_num - 1)
                            * self.patch_stride - 1
        start_indices = np.array(range(0, end_point, self.patch_stride))
        end_indices = start_indices + self.patch_size

        indices = torch.tensor([range(start_indices[i], end_indices[i])
                    for i in range(n)]).unsqueeze(0).unsqueeze(-1)
        indices = indices.repeat(b, 1, 1, c).to(main_part_loss.device)

        # Point-Granularity Alignment
        main_loss = torch.zeros((b, n, self.win_size -
                    self.padding_length, c)).to(main_part_loss.device)
        main_loss.scatter_(dim=2, index=indices, src=main_part_loss)

        non_zero_cnt = torch.count_nonzero(main_loss, dim=1)
        main_loss = main_loss.sum(1) / non_zero_cnt

        # Calculate the metric of the remained part
        if self.padding_length > 0:
            padding_loss = self.metric(outputs[:, -self.padding_length:, :],
                            batch_y[:, -self.padding_length:, :])
            padding_loss = padding_loss.repeat(1, self.padding_length, 1)
            total_loss = torch.concat([main_loss, padding_loss], dim=1)
        else:
            total_loss = main_loss
        return total_loss
```

---

## A.7 COMMAND USED FOR GENERATING THE SYNTHETIC DATASETS

We use the provided source code (Lai et al., 2021) without alterations as demonstrated below, except for adjusting the length parameter to generate a longer time series, to ensure a fair comparison.

---

**Algorithm 3** TODS Synthesis

---

```
"""
This code is based on the original implementation from the TODS project.
Original author: DATA Lab @ Rice University
Source URL: https://github.com/datamllab/tods
"""

# Set base values and behavior types
DIM_NUM = 5
BEHAVIOR = [sine, cosine, sine, cosine, sine]
CONFIG = {"freq": 0.04, "coef": 1.5, "offset": 0.0, "noise_amp": 0.05}
VALUES = [0.145, 0.128, 0.094, 0.077, 0.111, 0.145, 0.179, 0.214, 0.214]

# Generate training data
train_data = MultivariateDataGenerator(dim=DIM_NUM, stream_length=20000,
                  behavior=BEHAVIOR, behavior_config=CONFIG)

# Generate test data
test_data = MultivariateDataGenerator(dim=DIM_NUM, stream_length=5000,
                  behavior=BEHAVIOR, behavior_config=CONFIG)

# Add anomalies based on the specified anomaly type
for i in range(DIM_NUM):
    if anomaly_type == "global_anomaly":
        test_data.point_global_outliers(dim_no=i, ratio=0.01, factor=3.5,
                    radius=5)
    elif anomaly_type == "contextual_anomaly":
        test_data.point_contextual_outliers(dim_no=i, ratio=0.01, factor=2.5,
                    radius=5)
    elif anomaly_type == "shapelet_anomaly":
        test_data.collective_global_outliers(dim_no=i, ratio=0.01, radius=5,
            option='square', coef=1.5, noise_amp=0.03, level=20, freq=0.04,
          base=VALUES, offset=0.0)
    elif anomaly_type == "seasonal_anomaly":
        test_data.collective_seasonal_outliers(dim_no=i, ratio=0.01, factor=3,
            radius=5)
    elif anomaly_type == "trend_anomaly":
        test_data.collective_trend_outliers(dim_no=i, ratio=0.01, factor=0.5,
            radius=5)
    elif anomaly_type == "mixed_subsequence_anomaly":
        test_data.collective_global_outliers(dim_no=i, ratio=0.006, radius=5,
            option="square", coef=1.5, noise_amp=0.03, level=20, freq=0.04,
          base=VALUES, offset=0.0)
        test_data.collective_seasonal_outliers(dim_no=i, ratio=0.006, factor=3,
            radius=5)
        test_data.collective_trend_outliers(dim_no=i, ratio=0.006, factor=0.5,
            radius=5)
```

---

# B ADDITIONAL CASE STUDIES

As shown in Figure 6, we visualize the anomaly scores of various recent SOTAs to obtain an intuitive comparison of detecting accuracy. Our proposed CATCH shows most distinguishable anomaly scores in detecting both point and subsequence anomalies.

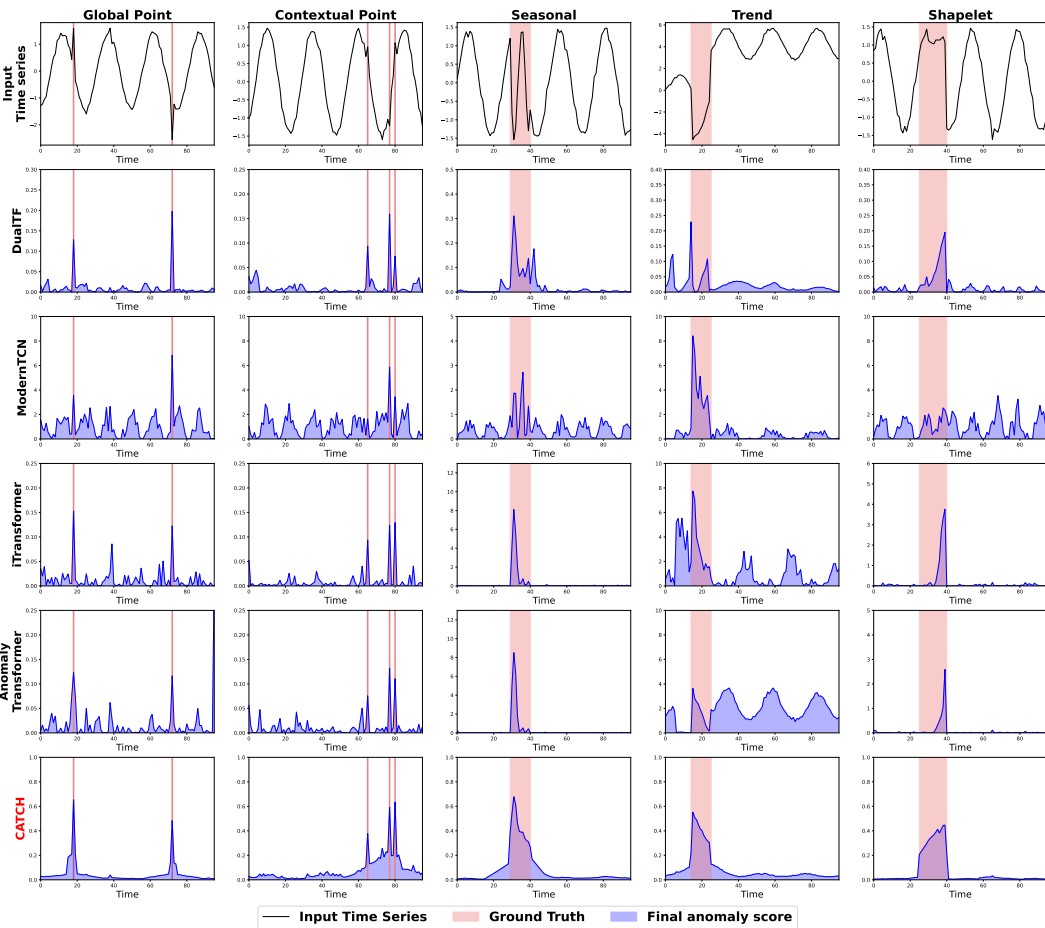

Figure 6: Visualization of anomaly scores from recent SOTAs for the TODS datasets.

## C   FULL EXPERIMENTAL RESULTS

The full MTSAD results are provided in the following section due to the space limitation of the main text. Tables 5, 6, 7, 8, 9, and 10, show the (Accuracy), (AUC-ROC, R-AUC-ROC, VUS-ROC), (AUC-PR, R-AUC-PR, VUS-PR), (Precision, Recall, F1-score), (Range-Recall, Range-Precision, Range-F1-score), (Affiliated-Precision, Affiliated-Recall, Affiliated-F1-score) results, respectively.

Table 5: Average Acc (Accuracy) measures for all datasets. The best results are highlighted in bold, and the second-best results are underlined.

| Dataset | Metric | CATCH | Modern | iTrans | DualTF | ATrans | DC | TsNet | Patch | DLin | NLin | AE | TFAD | Ocsvm | IF | PCA | HBOS |
|---|---|---|---|---|---|---|---|---|---|---|---|---|---|---|---|---|---|
| CICIDS | Acc | 0.758 | 0.750 | 0.762 | 0.565 | 0.745 | 0.750 | 0.750 | 0.751 | 0.757 | 0.754 | **0.994** | 0.952 | 0.082 | 0.852 | 0.656 | 0.913 |
| CalIt2 | Acc | 0.946 | 0.889 | 0.930 | 0.910 | 0.944 | 0.841 | 0.946 | 0.889 | 0.887 | 0.844 | 0.960 | 0.942 | 0.867 | **0.968** | 0.884 | 0.886 |
| Credit | Acc | 0.981 | 0.981 | 0.951 | 0.688 | 0.847 | 0.716 | 0.981 | 0.980 | 0.981 | 0.980 | 0.994 | 0.992 | 0.893 | **0.996** | 0.868 | 0.847 |
| GECCO | Acc | 0.984 | 0.984 | 0.981 | 0.612 | 0.983 | 0.979 | 0.984 | **0.989** | 0.981 | 0.987 | 0.888 | 0.985 | 0.055 | 0.989 | 0.240 | 0.590 |
| Genesis | Acc | **0.992** | 0.965 | 0.986 | 0.970 | 0.931 | 0.991 | 0.991 | 0.987 | 0.987 | 0.972 | 0.987 | 0.991 | 0.424 | 0.983 | 0.888 | 0.771 |
| MSL | Acc | 0.853 | 0.857 | 0.854 | 0.891 | 0.891 | **0.892** | 0.855 | 0.873 | 0.858 | 0.872 | 0.891 | 0.891 | 0.818 | 0.890 | 0.812 | 0.816 |
| NYC | Acc | **0.978** | 0.972 | 0.797 | 0.976 | 0.977 | 0.977 | 0.977 | 0.976 | 0.976 | 0.977 | 0.974 | 0.977 | 0.029 | 0.974 | 0.904 | 0.895 |
| PSM | Acc | 0.730 | 0.729 | 0.728 | 0.697 | 0.719 | 0.615 | 0.726 | 0.729 | 0.726 | 0.728 | 0.715 | 0.720 | **0.737** | 0.723 | 0.682 | 0.736 |
| SMD | Acc | 0.918 | 0.931 | 0.914 | 0.863 | 0.952 | 0.774 | 0.931 | 0.934 | 0.943 | 0.943 | **0.959** | 0.954 | 0.865 | 0.958 | 0.860 | 0.841 |
| SWAT | Acc | 0.857 | 0.836 | 0.794 | 0.635 | 0.867 | 0.865 | 0.873 | 0.837 | 0.822 | 0.794 | 0.709 | **0.878** | 0.196 | 0.877 | 0.375 | 0.748 |
| SMAP | Acc | 0.862 | 0.689 | 0.685 | 0.663 | 0.845 | 0.868 | 0.689 | 0.685 | 0.685 | 0.685 | **0.871** | 0.868 | 0.813 | 0.870 | 0.815 | 0.821 |
| ASD | Acc | 0.933 | 0.913 | 0.898 | 0.851 | 0.942 | 0.930 | 0.928 | 0.942 | 0.947 | 0.912 | 0.946 | 0.950 | 0.871 | **0.953** | 0.863 | 0.870 |
| Contextual4.9 | Acc | 0.961 | 0.734 | 0.957 | 0.713 | 0.641 | 0.618 | 0.734 | 0.888 | 0.943 | 0.742 | **0.965** | 0.874 | 0.888 | 0.956 | 0.848 | 0.865 |
| Contextual7.2 | Acc | **0.960** | 0.721 | 0.937 | 0.603 | 0.702 | 0.611 | 0.739 | 0.886 | 0.893 | 0.729 | 0.956 | 0.859 | 0.877 | 0.939 | 0.818 | 0.850 |
| Global4.8 | Acc | 0.993 | 0.896 | 0.994 | 0.833 | 0.174 | 0.989 | 0.989 | **0.995** | 0.993 | 0.895 | 0.980 | 0.876 | 0.903 | 0.977 | 0.698 | 0.872 |
| Global7.2 | Acc | 0.978 | 0.849 | 0.972 | 0.326 | 0.232 | 0.665 | **0.987** | 0.984 | 0.985 | 0.900 | 0.978 | 0.855 | 0.915 | 0.970 | 0.644 | 0.843 |
| Seasonal4.8 | Acc | **0.988** | 0.861 | 0.973 | 0.956 | 0.956 | 0.952 | 0.974 | 0.974 | 0.974 | 0.964 | 0.959 | 0.947 | 0.890 | 0.957 | 0.855 | 0.863 |
| Seasonal7.7 | Acc | **0.979** | 0.792 | 0.964 | 0.894 | 0.928 | 0.925 | 0.963 | 0.960 | 0.965 | 0.943 | 0.938 | 0.916 | 0.887 | 0.939 | 0.839 | 0.849 |
| Shapelet4.9 | Acc | **0.969** | 0.820 | 0.966 | 0.942 | 0.950 | 0.946 | 0.967 | 0.953 | 0.968 | 0.947 | 0.950 | 0.947 | 0.879 | 0.952 | 0.851 | 0.855 |
| Shapelet7.4 | Acc | **0.957** | 0.715 | 0.949 | 0.871 | 0.921 | 0.823 | 0.951 | 0.943 | 0.947 | 0.891 | 0.932 | 0.913 | 0.872 | 0.925 | 0.838 | 0.839 |
| Mixture5.7 | Acc | 0.903 | 0.893 | 0.885 | 0.658 | 0.928 | 0.940 | 0.896 | 0.889 | 0.400 | 0.882 | 0.692 | 0.921 | 0.183 | **0.948** | 0.183 | 0.190 |
| Mixture8.9 | Acc | 0.808 | 0.849 | 0.822 | 0.638 | 0.882 | 0.910 | 0.822 | 0.814 | 0.587 | 0.804 | 0.650 | 0.859 | 0.123 | **0.923** | 0.124 | 0.215 |
| Trend4.8 | Acc | 0.838 | 0.873 | 0.891 | 0.811 | 0.941 | 0.946 | 0.913 | 0.891 | 0.590 | 0.812 | 0.655 | 0.947 | 0.064 | **0.955** | 0.065 | 0.222 |
| Trend7.8 | Acc | 0.911 | 0.813 | 0.790 | 0.785 | 0.903 | 0.911 | 0.801 | 0.878 | 0.603 | 0.864 | 0.640 | 0.893 | 0.079 | **0.927** | 0.080 | 0.319 |
| $1^{st}$ **Count** | | **7** | 0 | 0 | 0 | 0 | 1 | 1 | 2 | 0 | 0 | 4 | 1 | 1 | 7 | 0 | 0 |

Table 6: Average A-R (AUC-ROC), R-A-R (R-AUC-ROC) and V-ROC (VUS-ROC) accuracy measures for all datasets. The best results are highlighted in bold, and the second-best results are underlined.

| Dataset | Metric | CATCH | Modern | iTrans | DualTF | ATrans | DC | TsNet | Patch | DLin | NLin | AE | TFAD | Ocsvm | IF | PCA | HBOS |
|---|---|---|---|---|---|---|---|---|---|---|---|---|---|---|---|---|---|
| CICIDS | A-R | **0.795** | 0.697 | 0.692 | 0.603 | 0.528 | 0.638 | 0.732 | 0.716 | 0.751 | 0.691 | 0.629 | 0.504 | 0.537 | 0.787 | 0.601 | 0.760 |
| | R-A-R | 0.737 | 0.637 | 0.584 | 0.434 | 0.392 | 0.492 | 0.684 | 0.650 | 0.692 | 0.609 | 0.574 | 0.498 | 0.481 | **0.739** | 0.478 | 0.720 |
| | V-ROC | 0.743 | 0.649 | 0.626 | 0.485 | 0.408 | 0.528 | 0.693 | 0.662 | 0.703 | 0.631 | 0.586 | 0.498 | 0.505 | **0.750** | 0.496 | 0.735 |
| CalIt2 | A-R | **0.838** | 0.676 | 0.791 | 0.574 | 0.533 | 0.527 | 0.771 | 0.808 | 0.752 | 0.695 | 0.767 | 0.504 | 0.804 | 0.775 | 0.790 | 0.798 |
| | R-A-R | **0.854** | 0.726 | 0.817 | 0.666 | 0.523 | 0.505 | 0.807 | 0.830 | 0.794 | 0.743 | 0.798 | 0.502 | 0.840 | 0.818 | 0.796 | 0.847 |
| | V-ROC | **0.848** | 0.716 | 0.809 | 0.630 | 0.517 | 0.503 | 0.796 | 0.824 | 0.781 | 0.728 | 0.789 | 0.501 | 0.828 | 0.802 | 0.786 | 0.831 |
| Credit | A-R | **0.958** | 0.957 | 0.934 | 0.703 | 0.552 | 0.504 | 0.957 | 0.957 | 0.954 | 0.948 | 0.909 | 0.500 | 0.953 | 0.860 | 0.871 | 0.951 |
| | R-A-R | **0.921** | 0.919 | 0.882 | 0.535 | 0.379 | 0.483 | 0.919 | 0.918 | 0.914 | 0.902 | 0.843 | 0.498 | 0.914 | 0.809 | 0.784 | 0.904 |
| | V-ROC | **0.917** | 0.908 | 0.881 | 0.575 | 0.398 | 0.483 | 0.914 | 0.911 | 0.908 | 0.897 | 0.840 | 0.498 | 0.905 | 0.818 | 0.805 | 0.896 |
| GECCO | A-R | **0.970** | 0.952 | 0.795 | 0.714 | 0.516 | 0.555 | 0.954 | 0.949 | 0.947 | 0.936 | 0.769 | 0.499 | 0.804 | 0.619 | 0.711 | 0.557 |
| | R-A-R | **0.990** | 0.978 | 0.884 | 0.725 | 0.501 | 0.609 | 0.977 | 0.984 | 0.987 | 0.977 | 0.634 | 0.501 | 0.757 | 0.680 | 0.598 | 0.510 |
| | V-ROC | **0.987** | 0.975 | 0.871 | 0.717 | 0.503 | 0.593 | 0.974 | 0.979 | 0.982 | 0.971 | 0.637 | 0.500 | 0.756 | 0.677 | 0.595 | 0.503 |
| Genesis | A-R | **0.974** | 0.676 | 0.690 | 0.937 | 0.947 | 0.659 | 0.913 | 0.685 | 0.696 | 0.755 | 0.931 | 0.497 | 0.733 | 0.549 | 0.815 | 0.897 |
| | R-A-R | **0.981** | 0.727 | 0.797 | 0.975 | 0.976 | 0.744 | 0.919 | 0.737 | 0.741 | 0.791 | 0.917 | 0.499 | 0.734 | 0.693 | 0.822 | 0.819 |
| | V-ROC | **0.978** | 0.729 | 0.773 | 0.971 | 0.970 | 0.730 | 0.913 | 0.728 | 0.735 | 0.787 | 0.916 | 0.499 | 0.733 | 0.660 | 0.816 | 0.827 |
| MSL | A-R | **0.664** | 0.633 | 0.611 | 0.576 | 0.508 | 0.507 | 0.613 | 0.637 | 0.624 | 0.592 | 0.562 | 0.500 | 0.524 | 0.524 | 0.552 | 0.574 |
| | R-A-R | **0.747** | 0.708 | 0.686 | 0.662 | 0.529 | 0.596 | 0.701 | 0.720 | 0.703 | 0.681 | 0.635 | 0.515 | 0.594 | 0.575 | 0.631 | 0.643 |
| | V-ROC | **0.735** | 0.701 | 0.678 | 0.652 | 0.527 | 0.587 | 0.692 | 0.712 | 0.695 | 0.672 | 0.628 | 0.514 | 0.590 | 0.571 | 0.622 | 0.635 |
| NYC | A-R | **0.816** | 0.466 | 0.640 | 0.633 | 0.671 | 0.549 | 0.791 | 0.709 | 0.768 | 0.671 | 0.504 | 0.502 | 0.456 | 0.475 | 0.666 | 0.446 |
| | R-A-R | **0.836** | 0.598 | 0.697 | 0.754 | 0.753 | 0.524 | 0.765 | 0.722 | 0.762 | 0.706 | 0.636 | 0.509 | 0.612 | 0.632 | 0.744 | 0.545 |
| | V-ROC | **0.827** | 0.579 | 0.689 | 0.737 | 0.722 | 0.530 | 0.771 | 0.719 | 0.760 | 0.701 | 0.619 | 0.502 | 0.589 | 0.610 | 0.730 | 0.536 |
| PSM | A-R | **0.652** | 0.593 | 0.592 | 0.600 | 0.514 | 0.501 | 0.592 | 0.586 | 0.580 | 0.585 | 0.650 | 0.500 | 0.619 | 0.542 | 0.648 | 0.620 |
| | R-A-R | **0.640** | 0.588 | 0.589 | 0.507 | 0.453 | 0.489 | 0.593 | 0.586 | 0.579 | 0.585 | 0.587 | 0.499 | 0.530 | 0.543 | 0.584 | 0.572 |
| | V-ROC | **0.639** | 0.589 | 0.588 | 0.507 | 0.451 | 0.479 | 0.593 | 0.585 | 0.579 | 0.585 | 0.589 | 0.499 | 0.532 | 0.542 | 0.585 | 0.575 |
| SMD | A-R | **0.811** | 0.722 | 0.745 | 0.631 | 0.508 | 0.502 | 0.727 | 0.736 | 0.728 | 0.738 | 0.774 | 0.500 | 0.602 | 0.664 | 0.679 | 0.626 |
| | R-A-R | **0.800** | 0.743 | 0.762 | 0.594 | 0.500 | 0.505 | 0.747 | 0.760 | 0.754 | 0.762 | 0.783 | 0.500 | 0.579 | 0.679 | 0.656 | 0.597 |
| | V-ROC | **0.797** | 0.742 | 0.761 | 0.592 | 0.499 | 0.505 | 0.746 | 0.758 | 0.751 | 0.760 | 0.782 | 0.500 | 0.578 | 0.678 | 0.655 | 0.597 |
| SWAT | A-R | 0.345 | 0.244 | 0.242 | 0.567 | 0.405 | 0.501 | 0.288 | 0.242 | 0.521 | 0.231 | 0.817 | 0.500 | 0.657 | 0.346 | 0.819 | **0.831** |
| | R-A-R | 0.473 | 0.357 | 0.353 | 0.611 | 0.437 | 0.511 | 0.402 | 0.354 | 0.588 | 0.338 | 0.645 | 0.509 | 0.628 | 0.431 | 0.717 | **0.815** |
| | V-ROC | 0.462 | 0.348 | 0.344 | 0.608 | 0.434 | 0.511 | 0.392 | 0.345 | 0.581 | 0.330 | 0.628 | 0.508 | 0.618 | 0.424 | 0.687 | **0.784** |
| SMAP | A-R | 0.504 | 0.455 | 0.409 | 0.478 | 0.511 | 0.522 | 0.453 | 0.448 | 0.397 | 0.434 | 0.522 | 0.500 | 0.393 | 0.487 | 0.396 | **0.585** |
| | R-A-R | 0.546 | 0.491 | 0.441 | 0.516 | 0.520 | 0.552 | 0.490 | 0.483 | 0.435 | 0.466 | 0.545 | 0.504 | 0.430 | 0.499 | 0.420 | **0.565** |
| | V-ROC | 0.543 | 0.489 | 0.439 | 0.515 | 0.520 | 0.552 | 0.489 | 0.482 | 0.435 | 0.465 | 0.544 | 0.504 | 0.428 | 0.499 | 0.419 | **0.566** |
| ASD | A-R | **0.824** | 0.692 | 0.759 | 0.579 | 0.506 | 0.520 | 0.805 | 0.760 | 0.739 | 0.690 | 0.704 | 0.502 | 0.588 | 0.618 | 0.656 | 0.603 |
| | R-A-R | **0.861** | 0.746 | 0.812 | 0.624 | 0.508 | 0.515 | 0.835 | 0.813 | 0.793 | 0.741 | 0.753 | 0.506 | 0.612 | 0.694 | 0.682 | 0.624 |
| | V-ROC | **0.853** | 0.739 | 0.804 | 0.611 | 0.507 | 0.516 | 0.831 | 0.806 | 0.784 | 0.735 | 0.745 | 0.506 | 0.608 | 0.685 | 0.674 | 0.619 |
| Contextual4.9 | A-R | **0.921** | 0.566 | 0.916 | 0.660 | 0.554 | 0.535 | 0.919 | 0.862 | 0.709 | 0.545 | 0.914 | 0.497 | 0.706 | 0.840 | 0.538 | 0.470 |
| | R-A-R | **0.859** | 0.386 | 0.851 | 0.504 | 0.396 | 0.405 | 0.856 | 0.761 | 0.543 | 0.391 | 0.848 | 0.498 | 0.517 | 0.736 | 0.364 | 0.310 |
| | V-ROC | **0.854** | 0.443 | 0.845 | 0.548 | 0.431 | 0.422 | 0.850 | 0.768 | 0.586 | 0.437 | 0.845 | 0.501 | 0.563 | 0.743 | 0.405 | 0.362 |
| Contextual7.2 | A-R | **0.899** | 0.558 | 0.893 | 0.536 | 0.539 | 0.514 | 0.897 | 0.845 | 0.691 | 0.515 | 0.879 | 0.510 | 0.716 | 0.802 | 0.538 | 0.458 |
| | R-A-R | **0.818** | 0.388 | 0.812 | 0.361 | 0.374 | 0.436 | 0.816 | 0.730 | 0.522 | 0.360 | 0.789 | 0.466 | 0.530 | 0.674 | 0.362 | 0.297 |
| | V-ROC | **0.822** | 0.442 | 0.814 | 0.424 | 0.406 | 0.443 | 0.819 | 0.746 | 0.567 | 0.409 | 0.797 | 0.472 | 0.585 | 0.693 | 0.401 | 0.351 |
| Global4.8 | A-R | **0.998** | 0.885 | 0.981 | 0.631 | 0.560 | 0.515 | 0.991 | 0.993 | 0.981 | 0.685 | 0.996 | 0.501 | 0.998 | 0.947 | 0.770 | 0.609 |
| | R-A-R | **0.995** | 0.804 | 0.963 | 0.464 | 0.388 | 0.383 | 0.982 | 0.986 | 0.962 | 0.554 | 0.991 | 0.463 | 0.995 | 0.916 | 0.654 | 0.454 |
| | V-ROC | **0.979** | 0.806 | 0.952 | 0.533 | 0.430 | 0.402 | 0.967 | 0.967 | 0.942 | 0.579 | 0.972 | 0.468 | 0.973 | 0.903 | 0.687 | 0.489 |
| Global7.2 | A-R | **0.997** | 0.861 | 0.971 | 0.559 | 0.568 | 0.513 | 0.988 | 0.992 | 0.978 | 0.665 | 0.996 | 0.499 | 0.995 | 0.928 | 0.745 | 0.608 |
| | R-A-R | **0.993** | 0.772 | 0.948 | 0.395 | 0.403 | 0.383 | 0.977 | 0.984 | 0.957 | 0.534 | 0.992 | 0.498 | 0.989 | 0.886 | 0.618 | 0.478 |
| | V-ROC | **0.979** | 0.772 | 0.937 | 0.442 | 0.439 | 0.397 | 0.962 | 0.966 | 0.939 | 0.560 | 0.972 | 0.502 | 0.970 | 0.878 | 0.653 | 0.512 |
| Seasonal4.8 | A-R | **0.999** | 0.511 | 0.945 | 0.865 | 0.590 | 0.661 | 0.958 | 0.922 | 0.819 | 0.613 | 0.943 | 0.498 | 0.840 | 0.921 | 0.447 | 0.522 |
| | R-A-R | **1.000** | 0.613 | 0.929 | 0.972 | 0.560 | 0.642 | 0.933 | 0.932 | 0.847 | 0.676 | 0.895 | 0.528 | 0.849 | 0.898 | 0.573 | 0.629 |
| | V-ROC | **0.999** | 0.596 | 0.929 | 0.950 | 0.567 | 0.634 | 0.932 | 0.929 | 0.839 | 0.668 | 0.896 | 0.522 | 0.845 | 0.897 | 0.544 | 0.609 |
| Seasonal7.7 | A-R | **0.997** | 0.513 | 0.946 | 0.537 | 0.579 | 0.628 | 0.959 | 0.921 | 0.827 | 0.634 | 0.954 | 0.506 | 0.817 | 0.915 | 0.426 | 0.509 |
| | R-A-R | **0.997** | 0.640 | 0.946 | 0.638 | 0.558 | 0.623 | 0.943 | 0.931 | 0.860 | 0.691 | 0.907 | 0.532 | 0.844 | 0.914 | 0.583 | 0.615 |
| | V-ROC | **0.994** | 0.615 | 0.945 | 0.616 | 0.559 | 0.611 | 0.946 | 0.929 | 0.851 | 0.680 | 0.905 | 0.527 | 0.838 | 0.913 | 0.546 | 0.602 |
| Shapelet4.9 | A-R | **0.982** | 0.527 | 0.856 | 0.623 | 0.532 | 0.627 | 0.871 | 0.812 | 0.670 | 0.552 | 0.857 | 0.502 | 0.651 | 0.767 | 0.490 | 0.342 |
| | R-A-R | **0.996** | 0.608 | 0.883 | 0.742 | 0.608 | 0.602 | 0.880 | 0.851 | 0.714 | 0.642 | 0.854 | 0.531 | 0.716 | 0.800 | 0.624 | 0.510 |
| | V-ROC | **0.992** | 0.599 | 0.874 | 0.723 | 0.594 | 0.596 | 0.875 | 0.844 | 0.712 | 0.627 | 0.851 | 0.525 | 0.711 | 0.794 | 0.601 | 0.485 |
| Shapelet7.4 | A-R | **0.957** | 0.518 | 0.872 | 0.523 | 0.506 | 0.567 | 0.882 | 0.824 | 0.697 | 0.573 | 0.874 | 0.503 | 0.660 | 0.728 | 0.545 | 0.332 |
| | R-A-R | **0.991** | 0.618 | 0.877 | 0.624 | 0.527 | 0.546 | 0.885 | 0.860 | 0.748 | 0.640 | 0.853 | 0.530 | 0.745 | 0.780 | 0.687 | 0.512 |
| | V-ROC | **0.984** | 0.606 | 0.876 | 0.603 | 0.518 | 0.537 | 0.885 | 0.853 | 0.741 | 0.633 | 0.856 | 0.527 | 0.727 | 0.771 | 0.658 | 0.486 |
| Mixture5.7 | A-R | **0.906** | 0.628 | 0.859 | 0.524 | 0.490 | 0.577 | 0.870 | 0.850 | 0.692 | 0.648 | 0.452 | 0.503 | 0.440 | 0.881 | 0.470 | 0.443 |
| | R-A-R | **0.910** | 0.746 | 0.909 | 0.649 | 0.466 | 0.519 | 0.910 | 0.902 | 0.759 | 0.738 | 0.390 | 0.511 | 0.386 | 0.876 | 0.394 | 0.465 |
| | V-ROC | 0.900 | 0.726 | 0.899 | 0.623 | 0.463 | 0.522 | **0.902** | 0.892 | 0.745 | 0.722 | 0.385 | 0.510 | 0.381 | 0.877 | 0.389 | 0.458 |
| Mixture8.9 | A-R | **0.879** | 0.596 | 0.834 | 0.523 | 0.509 | 0.562 | 0.845 | 0.819 | 0.650 | 0.636 | 0.513 | 0.500 | 0.503 | 0.874 | 0.497 | 0.492 |
| | R-A-R | **0.887** | 0.706 | 0.866 | 0.630 | 0.509 | 0.552 | 0.871 | 0.857 | 0.723 | 0.728 | 0.505 | 0.517 | 0.476 | 0.878 | 0.463 | 0.548 |
| | V-ROC | **0.880** | 0.687 | 0.860 | 0.610 | 0.501 | 0.542 | 0.865 | 0.850 | 0.708 | 0.710 | 0.490 | 0.516 | 0.462 | 0.876 | 0.446 | 0.531 |
| Trend4.8 | A-R | **0.944** | 0.705 | 0.881 | 0.561 | 0.502 | 0.530 | 0.877 | 0.868 | 0.733 | 0.719 | 0.477 | 0.502 | 0.463 | 0.943 | 0.525 | 0.532 |
| | R-A-R | **0.956** | 0.876 | 0.920 | 0.620 | 0.502 | 0.503 | 0.924 | 0.929 | 0.838 | 0.833 | 0.465 | 0.521 | 0.473 | 0.906 | 0.471 | 0.596 |
| | V-ROC | **0.953** | 0.843 | 0.915 | 0.617 | 0.504 | 0.506 | 0.917 | 0.920 | 0.818 | 0.811 | 0.449 | 0.519 | 0.458 | 0.910 | 0.460 | 0.582 |
| Trend7.8 | A-R | **0.917** | 0.821 | 0.827 | 0.578 | 0.542 | 0.501 | 0.845 | 0.858 | 0.800 | 0.780 | 0.509 | 0.501 | 0.488 | 0.880 | 0.509 | 0.530 |
| | R-A-R | **0.948** | 0.882 | 0.884 | 0.627 | 0.537 | 0.504 | 0.893 | 0.903 | 0.842 | 0.850 | 0.531 | 0.528 | 0.509 | 0.879 | 0.483 | 0.625 |
| | V-ROC | **0.940** | 0.874 | 0.872 | 0.616 | 0.530 | 0.502 | 0.883 | 0.895 | 0.835 | 0.839 | 0.509 | 0.525 | 0.490 | 0.878 | 0.471 | 0.603 |
| $1^{st}$ **Count** | | **63** | 0 | 0 | 0 | 0 | 0 | 1 | 0 | 0 | 0 | 0 | 0 | 0 | 2 | 0 | 6 |

Table 7: Average A-P (AUC-PR), R-A-P (R-AUC-PR) and V-PR (VUS-PR) accuracy measures for all datasets. The best results are highlighted in bold, and the second-best results are underlined.

| Dataset | Metric | CATCH | Modern | iTrans | DualTF | ATrans | DC | TsNet | Patch | DLin | NLin | AE | TFAD | Ocsvm | IF | PCA | HBOS |
|---|---|---|---|---|---|---|---|---|---|---|---|---|---|---|---|---|---|
| CICIDS | A-P | 0.002 | 0.001 | 0.002 | 0.002 | 0.001 | 0.002 | 0.002 | 0.001 | 0.002 | 0.001 | 0.001 | **0.003** | 0.001 | 0.003 | 0.001 | 0.003 |
| | R-A-P | 0.002 | 0.002 | 0.002 | 0.001 | 0.001 | 0.001 | 0.002 | 0.002 | 0.002 | 0.002 | 0.002 | 0.001 | 0.002 | **0.364** | 0.002 | 0.001 |
| | V-PR | 0.003 | 0.002 | 0.002 | 0.002 | 0.001 | 0.002 | 0.002 | 0.002 | 0.002 | 0.002 | 0.002 | 0.002 | **0.389** | 0.003 | 0.001 | 0.002 |
| CalIt2 | A-P | 0.114 | 0.054 | 0.106 | 0.057 | 0.045 | 0.035 | 0.078 | **0.116** | 0.097 | 0.054 | 0.084 | 0.030 | 0.095 | 0.080 | 0.073 | 0.080 |
| | R-A-P | **0.124** | 0.070 | 0.111 | 0.088 | 0.089 | 0.087 | 0.092 | 0.115 | 0.097 | 0.073 | 0.096 | 0.060 | 0.109 | 0.095 | 0.106 | 0.113 |
| | V-PR | **0.121** | 0.070 | 0.110 | 0.082 | 0.083 | 0.083 | 0.090 | 0.115 | 0.095 | 0.072 | 0.097 | 0.051 | 0.109 | 0.091 | 0.103 | 0.108 |
| Credit | A-P | 0.101 | 0.088 | 0.042 | 0.023 | 0.007 | 0.002 | 0.091 | 0.089 | 0.087 | 0.040 | 0.025 | 0.002 | 0.053 | 0.074 | 0.029 | **0.173** |
| | R-A-P | 0.053 | 0.054 | 0.024 | 0.009 | 0.002 | 0.002 | 0.056 | 0.054 | 0.053 | 0.053 | 0.020 | 0.002 | 0.038 | 0.040 | 0.009 | **0.081** |
| | V-PR | 0.051 | 0.051 | 0.024 | 0.011 | 0.002 | 0.002 | 0.053 | 0.051 | 0.050 | 0.051 | 0.020 | 0.002 | 0.037 | 0.039 | 0.012 | **0.076** |
| GECCO | A-P | 0.418 | **0.447** | 0.096 | 0.130 | 0.013 | 0.012 | 0.410 | 0.400 | 0.349 | 0.304 | 0.206 | 0.012 | 0.039 | 0.052 | 0.234 | 0.199 |
| | R-A-P | **0.473** | 0.459 | 0.134 | 0.050 | 0.022 | 0.020 | 0.428 | 0.444 | 0.416 | 0.372 | 0.033 | 0.019 | 0.098 | 0.085 | 0.047 | 0.032 |
| | V-PR | **0.465** | 0.461 | 0.128 | 0.049 | 0.022 | 0.019 | 0.429 | 0.439 | 0.406 | 0.363 | 0.033 | 0.018 | 0.101 | 0.083 | 0.046 | 0.033 |
| Genesis | A-P | **0.249** | 0.015 | 0.019 | 0.051 | 0.058 | 0.010 | 0.036 | 0.013 | 0.017 | 0.011 | 0.055 | 0.004 | 0.007 | 0.005 | 0.011 | 0.059 |
| | R-A-P | 0.384 | 0.015 | 0.021 | 0.101 | 0.101 | 0.012 | 0.038 | 0.013 | 0.016 | 0.013 | 0.047 | 0.003 | **0.506** | 0.011 | 0.015 | 0.090 |
| | V-PR | 0.371 | 0.015 | 0.020 | 0.097 | 0.095 | 0.012 | 0.037 | 0.013 | 0.016 | 0.014 | 0.047 | 0.003 | **0.506** | 0.010 | 0.015 | 0.087 |
| MSL | A-P | **0.167** | 0.146 | 0.151 | 0.156 | 0.107 | 0.107 | 0.146 | 0.157 | 0.147 | 0.140 | 0.148 | 0.106 | 0.153 | 0.114 | 0.157 | 0.132 |
| | R-A-P | **0.260** | 0.224 | 0.227 | 0.220 | 0.165 | 0.157 | 0.231 | 0.242 | 0.225 | 0.222 | 0.200 | 0.162 | 0.185 | 0.174 | 0.203 | 0.190 |
| | V-PR | **0.256** | 0.220 | 0.224 | 0.218 | 0.162 | 0.156 | 0.227 | 0.237 | 0.221 | 0.218 | 0.199 | 0.159 | 0.185 | 0.173 | 0.200 | 0.189 |
| NYC | A-P | **0.076** | 0.020 | 0.033 | 0.040 | 0.045 | 0.034 | 0.060 | 0.046 | 0.046 | 0.062 | 0.025 | 0.023 | 0.020 | 0.022 | 0.046 | 0.019 |
| | R-A-P | 0.120 | 0.038 | 0.051 | 0.065 | 0.096 | 0.045 | 0.069 | 0.059 | 0.061 | 0.066 | 0.046 | 0.046 | **0.333** | 0.047 | 0.076 | 0.031 |
| | V-PR | 0.114 | 0.037 | 0.051 | 0.063 | 0.087 | 0.047 | 0.070 | 0.059 | 0.061 | 0.065 | 0.045 | 0.047 | **0.318** | 0.045 | 0.072 | 0.032 |
| PSM | A-P | 0.434 | 0.385 | 0.383 | 0.411 | 0.298 | 0.281 | 0.391 | 0.378 | 0.371 | 0.376 | 0.465 | 0.279 | 0.418 | 0.334 | **0.468** | 0.394 |
| | R-A-P | **0.435** | 0.383 | 0.386 | 0.353 | 0.293 | 0.283 | 0.395 | 0.379 | 0.372 | 0.378 | 0.420 | 0.291 | 0.369 | 0.334 | 0.423 | 0.369 |
| | V-PR | **0.436** | 0.384 | 0.387 | 0.354 | 0.293 | 0.283 | 0.395 | 0.380 | 0.373 | 0.379 | 0.420 | 0.291 | 0.370 | 0.334 | 0.423 | 0.370 |
| SMD | A-P | 0.172 | 0.130 | 0.146 | 0.069 | 0.046 | 0.043 | 0.141 | 0.147 | 0.139 | 0.141 | **0.188** | 0.042 | 0.104 | 0.122 | 0.128 | 0.145 |
| | R-A-P | 0.159 | 0.130 | 0.145 | 0.070 | 0.055 | 0.046 | 0.140 | 0.152 | 0.145 | 0.145 | **0.182** | 0.052 | 0.080 | 0.099 | 0.109 | 0.088 |
| | V-PR | 0.159 | 0.130 | 0.145 | 0.070 | 0.054 | 0.046 | 0.140 | 0.152 | 0.144 | 0.145 | **0.181** | 0.052 | 0.081 | 0.099 | 0.109 | 0.088 |
| SWAT | A-P | 0.166 | 0.093 | 0.084 | 0.143 | 0.121 | 0.121 | 0.107 | 0.085 | 0.131 | 0.081 | 0.713 | 0.123 | 0.170 | 0.093 | 0.726 | **0.738** |
| | R-A-P | 0.251 | 0.138 | 0.122 | 0.172 | 0.111 | 0.154 | 0.178 | 0.125 | 0.170 | 0.113 | 0.446 | 0.165 | 0.535 | 0.133 | 0.533 | **0.586** |
| | V-PR | 0.241 | 0.132 | 0.118 | 0.171 | 0.109 | 0.153 | 0.169 | 0.121 | 0.167 | 0.111 | 0.434 | 0.164 | 0.523 | 0.129 | 0.504 | **0.561** |
| SMAP | A-P | 0.131 | 0.114 | 0.114 | 0.122 | 0.130 | 0.132 | 0.114 | 0.115 | 0.104 | 0.111 | 0.126 | 0.128 | 0.102 | 0.122 | 0.104 | **0.148** |
| | R-A-P | 0.155 | 0.129 | 0.126 | 0.141 | 0.152 | 0.149 | 0.131 | 0.132 | 0.118 | 0.126 | 0.142 | 0.134 | 0.117 | 0.135 | 0.118 | **0.165** |
| | V-PR | 0.155 | 0.129 | 0.126 | 0.141 | 0.152 | 0.150 | 0.131 | 0.132 | 0.118 | 0.127 | 0.143 | 0.133 | 0.117 | 0.136 | 0.118 | **0.166** |
| ASD | A-P | 0.231 | 0.158 | 0.164 | 0.095 | 0.052 | 0.049 | **0.245** | 0.174 | 0.170 | 0.132 | 0.203 | 0.052 | 0.142 | 0.195 | 0.144 | 0.133 |
| | R-A-P | **0.269** | 0.170 | 0.197 | 0.103 | 0.082 | 0.090 | 0.254 | 0.202 | 0.191 | 0.139 | 0.208 | 0.089 | 0.102 | 0.197 | 0.154 | 0.132 |
| | V-PR | **0.261** | 0.170 | 0.194 | 0.103 | 0.080 | 0.090 | 0.253 | 0.200 | 0.188 | 0.139 | 0.208 | 0.091 | 0.103 | 0.197 | 0.149 | 0.130 |
| Contextual4.9 | A-P | 0.754 | 0.090 | 0.754 | 0.086 | 0.070 | 0.054 | **0.773** | 0.640 | 0.274 | 0.055 | 0.732 | 0.049 | 0.522 | 0.323 | 0.066 | 0.044 |
| | R-A-P | 0.585 | 0.060 | 0.585 | 0.075 | 0.055 | 0.055 | **0.614** | 0.432 | 0.135 | 0.054 | 0.556 | 0.025 | 0.293 | 0.225 | 0.054 | 0.047 |
| | V-PR | 0.562 | 0.074 | 0.559 | 0.094 | 0.068 | 0.067 | **0.586** | 0.423 | 0.148 | 0.066 | 0.534 | 0.047 | 0.292 | 0.230 | 0.068 | 0.057 |
| Contextual7.2 | A-P | **0.770** | 0.106 | 0.756 | 0.087 | 0.099 | 0.074 | 0.767 | 0.672 | 0.309 | 0.090 | 0.721 | 0.109 | 0.542 | 0.374 | 0.105 | 0.075 |
| | R-A-P | **0.612** | 0.083 | 0.594 | 0.078 | 0.078 | 0.079 | 0.608 | 0.477 | 0.166 | 0.082 | 0.546 | 0.082 | 0.324 | 0.260 | 0.079 | 0.067 |
| | V-PR | **0.594** | 0.100 | 0.579 | 0.097 | 0.094 | 0.094 | 0.590 | 0.469 | 0.183 | 0.090 | 0.533 | 0.097 | 0.333 | 0.271 | 0.098 | 0.081 |
| Global4.8 | A-P | 0.978 | 0.552 | 0.939 | 0.087 | 0.064 | 0.051 | 0.959 | 0.959 | 0.927 | 0.099 | 0.971 | 0.098 | **0.981** | 0.435 | 0.136 | 0.113 |
| | R-A-P | 0.957 | 0.363 | 0.881 | 0.068 | 0.055 | 0.050 | 0.918 | 0.918 | 0.858 | 0.072 | 0.942 | 0.051 | **0.961** | 0.404 | 0.110 | 0.063 |
| | V-PR | 0.901 | 0.361 | 0.834 | 0.092 | 0.069 | 0.062 | 0.866 | 0.864 | 0.808 | 0.089 | 0.884 | 0.062 | **0.903** | 0.394 | 0.140 | 0.074 |
| Global7.2 | A-P | 0.974 | 0.517 | 0.921 | 0.088 | 0.098 | 0.074 | 0.947 | 0.960 | 0.912 | 0.134 | 0.973 | 0.074 | **0.977** | 0.492 | 0.182 | 0.106 |
| | R-A-P | 0.949 | 0.346 | 0.854 | 0.082 | 0.084 | 0.074 | 0.902 | 0.923 | 0.834 | 0.106 | 0.948 | 0.060 | **0.953** | 0.429 | 0.145 | 0.090 |
| | V-PR | **0.905** | 0.347 | 0.818 | 0.099 | 0.101 | 0.089 | 0.860 | 0.876 | 0.796 | 0.122 | 0.899 | 0.067 | 0.904 | 0.426 | 0.182 | 0.107 |
| Seasonal4.8 | A-P | **0.975** | 0.055 | 0.865 | 0.476 | 0.158 | 0.128 | 0.889 | 0.776 | 0.622 | 0.063 | 0.884 | 0.084 | 0.732 | 0.727 | 0.061 | 0.058 |
| | R-A-P | **0.994** | 0.087 | 0.739 | 0.719 | 0.162 | 0.223 | 0.750 | 0.681 | 0.540 | 0.099 | 0.718 | 0.152 | 0.604 | 0.679 | 0.092 | 0.088 |
| | V-PR | **0.987** | 0.084 | 0.755 | 0.676 | 0.165 | 0.210 | 0.763 | 0.695 | 0.551 | 0.099 | 0.737 | 0.138 | 0.621 | 0.679 | 0.091 | 0.085 |
| Seasonal7.7 | A-P | **0.963** | 0.092 | 0.881 | 0.083 | 0.164 | 0.148 | 0.906 | 0.773 | 0.645 | 0.109 | 0.900 | 0.133 | 0.734 | 0.733 | 0.116 | 0.080 |
| | R-A-P | **0.978** | 0.148 | 0.808 | 0.139 | 0.192 | 0.260 | 0.805 | 0.723 | 0.599 | 0.163 | 0.758 | 0.195 | 0.645 | 0.752 | 0.169 | 0.132 |
| | V-PR | **0.963** | 0.141 | 0.819 | 0.134 | 0.194 | 0.246 | 0.823 | 0.732 | 0.605 | 0.159 | 0.782 | 0.185 | 0.658 | 0.743 | 0.160 | 0.132 |
| Shapelet4.9 | A-P | **0.774** | 0.065 | 0.633 | 0.124 | 0.066 | 0.142 | 0.683 | 0.499 | 0.382 | 0.053 | 0.659 | 0.102 | 0.482 | 0.282 | 0.094 | 0.043 |
| | R-A-P | **0.940** | 0.089 | 0.564 | 0.199 | 0.092 | 0.153 | 0.588 | 0.463 | 0.314 | 0.090 | 0.556 | 0.144 | 0.410 | 0.319 | 0.138 | 0.070 |
| | V-PR | **0.887** | 0.089 | 0.571 | 0.195 | 0.090 | 0.147 | 0.599 | 0.465 | 0.322 | 0.089 | 0.568 | 0.137 | 0.422 | 0.311 | 0.135 | 0.071 |
| Shapelet7.4 | A-P | **0.774** | 0.087 | 0.651 | 0.089 | 0.109 | 0.133 | 0.708 | 0.542 | 0.449 | 0.089 | 0.715 | 0.118 | 0.537 | 0.295 | 0.136 | 0.062 |
| | R-A-P | **0.921** | 0.132 | 0.608 | 0.149 | 0.143 | 0.169 | 0.639 | 0.528 | 0.416 | 0.137 | 0.619 | 0.175 | 0.488 | 0.352 | 0.241 | 0.104 |
| | V-PR | **0.877** | 0.133 | 0.616 | 0.148 | 0.137 | 0.164 | 0.653 | 0.534 | 0.422 | 0.136 | 0.634 | 0.171 | 0.496 | 0.343 | 0.224 | 0.101 |
| Mixture5.7 | A-P | 0.368 | 0.196 | 0.238 | 0.060 | 0.060 | 0.088 | 0.268 | 0.273 | 0.172 | 0.104 | 0.068 | 0.059 | 0.050 | **0.591** | 0.082 | 0.051 |
| | R-A-P | 0.509 | 0.276 | 0.349 | 0.110 | 0.092 | 0.146 | 0.363 | 0.369 | 0.204 | 0.160 | 0.091 | 0.163 | 0.125 | **0.579** | 0.092 | 0.122 |
| | V-PR | 0.477 | 0.264 | 0.339 | 0.106 | 0.090 | 0.143 | 0.352 | 0.356 | 0.197 | 0.157 | 0.089 | 0.154 | 0.122 | **0.577** | 0.090 | 0.125 |
| Mixture8.9 | A-P | 0.355 | 0.182 | 0.311 | 0.091 | 0.091 | 0.133 | 0.365 | 0.298 | 0.197 | 0.167 | 0.128 | 0.091 | 0.090 | **0.593** | 0.090 | 0.089 |
| | R-A-P | 0.490 | 0.253 | 0.386 | 0.160 | 0.152 | 0.199 | 0.403 | 0.364 | 0.239 | 0.243 | 0.165 | 0.145 | 0.250 | **0.627** | 0.154 | 0.226 |
| | V-PR | 0.468 | 0.247 | 0.375 | 0.157 | 0.150 | 0.192 | 0.393 | 0.357 | 0.233 | 0.236 | 0.162 | 0.149 | 0.246 | **0.619** | 0.151 | 0.221 |
| Trend4.8 | A-P | 0.377 | 0.215 | 0.254 | 0.062 | 0.048 | 0.063 | 0.274 | 0.228 | 0.155 | 0.133 | 0.056 | 0.058 | 0.044 | **0.615** | 0.054 | 0.054 |
| | R-A-P | 0.538 | 0.343 | 0.368 | 0.093 | 0.092 | 0.093 | 0.400 | 0.385 | 0.241 | 0.223 | 0.090 | 0.091 | 0.330 | **0.589** | 0.092 | 0.239 |
| | V-PR | 0.516 | 0.329 | 0.360 | 0.092 | 0.097 | 0.092 | 0.389 | 0.375 | 0.237 | 0.216 | 0.089 | 0.091 | 0.318 | **0.588** | 0.091 | 0.233 |
| Trend7.8 | A-P | 0.510 | **0.520** | 0.327 | 0.126 | 0.106 | 0.078 | 0.384 | 0.501 | 0.459 | 0.359 | 0.091 | 0.088 | 0.075 | 0.511 | 0.075 | 0.082 |
| | R-A-P | **0.665** | 0.514 | 0.399 | 0.173 | 0.171 | 0.128 | 0.435 | 0.521 | 0.429 | 0.391 | 0.154 | 0.157 | 0.261 | 0.550 | 0.135 | 0.250 |
| | V-PR | **0.637** | 0.515 | 0.386 | 0.169 | 0.166 | 0.125 | 0.424 | 0.512 | 0.429 | 0.389 | 0.152 | 0.150 | 0.252 | 0.543 | 0.133 | 0.240 |
| $1^{st}$ **Count** | | **31** | 2 | 0 | 0 | 0 | 0 | 4 | 1 | 0 | 0 | 3 | 1 | 11 | 9 | 1 | 9 |

Table 8: Average P (Precision), R (Recall) and F1 (F1-score) accuracy measures for all datasets. The best results are highlighted in bold, and the second-best results are underlined.

| Dataset | Metric | CATCH | Modern | iTrans | DualTF | ATrans | DC | TsNet | Patch | DLin | NLin | AE | TFAD | Ocsvm | IF | PCA | HBOS |
|---|---|---|---|---|---|---|---|---|---|---|---|---|---|---|---|---|---|
| CICIDS | P | **0.003** | 0.001 | 0.002 | 0.001 | 0.001 | 0.001 | 0.001 | 0.001 | 0.002 | 0.001 | 0.000 | 0.001 | 0.001 | 0.000 | 0.001 | 0.000 |
| | R | 0.759 | 0.278 | 0.481 | 0.595 | 0.253 | 0.266 | 0.354 | 0.380 | 0.506 | 0.380 | 0.000 | 0.038 | **0.987** | 0.063 | 0.405 | 0.013 |
| | F1 | **0.006** | 0.002 | 0.004 | 0.003 | 0.002 | 0.002 | 0.003 | 0.003 | 0.004 | 0.003 | 0.000 | 0.001 | 0.002 | 0.001 | 0.002 | 0.000 |
| CalIt2 | P | **0.138** | 0.074 | 0.124 | 0.073 | 0.085 | 0.034 | 0.104 | 0.091 | 0.083 | 0.066 | 0.067 | 0.038 | 0.089 | 0.125 | 0.084 | 0.075 |
| | R | 0.162 | 0.243 | 0.230 | 0.176 | 0.095 | 0.162 | 0.108 | 0.311 | 0.284 | 0.324 | 0.027 | 0.041 | **0.378** | 0.014 | 0.297 | 0.257 |
| | F1 | 0.149 | 0.114 | **0.161** | 0.103 | 0.090 | 0.056 | 0.106 | 0.141 | 0.128 | 0.109 | 0.038 | 0.039 | 0.144 | 0.024 | 0.131 | 0.117 |
| Credit | P | 0.059 | 0.058 | 0.026 | 0.003 | 0.001 | 0.001 | 0.058 | 0.058 | 0.057 | 0.057 | 0.047 | 0.005 | 0.013 | **0.156** | 0.009 | 0.009 |
| | R | 0.758 | 0.740 | 0.843 | 0.596 | 0.139 | 0.238 | 0.740 | 0.749 | 0.726 | 0.744 | 0.135 | 0.018 | 0.901 | 0.283 | 0.731 | **0.915** |
| | F1 | 0.110 | 0.107 | 0.051 | 0.006 | 0.003 | 0.003 | 0.107 | 0.107 | 0.105 | 0.107 | 0.069 | 0.007 | 0.026 | **0.201** | 0.017 | 0.018 |
| GECCO | P | 0.380 | 0.373 | 0.173 | 0.009 | 0.012 | 0.008 | 0.379 | **0.475** | 0.332 | 0.384 | 0.021 | 0.009 | 0.011 | 0.214 | 0.014 | 0.014 |
| | R | 0.818 | 0.779 | 0.207 | 0.340 | 0.008 | 0.008 | 0.804 | 0.585 | 0.781 | 0.460 | 0.215 | 0.004 | **0.993** | 0.012 | 0.988 | 0.542 |
| | F1 | 0.518 | 0.504 | 0.189 | 0.018 | 0.010 | 0.008 | 0.516 | **0.524** | 0.466 | 0.418 | 0.039 | 0.006 | 0.022 | 0.023 | 0.027 | 0.027 |
| Genesis | P | 0.116 | 0.015 | 0.065 | 0.066 | 0.053 | 0.016 | **0.119** | 0.075 | 0.055 | 0.055 | 0.047 | 0.000 | 0.007 | 0.006 | 0.017 | 0.017 |
| | R | 0.160 | 0.120 | 0.180 | 0.500 | 0.960 | 0.020 | 0.200 | 0.200 | 0.140 | 0.080 | 0.120 | 0.000 | **1.000** | 0.020 | 0.480 | 0.980 |
| | F1 | 0.134 | 0.027 | 0.095 | 0.116 | 0.100 | 0.018 | **0.149** | 0.109 | 0.079 | 0.022 | 0.068 | 0.000 | 0.014 | 0.009 | 0.033 | 0.033 |
| MSL | P | 0.185 | 0.166 | 0.158 | **0.248** | 0.143 | 0.181 | 0.166 | 0.194 | 0.173 | 0.193 | 0.219 | 0.107 | 0.128 | 0.180 | 0.130 | 0.127 |
| | R | 0.117 | 0.090 | 0.089 | 0.019 | 0.008 | 0.008 | 0.093 | 0.066 | 0.093 | 0.067 | 0.014 | 0.005 | 0.125 | 0.014 | **0.137** | 0.128 |
| | F1 | **0.143** | 0.117 | 0.114 | 0.035 | 0.015 | 0.016 | 0.119 | 0.098 | 0.121 | 0.100 | 0.026 | 0.010 | 0.126 | 0.025 | 0.133 | 0.127 |
| NYC | P | **1.000** | 0.000 | 0.034 | 0.111 | 0.333 | 0.250 | 0.000 | 0.000 | 0.000 | 0.000 | 0.059 | 0.000 | 0.022 | 0.056 | 0.059 | 0.000 |
| | R | 0.010 | 0.000 | 0.293 | 0.010 | 0.010 | 0.010 | 0.000 | 0.000 | 0.000 | 0.000 | 0.010 | 0.000 | **0.980** | 0.010 | 0.222 | 0.000 |
| | F1 | 0.020 | 0.000 | 0.061 | 0.019 | 0.020 | 0.019 | 0.000 | 0.000 | 0.000 | 0.000 | 0.017 | 0.000 | 0.043 | 0.017 | **0.094** | 0.000 |
| PSM | P | 0.624 | 0.653 | 0.600 | 0.461 | 0.396 | 0.278 | 0.591 | 0.653 | 0.658 | 0.644 | 0.444 | 0.282 | 0.548 | **0.802** | 0.427 | 0.544 |
| | R | 0.064 | 0.047 | 0.058 | 0.559 | 0.025 | 0.243 | 0.048 | 0.047 | 0.025 | 0.047 | 0.110 | 0.005 | 0.299 | 0.004 | 0.421 | 0.302 |
| | F1 | 0.116 | 0.089 | 0.105 | **0.506** | 0.046 | 0.259 | 0.088 | 0.088 | 0.047 | 0.087 | 0.176 | 0.010 | 0.387 | 0.007 | 0.424 | 0.389 |
| SMD | P | 0.194 | 0.151 | 0.162 | 0.091 | 0.112 | 0.042 | 0.176 | 0.185 | 0.197 | 0.201 | **0.706** | 0.043 | 0.107 | 0.454 | 0.117 | 0.095 |
| | R | 0.305 | 0.145 | 0.257 | 0.257 | 0.022 | 0.201 | 0.181 | 0.173 | 0.124 | 0.126 | 0.007 | 0.005 | 0.304 | 0.020 | **0.359** | 0.332 |
| | F1 | **0.237** | 0.148 | 0.198 | 0.135 | 0.037 | 0.069 | 0.178 | 0.179 | 0.152 | 0.155 | 0.014 | 0.009 | 0.158 | 0.039 | 0.176 | 0.148 |
| SWAT | P | 0.195 | 0.119 | 0.088 | 0.143 | 0.105 | 0.107 | **0.329** | 0.110 | 0.308 | 0.088 | 0.151 | 0.135 | 0.130 | 0.128 | 0.152 | 0.290 |
| | R | 0.056 | 0.055 | 0.074 | 0.404 | 0.013 | 0.016 | 0.044 | 0.048 | 0.375 | 0.074 | 0.302 | 0.001 | **0.990** | 0.002 | 0.907 | 0.742 |
| | F1 | 0.087 | 0.075 | 0.080 | 0.212 | 0.023 | 0.028 | 0.077 | 0.067 | 0.338 | 0.080 | 0.201 | 0.001 | 0.230 | 0.004 | 0.261 | **0.417** |
| SMAP | P | **0.236** | 0.096 | 0.086 | 0.119 | 0.154 | 0.121 | 0.094 | 0.086 | 0.086 | 0.084 | 0.084 | 0.125 | 0.078 | 0.092 | 0.084 | 0.092 |
| | R | 0.035 | 0.171 | 0.151 | **0.256** | 0.047 | 0.005 | 0.167 | 0.151 | 0.152 | 0.148 | 0.001 | 0.005 | 0.042 | 0.002 | 0.045 | 0.045 |
| | F1 | 0.061 | 0.123 | 0.110 | **0.163** | 0.072 | 0.010 | 0.121 | 0.109 | 0.110 | 0.107 | 0.001 | 0.010 | 0.055 | 0.004 | 0.059 | 0.060 |
| ASD | P | 0.309 | 0.237 | 0.230 | 0.116 | 0.076 | 0.069 | 0.348 | 0.261 | 0.301 | 0.207 | 0.222 | 0.077 | 0.063 | **0.455** | 0.088 | 0.100 |
| | R | 0.225 | 0.178 | 0.253 | 0.183 | 0.022 | 0.043 | **0.299** | 0.147 | 0.135 | 0.177 | 0.101 | 0.010 | 0.160 | 0.063 | 0.265 | 0.242 |
| | F1 | 0.219 | 0.158 | 0.184 | 0.107 | 0.031 | 0.047 | **0.258** | 0.177 | 0.163 | 0.146 | 0.129 | 0.017 | 0.085 | 0.108 | 0.125 | 0.131 |
| Contextual4.9 | P | 0.584 | 0.067 | 0.555 | 0.044 | 0.051 | 0.049 | 0.101 | 0.264 | 0.446 | 0.119 | **0.670** | 0.041 | 0.234 | 0.577 | 0.072 | 0.029 |
| | R | **0.733** | 0.340 | 0.696 | 0.235 | 0.356 | 0.364 | 0.559 | 0.713 | 0.656 | 0.656 | 0.591 | 0.069 | 0.559 | 0.393 | 0.174 | 0.053 |
| | F1 | **0.650** | 0.112 | 0.618 | 0.075 | 0.089 | 0.086 | 0.172 | 0.385 | 0.531 | 0.201 | 0.628 | 0.051 | 0.330 | 0.467 | 0.102 | 0.037 |
| Contextual7.2 | P | 0.744 | 0.095 | 0.548 | 0.085 | 0.080 | 0.067 | 0.157 | 0.352 | 0.373 | 0.160 | **0.775** | 0.084 | 0.306 | 0.633 | 0.103 | 0.053 |
| | R | 0.686 | 0.336 | **0.719** | 0.461 | 0.297 | 0.342 | 0.600 | 0.700 | 0.717 | 0.650 | 0.556 | 0.097 | 0.556 | 0.350 | 0.197 | 0.064 |
| | F1 | **0.714** | 0.148 | 0.622 | 0.143 | 0.126 | 0.112 | 0.249 | 0.468 | 0.491 | 0.257 | 0.647 | 0.090 | 0.394 | 0.451 | 0.135 | 0.058 |
| Global4.8 | P | 0.936 | 0.277 | **1.000** | 0.071 | 0.048 | 0.046 | 0.900 | 1.000 | 0.977 | 0.290 | 0.747 | 0.046 | 0.331 | 0.766 | 0.109 | 0.086 |
| | R | 0.921 | 0.721 | 0.875 | 0.204 | 0.854 | 0.329 | 0.858 | 0.888 | 0.825 | 0.875 | 0.875 | 0.079 | **0.996** | 0.750 | 0.738 | 0.175 |
| | F1 | 0.929 | 0.400 | 0.933 | 0.105 | 0.090 | 0.080 | 0.878 | **0.940** | 0.921 | 0.429 | 0.806 | 0.058 | 0.496 | 0.758 | 0.190 | 0.116 |
| Global7.2 | P | 0.804 | 0.281 | 0.764 | 0.075 | 0.074 | 0.077 | **0.984** | 0.873 | 0.915 | 0.401 | 0.871 | 0.060 | 0.458 | 0.834 | 0.134 | 0.062 |
| | R | 0.925 | 0.713 | 0.883 | 0.738 | 0.836 | 0.334 | 0.836 | 0.916 | 0.872 | 0.802 | 0.808 | 0.070 | **0.983** | 0.727 | 0.721 | 0.084 |
| | F1 | 0.860 | 0.403 | 0.819 | 0.136 | 0.135 | 0.125 | **0.904** | 0.894 | 0.893 | 0.534 | 0.838 | 0.065 | 0.625 | 0.777 | 0.226 | 0.071 |
| Seasonal4.8 | P | 0.994 | 0.065 | **1.000** | 0.653 | 0.889 | 0.531 | 1.000 | 1.000 | 1.000 | 0.984 | 0.630 | 0.000 | 0.269 | 0.610 | 0.062 | 0.043 |
| | R | **0.747** | 0.141 | 0.444 | 0.195 | 0.100 | 0.071 | 0.461 | 0.456 | 0.461 | 0.253 | 0.361 | 0.000 | 0.743 | 0.311 | 0.141 | 0.087 |
| | F1 | **0.853** | 0.089 | 0.615 | 0.300 | 0.179 | 0.125 | 0.631 | 0.627 | 0.631 | 0.403 | 0.459 | 0.000 | 0.395 | 0.412 | 0.086 | 0.058 |
| Seasonal7.7 | P | 0.961 | 0.080 | **1.000** | 0.092 | 0.968 | 0.597 | 0.995 | 0.966 | 1.000 | 0.963 | 0.692 | 0.323 | 0.379 | 0.766 | 0.099 | 0.103 |
| | R | **0.765** | 0.160 | 0.530 | 0.041 | 0.078 | 0.103 | 0.522 | 0.506 | 0.543 | 0.271 | 0.359 | 0.080 | 0.726 | 0.313 | 0.134 | 0.124 |
| | F1 | **0.852** | 0.107 | 0.693 | 0.057 | 0.144 | 0.176 | 0.685 | 0.664 | 0.704 | 0.423 | 0.473 | 0.128 | 0.498 | 0.444 | 0.114 | 0.113 |
| Shapelet4.9 | P | 0.979 | 0.064 | 0.845 | 0.093 | 0.438 | 0.315 | **0.988** | 0.519 | 0.898 | 0.418 | 0.479 | 0.043 | 0.203 | 0.542 | 0.075 | 0.026 |
| | R | 0.384 | 0.196 | 0.380 | 0.020 | 0.057 | 0.094 | 0.335 | 0.453 | 0.396 | 0.208 | 0.286 | 0.004 | **0.506** | 0.131 | 0.180 | 0.053 |
| | F1 | **0.551** | 0.096 | 0.524 | 0.033 | 0.101 | 0.145 | 0.500 | 0.484 | 0.550 | 0.278 | 0.358 | 0.007 | 0.290 | 0.211 | 0.105 | 0.035 |
| Shapelet7.4 | P | **0.994** | 0.080 | 0.785 | 0.132 | 0.302 | 0.151 | 0.925 | 0.696 | 0.741 | 0.283 | 0.583 | 0.235 | 0.296 | 0.489 | 0.147 | 0.030 |
| | R | 0.418 | 0.272 | 0.423 | 0.132 | 0.051 | 0.299 | 0.367 | 0.407 | 0.307 | 0.275 | 0.075 | | 0.523 | 0.119 | 0.245 | 0.038 |
| | F1 | **0.588** | 0.124 | 0.550 | 0.132 | 0.088 | 0.201 | 0.525 | 0.514 | 0.558 | 0.295 | 0.374 | 0.114 | 0.378 | 0.191 | 0.184 | 0.034 |
| Mixture5.7 | P | 0.321 | 0.211 | 0.260 | 0.058 | 0.100 | 0.370 | 0.287 | 0.272 | 0.071 | 0.220 | 0.066 | 0.113 | 0.062 | **0.627** | 0.062 | 0.049 |
| | R | 0.627 | 0.314 | 0.544 | 0.324 | 0.031 | 0.059 | 0.551 | 0.557 | 0.784 | 0.415 | 0.334 | 0.056 | **0.944** | 0.223 | 0.944 | 0.707 |
| | F1 | **0.425** | 0.252 | 0.352 | 0.098 | 0.048 | 0.102 | 0.377 | 0.366 | 0.131 | 0.288 | 0.111 | 0.075 | 0.117 | 0.329 | 0.117 | 0.091 |
| Mixture8.9 | P | 0.274 | 0.200 | 0.290 | 0.089 | 0.055 | 0.452 | 0.287 | 0.275 | 0.118 | 0.225 | 0.092 | 0.089 | 0.092 | **0.704** | 0.092 | 0.090 |
| | R | 0.703 | 0.234 | 0.688 | 0.333 | 0.020 | 0.063 | 0.672 | 0.670 | 0.562 | 0.492 | 0.333 | 0.063 | 0.998 | 0.225 | **1.000** | 0.854 |
| | F1 | 0.394 | 0.216 | **0.408** | 0.140 | 0.030 | 0.110 | 0.402 | 0.390 | 0.195 | 0.309 | 0.145 | 0.074 | 0.168 | 0.341 | 0.169 | 0.162 |
| Trend4.8 | P | 0.215 | 0.171 | 0.231 | 0.062 | 0.034 | 0.167 | 0.243 | 0.212 | 0.073 | 0.158 | 0.048 | 0.000 | 0.049 | **0.600** | 0.049 | 0.049 |
| | R | 0.888 | 0.427 | 0.544 | 0.207 | 0.008 | 0.029 | 0.386 | 0.461 | 0.643 | 0.668 | 0.324 | 0.000 | 0.996 | 0.224 | **1.000** | 0.822 |
| | F1 | **0.346** | 0.245 | 0.324 | 0.096 | 0.013 | 0.049 | 0.299 | 0.290 | 0.131 | 0.255 | 0.083 | 0.000 | 0.093 | 0.326 | 0.093 | 0.092 |
| Trend7.8 | P | 0.438 | 0.241 | 0.211 | 0.120 | 0.107 | 0.049 | 0.231 | 0.331 | 0.130 | 0.274 | 0.076 | 0.127 | 0.078 | **0.641** | 0.078 | 0.084 |
| | R | 0.501 | 0.653 | 0.620 | 0.278 | 0.033 | 0.008 | 0.668 | 0.555 | 0.717 | 0.455 | 0.324 | 0.064 | 0.995 | 0.129 | **1.000** | 0.787 |
| | F1 | **0.468** | 0.352 | 0.314 | 0.168 | 0.051 | 0.013 | 0.343 | 0.415 | 0.220 | 0.342 | 0.123 | 0.085 | 0.144 | 0.214 | 0.145 | 0.152 |
| 1st Count | | **20** | 0 | 6 | 5 | 0 | 0 | 8 | 3 | 0 | 0 | 3 | 0 | 11 | 8 | 6 | 2 |

Table 9: Average R-R (Range-Recall), R-P (Range-Precision) and R-F (Range-F1-score) accuracy measures for all datasets. The best results are highlighted in bold, and the second-best results are underlined.

| Dataset | Metric | CATCH | Modern | iTrans | DualTF | ATrans | DC | TsNet | Patch | DLin | NLin | AE | TFAD | Ocsvm | IF | PCA | HBOS |
|---|---|---|---|---|---|---|---|---|---|---|---|---|---|---|---|---|---|
| CICIDS | R-R | 0.786 | 0.277 | 0.494 | 0.586 | 0.243 | 0.283 | 0.369 | 0.389 | 0.523 | 0.386 | 0.000 | 0.043 | **0.986** | 0.057 | 0.394 | 0.009 |
| | R-P | **0.004** | 0.002 | 0.002 | 0.001 | 0.001 | 0.001 | 0.002 | 0.002 | 0.003 | 0.001 | 0.000 | 0.001 | 0.004 | 0.001 | 0.001 | 0.000 |
| | R-F | 0.007 | 0.003 | 0.004 | 0.003 | 0.003 | 0.002 | 0.003 | 0.004 | 0.005 | 0.003 | 0.000 | 0.001 | **0.008** | 0.001 | 0.003 | 0.000 |
| CalIt2 | R-R | 0.278 | 0.318 | 0.344 | 0.209 | 0.161 | 0.205 | 0.161 | **0.424** | 0.315 | 0.325 | 0.055 | 0.074 | 0.359 | 0.031 | 0.280 | 0.309 |
| | R-P | **0.139** | 0.069 | 0.113 | 0.056 | 0.105 | 0.031 | 0.095 | 0.077 | 0.091 | 0.064 | 0.067 | 0.038 | 0.096 | 0.125 | 0.099 | 0.086 |
| | R-F | **0.185** | 0.114 | 0.170 | 0.088 | 0.127 | 0.053 | 0.120 | 0.131 | 0.142 | 0.107 | 0.060 | 0.051 | 0.152 | 0.049 | 0.147 | 0.134 |
| Credit | R-R | 0.735 | 0.716 | 0.828 | 0.564 | 0.150 | 0.234 | 0.716 | 0.725 | 0.701 | 0.721 | 0.135 | 0.020 | 0.892 | 0.292 | 0.709 | **0.907** |
| | R-P | 0.054 | 0.052 | 0.024 | 0.002 | 0.003 | 0.002 | 0.051 | 0.052 | 0.049 | 0.052 | 0.047 | 0.005 | 0.012 | **0.156** | 0.008 | 0.009 |
| | R-F | 0.101 | 0.097 | 0.047 | 0.004 | 0.007 | 0.003 | 0.096 | 0.096 | 0.092 | 0.097 | 0.070 | 0.007 | 0.023 | **0.204** | 0.015 | 0.017 |
| GECCO | R-R | 0.795 | 0.644 | 0.266 | 0.146 | 0.063 | 0.040 | 0.782 | 0.366 | 0.790 | 0.274 | 0.188 | 0.030 | 0.953 | 0.030 | **0.986** | 0.361 |
| | R-P | 0.065 | 0.086 | 0.111 | 0.039 | 0.013 | 0.008 | 0.053 | 0.289 | 0.042 | **0.296** | 0.021 | 0.009 | 0.006 | 0.214 | 0.041 | 0.016 |
| | R-F | 0.119 | 0.152 | 0.156 | 0.062 | 0.022 | 0.013 | 0.099 | 0.323 | 0.080 | 0.285 | 0.038 | 0.013 | 0.011 | 0.052 | 0.079 | 0.031 |
| Genesis | R-R | 0.497 | 0.174 | 0.507 | 0.550 | 0.855 | 0.079 | 0.211 | 0.385 | 0.180 | 0.162 | 0.356 | 0.000 | **1.000** | 0.077 | 0.325 | 0.861 |
| | R-P | **0.119** | 0.014 | 0.057 | 0.030 | 0.005 | 0.016 | 0.086 | 0.067 | 0.059 | 0.013 | 0.047 | 0.000 | 0.003 | 0.006 | 0.035 | 0.002 |
| | R-F | **0.192** | 0.026 | 0.102 | 0.058 | 0.010 | 0.027 | 0.122 | 0.114 | 0.089 | 0.024 | 0.083 | 0.000 | 0.005 | 0.011 | 0.063 | 0.005 |
| MSL | R-R | **0.241** | 0.194 | 0.182 | 0.049 | 0.150 | 0.126 | 0.224 | 0.176 | 0.202 | 0.162 | 0.110 | 0.138 | 0.202 | 0.112 | 0.237 | 0.199 |
| | R-P | 0.150 | 0.129 | 0.125 | **0.267** | 0.143 | 0.179 | 0.130 | 0.136 | 0.130 | 0.137 | 0.219 | 0.107 | 0.098 | 0.180 | 0.113 | 0.119 |
| | R-F | **0.185** | 0.155 | 0.148 | 0.084 | 0.146 | 0.148 | 0.164 | 0.153 | 0.158 | 0.149 | 0.146 | 0.120 | 0.132 | 0.138 | 0.153 | 0.149 |
| NYC | R-R | 0.208 | 0.000 | 0.223 | 0.208 | 0.208 | 0.208 | 0.000 | 0.000 | 0.000 | 0.000 | 0.208 | 0.000 | 0.461 | 0.208 | 0.289 | 0.000 |
| | R-P | **1.000** | 0.000 | 0.023 | 0.143 | 0.333 | 0.250 | 0.000 | 0.000 | 0.000 | 0.000 | 0.059 | 0.000 | 0.040 | 0.056 | 0.053 | 0.000 |
| | R-F | 0.344 | 0.000 | 0.042 | 0.169 | 0.256 | 0.227 | 0.000 | 0.000 | 0.000 | 0.000 | 0.092 | 0.000 | 0.074 | 0.088 | 0.089 | 0.000 |
| PSM | R-R | 0.450 | 0.376 | 0.470 | 0.410 | 0.133 | 0.211 | 0.455 | 0.370 | 0.399 | 0.359 | 0.230 | 0.054 | 0.133 | 0.139 | **0.505** | 0.265 |
| | R-P | 0.557 | 0.553 | 0.533 | 0.456 | 0.374 | 0.274 | 0.537 | 0.557 | 0.584 | 0.587 | 0.444 | 0.282 | 0.385 | **0.802** | 0.459 | 0.467 |
| | R-F | 0.498 | 0.448 | 0.499 | 0.432 | 0.197 | 0.238 | 0.493 | 0.444 | 0.474 | 0.446 | 0.303 | 0.091 | 0.198 | 0.237 | 0.481 | 0.338 |
| SMD | R-R | **0.478** | 0.378 | 0.360 | 0.187 | 0.126 | 0.276 | 0.385 | 0.426 | 0.400 | 0.384 | 0.079 | 0.053 | 0.323 | 0.122 | 0.404 | 0.297 |
| | R-P | 0.095 | 0.092 | 0.115 | 0.055 | 0.101 | 0.042 | 0.110 | 0.121 | 0.124 | 0.131 | **0.706** | 0.043 | 0.067 | 0.454 | 0.123 | 0.062 |
| | R-F | 0.158 | 0.148 | 0.175 | 0.085 | 0.112 | 0.073 | 0.171 | 0.189 | 0.189 | 0.171 | 0.143 | 0.047 | 0.111 | 0.192 | 0.188 | 0.103 |
| SWAT | R-R | 0.184 | 0.200 | 0.203 | 0.220 | 0.203 | 0.203 | 0.201 | 0.190 | 0.204 | 0.203 | 0.173 | 0.115 | **0.924** | 0.133 | 0.719 | 0.442 |
| | R-P | 0.124 | 0.072 | 0.066 | 0.122 | 0.115 | 0.120 | 0.140 | 0.079 | **0.233** | 0.072 | 0.151 | 0.135 | 0.014 | 0.128 | 0.071 | 0.067 |
| | R-F | 0.148 | 0.106 | 0.099 | 0.157 | 0.147 | 0.151 | 0.165 | 0.112 | **0.217** | 0.106 | 0.161 | 0.124 | 0.027 | 0.130 | 0.129 | 0.116 |
| SMAP | R-R | 0.182 | 0.276 | 0.211 | 0.226 | 0.204 | 0.140 | **0.298** | 0.234 | 0.280 | 0.230 | 0.054 | 0.118 | 0.134 | 0.092 | 0.137 | 0.128 |
| | R-P | **0.164** | 0.087 | 0.088 | 0.110 | 0.120 | 0.121 | 0.084 | 0.079 | 0.107 | 0.080 | 0.084 | 0.125 | 0.079 | 0.092 | 0.086 | 0.075 |
| | R-F | **0.173** | 0.133 | 0.124 | 0.148 | 0.151 | 0.130 | 0.131 | 0.118 | 0.155 | 0.118 | 0.066 | 0.122 | 0.100 | 0.092 | 0.094 | 0.094 |
| ASD | R-R | 0.303 | 0.255 | 0.291 | 0.145 | 0.097 | 0.116 | **0.328** | 0.265 | 0.232 | 0.215 | 0.174 | 0.062 | 0.166 | 0.196 | 0.312 | 0.217 |
| | R-P | 0.212 | 0.185 | 0.165 | 0.165 | 0.081 | 0.070 | 0.218 | 0.182 | 0.232 | 0.181 | 0.222 | 0.077 | 0.087 | **0.455** | 0.106 | 0.084 |
| | R-F | 0.219 | 0.192 | 0.176 | 0.098 | 0.084 | 0.083 | 0.237 | 0.185 | 0.207 | 0.179 | 0.185 | 0.066 | 0.102 | **0.266** | 0.146 | 0.111 |
| Contextual4.9 | R-R | **0.733** | 0.341 | 0.697 | 0.232 | 0.347 | 0.361 | 0.556 | 0.709 | 0.656 | 0.659 | 0.603 | 0.070 | 0.559 | 0.396 | 0.181 | 0.053 |
| | R-P | 0.586 | 0.071 | 0.559 | 0.041 | 0.057 | 0.047 | 0.139 | 0.267 | 0.440 | 0.125 | **0.670** | 0.041 | 0.240 | 0.577 | 0.061 | 0.025 |
| | R-F | **0.651** | 0.118 | 0.620 | 0.069 | 0.097 | 0.083 | 0.222 | 0.388 | 0.526 | 0.210 | 0.635 | 0.052 | 0.336 | 0.470 | 0.092 | 0.034 |
| Contextual7.2 | R-R | 0.692 | 0.335 | **0.725** | 0.453 | 0.291 | 0.345 | 0.597 | 0.697 | 0.717 | 0.654 | 0.580 | 0.096 | 0.553 | 0.360 | 0.192 | 0.061 |
| | R-P | 0.744 | 0.100 | 0.546 | 0.070 | 0.080 | 0.060 | 0.227 | 0.353 | 0.365 | 0.164 | **0.775** | 0.084 | 0.304 | 0.633 | 0.087 | 0.053 |
| | R-F | **0.717** | 0.154 | 0.623 | 0.122 | 0.125 | 0.103 | 0.328 | 0.468 | 0.483 | 0.263 | 0.664 | 0.090 | 0.392 | 0.459 | 0.120 | 0.057 |
| Global4.8 | R-R | 0.919 | 0.726 | 0.874 | 0.206 | 0.859 | 0.327 | 0.855 | 0.887 | 0.870 | 0.823 | 0.887 | 0.078 | **0.996** | 0.759 | 0.733 | 0.178 |
| | R-P | 0.939 | 0.292 | **1.000** | 0.073 | 0.041 | 0.043 | 0.900 | 1.000 | 0.978 | 0.362 | 0.747 | 0.046 | 0.338 | 0.766 | 0.113 | 0.116 |
| | R-F | 0.929 | 0.416 | 0.933 | 0.108 | 0.079 | 0.076 | 0.877 | **0.940** | 0.921 | 0.503 | 0.811 | 0.058 | 0.505 | 0.762 | 0.196 | 0.140 |
| Global7.2 | R-R | 0.922 | 0.716 | 0.881 | 0.742 | 0.840 | 0.320 | 0.832 | 0.916 | 0.869 | 0.801 | 0.849 | 0.068 | **0.982** | 0.761 | 0.712 | 0.083 |
| | R-P | 0.832 | 0.297 | 0.773 | 0.078 | 0.052 | 0.072 | **0.984** | 0.874 | 0.917 | 0.460 | 0.871 | 0.060 | 0.457 | 0.834 | 0.125 | 0.070 |
| | R-F | 0.874 | 0.420 | 0.824 | 0.141 | 0.097 | 0.117 | **0.902** | 0.894 | 0.892 | 0.584 | 0.860 | 0.064 | 0.624 | 0.796 | 0.213 | 0.076 |
| Seasonal4.8 | R-R | 0.475 | 0.244 | 0.464 | 0.247 | 0.238 | 0.170 | 0.453 | 0.441 | 0.433 | 0.364 | 0.280 | 0.000 | **0.478** | 0.280 | 0.238 | 0.187 |
| | R-P | 0.999 | 0.062 | **1.000** | 0.597 | 0.864 | 0.500 | 1.000 | 1.000 | 1.000 | 0.966 | 0.630 | 0.000 | 0.113 | 0.610 | 0.065 | 0.041 |
| | R-F | **0.644** | 0.099 | 0.634 | 0.350 | 0.373 | 0.254 | 0.624 | 0.612 | 0.604 | 0.529 | 0.387 | 0.000 | 0.182 | 0.383 | 0.102 | 0.067 |
| Seasonal7.7 | R-R | **0.480** | 0.254 | 0.425 | 0.089 | 0.173 | 0.191 | 0.423 | 0.409 | 0.397 | 0.355 | 0.278 | 0.232 | 0.453 | 0.278 | 0.187 | 0.223 |
| | R-P | 0.944 | 0.071 | **1.000** | 0.112 | 0.962 | 0.588 | 0.993 | 0.917 | 1.000 | 0.926 | 0.692 | 0.323 | 0.178 | 0.766 | 0.063 | 0.149 |
| | R-F | **0.636** | 0.111 | 0.596 | 0.099 | 0.293 | 0.289 | 0.594 | 0.566 | 0.569 | 0.513 | 0.396 | 0.270 | 0.256 | 0.408 | 0.094 | 0.179 |
| Shapelet4.9 | R-R | 0.447 | 0.297 | 0.431 | 0.057 | 0.154 | 0.174 | 0.398 | 0.445 | 0.445 | 0.249 | 0.279 | 0.012 | **0.496** | 0.244 | 0.224 | 0.116 |
| | R-P | 0.944 | 0.065 | 0.679 | 0.104 | 0.433 | 0.382 | **0.964** | 0.348 | 0.778 | 0.213 | 0.479 | 0.043 | 0.083 | 0.542 | 0.057 | 0.036 |
| | R-F | **0.607** | 0.107 | 0.527 | 0.074 | 0.227 | 0.240 | 0.564 | 0.391 | 0.566 | 0.230 | 0.353 | 0.018 | 0.143 | 0.336 | 0.091 | 0.055 |
| Shapelet7.4 | R-R | 0.471 | 0.304 | 0.473 | 0.197 | 0.138 | 0.343 | 0.442 | 0.457 | 0.466 | 0.365 | 0.277 | 0.221 | **0.517** | 0.253 | 0.315 | 0.096 |
| | R-P | **0.980** | 0.083 | 0.620 | 0.123 | 0.309 | 0.153 | 0.818 | 0.479 | 0.598 | 0.211 | 0.583 | 0.235 | 0.118 | 0.489 | 0.114 | 0.042 |
| | R-F | **0.636** | 0.130 | 0.536 | 0.151 | 0.191 | 0.212 | 0.574 | 0.468 | 0.524 | 0.268 | 0.375 | 0.228 | 0.193 | 0.333 | 0.168 | 0.058 |
| Mixture5.7 | R-R | 0.601 | 0.341 | 0.508 | 0.347 | 0.076 | 0.119 | 0.515 | 0.487 | 0.705 | 0.364 | 0.279 | 0.159 | **0.954** | 0.279 | 0.954 | 0.718 |
| | R-P | 0.455 | 0.091 | 0.355 | 0.062 | 0.100 | 0.333 | 0.509 | 0.344 | 0.066 | 0.233 | 0.066 | 0.113 | 0.053 | **0.627** | 0.091 | 0.096 |
| | R-F | **0.518** | 0.143 | 0.418 | 0.106 | 0.086 | 0.175 | 0.512 | 0.403 | 0.121 | 0.284 | 0.107 | 0.132 | 0.100 | 0.386 | 0.165 | 0.170 |
| Mixture8.9 | R-R | 0.670 | 0.293 | 0.595 | 0.343 | 0.057 | 0.145 | 0.599 | 0.558 | 0.594 | 0.465 | 0.279 | 0.178 | 0.998 | 0.267 | **1.000** | 0.806 |
| | R-P | 0.516 | 0.148 | 0.401 | 0.089 | 0.070 | 0.443 | 0.511 | 0.342 | 0.108 | 0.217 | 0.092 | 0.089 | 0.000 | **0.704** | 0.000 | 0.116 |
| | R-F | **0.583** | 0.197 | 0.479 | 0.141 | 0.063 | 0.218 | 0.551 | 0.424 | 0.183 | 0.296 | 0.139 | 0.118 | 0.000 | 0.387 | 0.000 | 0.203 |
| Trend4.8 | R-R | 0.910 | 0.492 | 0.562 | 0.277 | 0.023 | 0.082 | 0.474 | 0.483 | 0.625 | 0.538 | 0.280 | 0.000 | 0.997 | 0.268 | **1.000** | 0.799 |
| | R-P | 0.247 | 0.157 | 0.205 | 0.073 | 0.037 | 0.167 | 0.273 | 0.169 | 0.064 | 0.117 | 0.048 | 0.000 | 0.000 | **0.600** | 0.002 | 0.050 |
| | R-F | **0.388** | 0.238 | 0.300 | 0.115 | 0.029 | 0.110 | 0.347 | 0.251 | 0.116 | 0.192 | 0.081 | 0.000 | 0.001 | 0.371 | 0.004 | 0.094 |
| Trend7.8 | R-R | 0.529 | 0.680 | 0.561 | 0.316 | 0.095 | 0.015 | 0.681 | 0.546 | 0.715 | 0.432 | 0.279 | 0.183 | 0.996 | 0.220 | **1.000** | 0.768 |
| | R-P | 0.242 | 0.122 | 0.168 | 0.099 | 0.146 | 0.048 | 0.187 | 0.165 | 0.082 | 0.155 | 0.076 | 0.127 | 0.001 | **0.641** | 0.002 | 0.064 |
| | R-F | **0.332** | 0.207 | 0.258 | 0.150 | 0.115 | 0.023 | 0.294 | 0.253 | 0.147 | 0.229 | 0.119 | 0.150 | 0.002 | 0.327 | 0.004 | 0.117 |
| 1st **Count** | | **25** | 0 | 5 | 1 | 0 | 0 | 5 | 3 | 2 | 2 | 3 | 0 | 11 | 9 | 5 | 1 |

Table 10: Average Aff-P (Affiliated-Precision), Aff-R (Affiliated-Recall) and Aff-F (Affiliated-F1-score) accuracy measures for all datasets. The best results are highlighted in bold, and the second-best results are underlined.

| Dataset | Metric | CATCH | Modern | iTrans | DualTF | ATrans | DC | TsNet | Patch | DLin | NLin | AE | TFAD | Ocsvm | IF | PCA | HBOS |
|---|---|---|---|---|---|---|---|---|---|---|---|---|---|---|---|---|---|
| CICIDS | Aff-P | **0.667** | 0.563 | 0.586 | 0.553 | 0.531 | 0.533 | 0.566 | 0.569 | 0.575 | 0.579 | 0.543 | 0.550 | 0.530 | 0.548 | 0.541 | 0.538 |
| | Aff-R | 0.959 | 0.781 | 0.892 | 0.926 | 0.593 | 0.882 | 0.783 | 0.785 | 0.799 | 0.794 | 0.156 | 0.611 | **1.000** | 0.672 | 0.724 | 0.545 |
| | Aff-F | **0.787** | 0.654 | 0.708 | 0.692 | 0.560 | 0.664 | 0.657 | 0.660 | 0.669 | 0.669 | 0.243 | 0.579 | 0.693 | 0.604 | 0.619 | 0.542 |
| CalIt2 | Aff-P | **0.742** | 0.650 | 0.703 | 0.617 | 0.645 | 0.571 | 0.691 | 0.667 | 0.668 | 0.616 | 0.560 | 0.619 | 0.652 | 0.539 | 0.688 | 0.620 |
| | Aff-R | 0.955 | 0.975 | 0.963 | 0.959 | 0.838 | 0.894 | 0.932 | 0.976 | 0.976 | **0.984** | 0.617 | 0.933 | 0.982 | 0.321 | 0.869 | 0.969 |
| | Aff-F | **0.835** | 0.780 | 0.812 | 0.751 | 0.729 | 0.697 | 0.794 | 0.793 | 0.793 | 0.757 | 0.587 | 0.744 | 0.783 | 0.402 | 0.768 | 0.756 |
| Credit | Aff-P | 0.618 | 0.611 | 0.559 | 0.513 | 0.521 | 0.488 | 0.610 | 0.612 | 0.605 | 0.608 | 0.560 | 0.585 | 0.556 | **0.658** | 0.564 | 0.533 |
| | Aff-R | 0.956 | 0.952 | 0.984 | 0.935 | 0.865 | 0.898 | 0.953 | 0.955 | 0.948 | 0.952 | 0.562 | 0.757 | 0.997 | 0.612 | 0.959 | **0.999** |
| | Aff-F | **0.750** | 0.744 | 0.713 | 0.663 | 0.650 | 0.632 | 0.744 | 0.746 | 0.738 | 0.742 | 0.561 | 0.660 | 0.714 | 0.634 | 0.710 | 0.695 |
| GECCO | Aff-P | 0.832 | 0.808 | 0.735 | 0.633 | 0.690 | 0.567 | 0.810 | 0.831 | 0.808 | 0.793 | **0.836** | 0.526 | 0.499 | 0.647 | 0.646 | 0.620 |
| | Aff-R | 0.998 | 0.998 | 0.979 | 0.786 | 0.903 | 0.872 | 0.997 | 0.995 | 0.997 | 0.992 | 0.810 | 0.776 | **1.000** | 0.315 | 1.000 | 0.827 |
| | Aff-F | **0.908** | 0.893 | 0.839 | 0.701 | 0.782 | 0.687 | 0.894 | 0.906 | 0.893 | 0.882 | 0.823 | 0.627 | 0.666 | 0.424 | 0.785 | 0.708 |
| Genesis | Aff-P | **0.835** | 0.728 | 0.822 | 0.683 | 0.749 | 0.659 | 0.780 | 0.763 | 0.772 | 0.724 | 0.759 | 0.437 | 0.512 | 0.673 | 0.691 | 0.564 |
| | Aff-R | 0.966 | 0.974 | 0.972 | 0.996 | **1.000** | 0.943 | 0.968 | 0.974 | 0.959 | 0.971 | 0.976 | 0.687 | 1.000 | 0.951 | 0.991 | 1.000 |
| | Aff-F | **0.896** | 0.833 | 0.891 | 0.810 | 0.856 | 0.776 | 0.864 | 0.856 | 0.856 | 0.829 | 0.854 | 0.535 | 0.677 | 0.788 | 0.814 | 0.721 |
| MSL | Aff-P | **0.599** | 0.578 | 0.566 | 0.562 | 0.549 | 0.576 | 0.589 | 0.584 | 0.577 | 0.584 | 0.521 | 0.516 | 0.497 | 0.502 | 0.538 | 0.520 |
| | Aff-R | 0.966 | 0.975 | 0.951 | 0.618 | 0.933 | 0.874 | 0.973 | 0.952 | 0.975 | 0.948 | 0.781 | 0.936 | 0.902 | 0.697 | 0.914 | **0.982** |
| | Aff-F | **0.740** | 0.726 | 0.710 | 0.588 | 0.692 | 0.694 | 0.734 | 0.724 | 0.725 | 0.722 | 0.625 | 0.665 | 0.641 | 0.584 | 0.678 | 0.680 |
| NYC | Aff-P | **1.000** | 0.639 | 0.520 | 0.551 | 0.751 | 0.769 | 0.750 | 0.719 | 0.814 | 0.817 | 0.529 | 0.689 | 0.500 | 0.481 | 0.516 | 0.525 |
| | Aff-R | 0.989 | 0.965 | 0.999 | 0.988 | 0.988 | 0.980 | 0.843 | 0.843 | 0.843 | 0.821 | 0.989 | 0.689 | **1.000** | 0.989 | 0.997 | 0.946 |
| | Aff-F | **0.994** | 0.769 | 0.684 | 0.708 | 0.853 | 0.862 | 0.794 | 0.776 | 0.819 | 0.689 | 0.689 | 0.689 | 0.667 | 0.648 | 0.680 | 0.675 |
| PSM | Aff-P | 0.808 | 0.734 | 0.765 | 0.622 | 0.600 | 0.538 | 0.762 | 0.739 | 0.777 | 0.762 | 0.776 | 0.543 | 0.652 | **0.904** | 0.712 | 0.621 |
| | Aff-R | 0.918 | 0.941 | **0.966** | 0.868 | 0.871 | 0.932 | 0.939 | 0.948 | 0.893 | 0.942 | 0.649 | 0.744 | 0.447 | 0.472 | 0.692 | 0.700 |
| | Aff-F | **0.859** | 0.825 | 0.854 | 0.725 | 0.710 | 0.682 | 0.842 | 0.831 | 0.831 | 0.843 | 0.707 | 0.628 | 0.531 | 0.620 | 0.702 | 0.658 |
| SMD | Aff-P | 0.773 | 0.755 | 0.736 | 0.527 | 0.607 | 0.510 | 0.745 | 0.748 | 0.761 | 0.762 | **0.889** | 0.517 | 0.649 | 0.801 | 0.680 | 0.557 |
| | Aff-R | 0.938 | 0.948 | 0.943 | 0.956 | 0.895 | **0.998** | 0.938 | 0.970 | 0.940 | 0.946 | 0.291 | 0.913 | 0.866 | 0.513 | 0.807 | 0.722 |
| | Aff-F | **0.847** | 0.840 | 0.827 | 0.679 | 0.724 | 0.675 | 0.831 | 0.845 | 0.841 | 0.844 | 0.439 | 0.660 | 0.742 | 0.626 | 0.738 | 0.629 |
| SWAT | Aff-P | 0.623 | 0.593 | 0.568 | 0.535 | 0.539 | 0.539 | **0.698** | 0.582 | 0.586 | 0.565 | 0.604 | 0.613 | 0.542 | 0.574 | 0.571 | 0.527 |
| | Aff-R | 0.956 | 0.942 | 0.976 | 0.993 | 0.981 | 0.982 | 0.918 | 0.932 | 0.988 | 0.972 | 0.949 | 0.777 | **0.999** | 0.835 | 0.932 | 0.847 |
| | Aff-F | 0.755 | 0.728 | 0.718 | 0.695 | 0.696 | 0.696 | **0.793** | 0.716 | 0.736 | 0.715 | 0.738 | 0.686 | 0.703 | 0.681 | 0.708 | 0.650 |
| SMAP | Aff-P | **0.574** | 0.472 | 0.439 | 0.509 | 0.545 | 0.548 | 0.475 | 0.450 | 0.455 | 0.447 | 0.416 | 0.515 | 0.392 | 0.425 | 0.399 | 0.399 |
| | Aff-R | 0.895 | 0.971 | 0.885 | **0.997** | 0.991 | 0.973 | 0.972 | 0.931 | 0.953 | 0.914 | 0.524 | 0.979 | 0.703 | 0.645 | 0.687 | 0.703 |
| | Aff-F | 0.699 | 0.635 | 0.587 | 0.674 | **0.703** | 0.701 | 0.638 | 0.606 | 0.616 | 0.601 | 0.463 | 0.675 | 0.503 | 0.512 | 0.505 | 0.509 |
| ASD | Aff-P | 0.693 | 0.669 | 0.660 | 0.551 | 0.553 | 0.569 | 0.693 | 0.662 | 0.690 | 0.650 | 0.600 | 0.532 | 0.505 | **0.761** | 0.532 | 0.538 |
| | Aff-R | 0.966 | 0.953 | **0.969** | 0.760 | 0.870 | 0.921 | 0.963 | 0.954 | 0.918 | 0.944 | 0.942 | 0.785 | 0.810 | 0.842 | 0.871 | 0.891 |
| | Aff-F | **0.804** | 0.782 | 0.780 | 0.604 | 0.674 | 0.702 | 0.800 | 0.777 | 0.782 | 0.766 | 0.731 | 0.630 | 0.617 | 0.781 | 0.656 | 0.669 |
| Contextual4.9 | Aff-P | 0.838 | 0.505 | 0.821 | 0.503 | 0.517 | 0.490 | 0.552 | 0.669 | 0.773 | 0.563 | **0.861** | 0.526 | 0.619 | 0.861 | 0.488 | 0.487 |
| | Aff-R | 0.812 | 0.814 | 0.761 | 0.776 | 0.792 | 0.779 | 0.837 | 0.883 | 0.806 | **0.908** | 0.674 | 0.666 | 0.769 | 0.582 | 0.463 | 0.547 |
| | Aff-F | **0.825** | 0.623 | 0.790 | 0.611 | 0.625 | 0.602 | 0.666 | 0.761 | 0.789 | 0.695 | 0.756 | 0.588 | 0.686 | 0.694 | 0.475 | 0.515 |
| Contextual7.2 | Aff-P | **0.907** | 0.508 | 0.826 | 0.519 | 0.498 | 0.496 | 0.566 | 0.705 | 0.709 | 0.585 | 0.901 | 0.523 | 0.660 | 0.877 | 0.505 | 0.476 |
| | Aff-R | 0.748 | 0.782 | 0.803 | **0.905** | 0.685 | 0.735 | 0.806 | 0.851 | 0.841 | 0.887 | 0.647 | 0.580 | 0.760 | 0.535 | 0.447 | 0.422 |
| | Aff-F | **0.820** | 0.616 | 0.814 | 0.660 | 0.577 | 0.592 | 0.665 | 0.771 | 0.770 | 0.705 | 0.753 | 0.550 | 0.707 | 0.665 | 0.474 | 0.447 |
| Global4.8 | Aff-P | 0.985 | 0.661 | **1.000** | 0.549 | 0.506 | 0.479 | 0.964 | 1.000 | 0.994 | 0.721 | 0.916 | 0.520 | 0.713 | 0.938 | 0.599 | 0.535 |
| | Aff-R | 0.922 | 0.867 | 0.876 | 0.749 | 0.933 | 0.737 | 0.865 | 0.889 | 0.875 | 0.907 | 0.912 | 0.679 | **0.996** | 0.900 | 0.894 | 0.554 |
| | Aff-F | **0.953** | 0.750 | 0.934 | 0.633 | 0.656 | 0.581 | 0.912 | 0.941 | 0.930 | 0.804 | 0.914 | 0.589 | 0.831 | 0.919 | 0.717 | 0.544 |
| Global7.2 | Aff-P | 0.954 | 0.662 | 0.926 | 0.515 | 0.506 | 0.481 | **0.993** | 0.949 | 0.975 | 0.769 | 0.958 | 0.513 | 0.771 | 0.950 | 0.572 | 0.514 |
| | Aff-R | 0.935 | 0.857 | 0.893 | 0.939 | 0.937 | 0.648 | 0.836 | 0.928 | 0.881 | 0.861 | 0.895 | 0.577 | **0.993** | 0.866 | 0.872 | 0.510 |
| | Aff-F | **0.945** | 0.746 | 0.909 | 0.665 | 0.657 | 0.552 | 0.907 | 0.938 | 0.925 | 0.812 | 0.925 | 0.543 | 0.868 | 0.906 | 0.691 | 0.512 |
| Seasonal4.8 | Aff-P | **1.000** | 0.533 | 1.000 | 0.950 | 0.945 | 0.912 | 1.000 | 1.000 | 1.000 | 0.993 | 0.870 | 0.573 | 0.654 | 0.859 | 0.534 | 0.509 |
| | Aff-R | **0.998** | 0.955 | 0.986 | 0.761 | 0.784 | 0.847 | 0.987 | 1.000 | 0.988 | 0.931 | 0.994 | 0.485 | 0.998 | 0.993 | 0.893 | 0.944 |
| | Aff-F | **0.999** | 0.684 | 0.993 | 0.845 | 0.857 | 0.879 | 0.993 | 0.994 | 0.994 | 0.961 | 0.928 | 0.525 | 0.790 | 0.921 | 0.668 | 0.661 |
| Seasonal7.7 | Aff-P | 0.992 | 0.534 | **1.000** | 0.628 | 0.970 | 0.882 | 0.998 | 0.985 | 1.000 | 0.982 | 0.868 | 0.788 | 0.697 | 0.925 | 0.482 | 0.546 |
| | Aff-R | **0.998** | 0.934 | 0.983 | 0.809 | 0.570 | 0.800 | 0.982 | 0.983 | 0.986 | 0.901 | 0.991 | 0.916 | 0.994 | 0.988 | 0.817 | 0.919 |
| | Aff-F | **0.995** | 0.679 | 0.992 | 0.707 | 0.718 | 0.839 | 0.990 | 0.984 | 0.993 | 0.940 | 0.925 | 0.847 | 0.820 | 0.956 | 0.606 | 0.685 |
| Shapelet4.9 | Aff-P | **0.997** | 0.517 | 0.952 | 0.653 | 0.779 | 0.763 | 0.968 | 0.874 | 0.979 | 0.646 | 0.732 | 0.603 | 0.606 | 0.832 | 0.547 | 0.492 |
| | Aff-R | 0.977 | 0.976 | 0.979 | 0.732 | 0.704 | 0.743 | 0.871 | 0.983 | 0.979 | 0.877 | **0.992** | 0.512 | 0.988 | 0.965 | 0.898 | 0.940 |
| | Aff-F | **0.987** | 0.676 | 0.966 | 0.690 | 0.740 | 0.753 | 0.917 | 0.925 | 0.979 | 0.744 | 0.842 | 0.554 | 0.751 | 0.894 | 0.680 | 0.646 |
| Shapelet7.4 | Aff-P | **0.999** | 0.514 | 0.941 | 0.591 | 0.658 | 0.641 | 0.965 | 0.913 | 0.914 | 0.656 | 0.827 | 0.738 | 0.662 | 0.821 | 0.567 | 0.499 |
| | Aff-R | 0.969 | 0.975 | 0.972 | 0.842 | 0.659 | 0.823 | 0.967 | 0.970 | 0.974 | 0.943 | **0.985** | 0.907 | 0.983 | 0.949 | 0.870 | 0.873 |
| | Aff-F | **0.983** | 0.673 | 0.957 | 0.695 | 0.658 | 0.721 | 0.966 | 0.940 | 0.943 | 0.774 | 0.899 | 0.814 | 0.791 | 0.880 | 0.686 | 0.635 |
| Mixture5.7 | Aff-P | 0.863 | 0.628 | 0.867 | 0.555 | 0.541 | 0.802 | 0.875 | 0.867 | 0.581 | 0.779 | 0.560 | 0.620 | 0.539 | **0.879** | 0.543 | 0.501 |
| | Aff-R | 0.991 | 0.940 | 0.920 | 0.910 | 0.721 | 0.737 | 0.920 | 0.922 | 0.959 | 0.895 | 0.992 | 0.884 | **0.998** | 0.970 | 0.997 | 0.969 |
| | Aff-F | **0.923** | 0.753 | 0.893 | 0.689 | 0.618 | 0.768 | 0.897 | 0.894 | 0.723 | 0.833 | 0.716 | 0.729 | 0.700 | 0.922 | 0.703 | 0.660 |
| Mixture8.9 | Aff-P | 0.841 | 0.607 | 0.838 | 0.510 | 0.502 | 0.814 | 0.820 | 0.797 | 0.568 | 0.744 | 0.520 | 0.561 | 0.517 | **0.868** | 0.518 | 0.515 |
| | Aff-R | 0.988 | 0.871 | 0.991 | 0.954 | 0.608 | 0.718 | 0.989 | 0.990 | 0.975 | 0.929 | 0.991 | 0.897 | **1.000** | 0.947 | 1.000 | 0.995 |
| | Aff-F | **0.909** | 0.716 | 0.908 | 0.665 | 0.550 | 0.763 | 0.897 | 0.883 | 0.718 | 0.827 | 0.682 | 0.690 | 0.682 | 0.906 | 0.683 | 0.678 |
| Trend4.8 | Aff-P | 0.861 | 0.812 | 0.831 | 0.505 | 0.560 | 0.751 | 0.861 | 0.840 | 0.571 | 0.765 | 0.502 | 0.571 | 0.502 | **0.877** | 0.505 | 0.503 |
| | Aff-R | 0.997 | 0.974 | 0.984 | 0.937 | 0.746 | 0.803 | 0.929 | 0.978 | 0.991 | 0.994 | 0.995 | 0.553 | **1.000** | 0.942 | 1.000 | 0.996 |
| | Aff-F | **0.924** | 0.886 | 0.901 | 0.656 | 0.640 | 0.776 | 0.893 | 0.904 | 0.724 | 0.865 | 0.667 | 0.562 | 0.668 | 0.909 | 0.671 | 0.668 |
| Trend7.8 | Aff-P | 0.809 | 0.716 | 0.707 | 0.522 | 0.609 | 0.656 | 0.722 | 0.770 | 0.582 | 0.712 | 0.516 | 0.628 | 0.519 | **0.865** | 0.517 | 0.506 |
| | Aff-R | 0.916 | 0.978 | 0.985 | 0.848 | 0.679 | 0.627 | 0.981 | 0.961 | 0.975 | 0.946 | 0.991 | 0.848 | **1.000** | 0.842 | 1.000 | 0.971 |
| | Aff-F | **0.859** | 0.827 | 0.823 | 0.647 | 0.642 | 0.641 | 0.832 | 0.855 | 0.729 | 0.812 | 0.679 | 0.721 | 0.684 | 0.853 | 0.681 | 0.665 |
| 1st Count | | **34** | 0 | 4 | 2 | 2 | 1 | 3 | 0 | 0 | 2 | 5 | 0 | 10 | 7 | 0 | 2 |

Table 11: Average A-R (AUC-ROC) and Aff-F (Affiliated-F1) accuracy measures for 4 synthetic datasets of different types of anomalies. The best results are highlighted in bold, and the second-best results are underlined.

| Model | CATCH | | Modern | | iTrans | | DualTF | | TFAD | |
|---|---|---|---|---|---|---|---|---|---|---|
| Metric | Aff-F | A-R | Aff-F | A-R | Aff-F | A-R | Aff-F | A-R | Aff-F | A-R |
| Low-frequency anomalies | **0.998** | **0.996** | 0.811 | 0.873 | 0.850 | 0.929 | 0.744 | 0.662 | 0.700 | 0.538 |
| Mid-frequency anomalies | **0.927** | **0.987** | 0.814 | 0.859 | 0.913 | 0.984 | 0.723 | 0.575 | 0.638 | 0.519 |
| High-frequency anomalies | **0.915** | **0.987** | 0.738 | 0.809 | 0.864 | 0.927 | 0.462 | 0.602 | 0.668 | 0.513 |
| Not distinctly frequency band | **0.873** | **0.917** | 0.775 | 0.765 | 0.852 | 0.910 | 0.720 | 0.630 | 0.619 | 0.495 |

# D  EMPIRICAL VERIFICATION SHOWS THAT CATCH CAN PERFORM FINE-GRAINED MODELING IN EACH FREQUENCY BAND

We design a set of experiments to validate the advantages of our method in capturing fine-grained frequency characteristics. We use the synthetic dataset creation method provided by TODS (Lai et al., 2021) to generate four synthetic datasets—see Figure 7, simulating scenarios with known frequency-specific anomalies: low-frequency anomalies, medium-frequency anomalies, high-frequency anomalies, and anomalies that are not distinctly separated across different frequency bands. We then test CATCH and several state-of-the-art methods on these datasets.

The experimental results in Table 11 show that CATCH consistently outperforms other algorithms across all four datasets. Notably, most algorithms perform poorly when dealing with time series containing high-frequency anomalies, while CATCH demonstrates outstanding performance. Moreover, even on time series where anomalies are uniformly distributed across each frequency band, CATCH still achieves outstanding results. These findings provide strong evidence that CATCH excels in capturing fine-grained frequency characteristics, significantly enhancing the performance of time series anomaly detection.

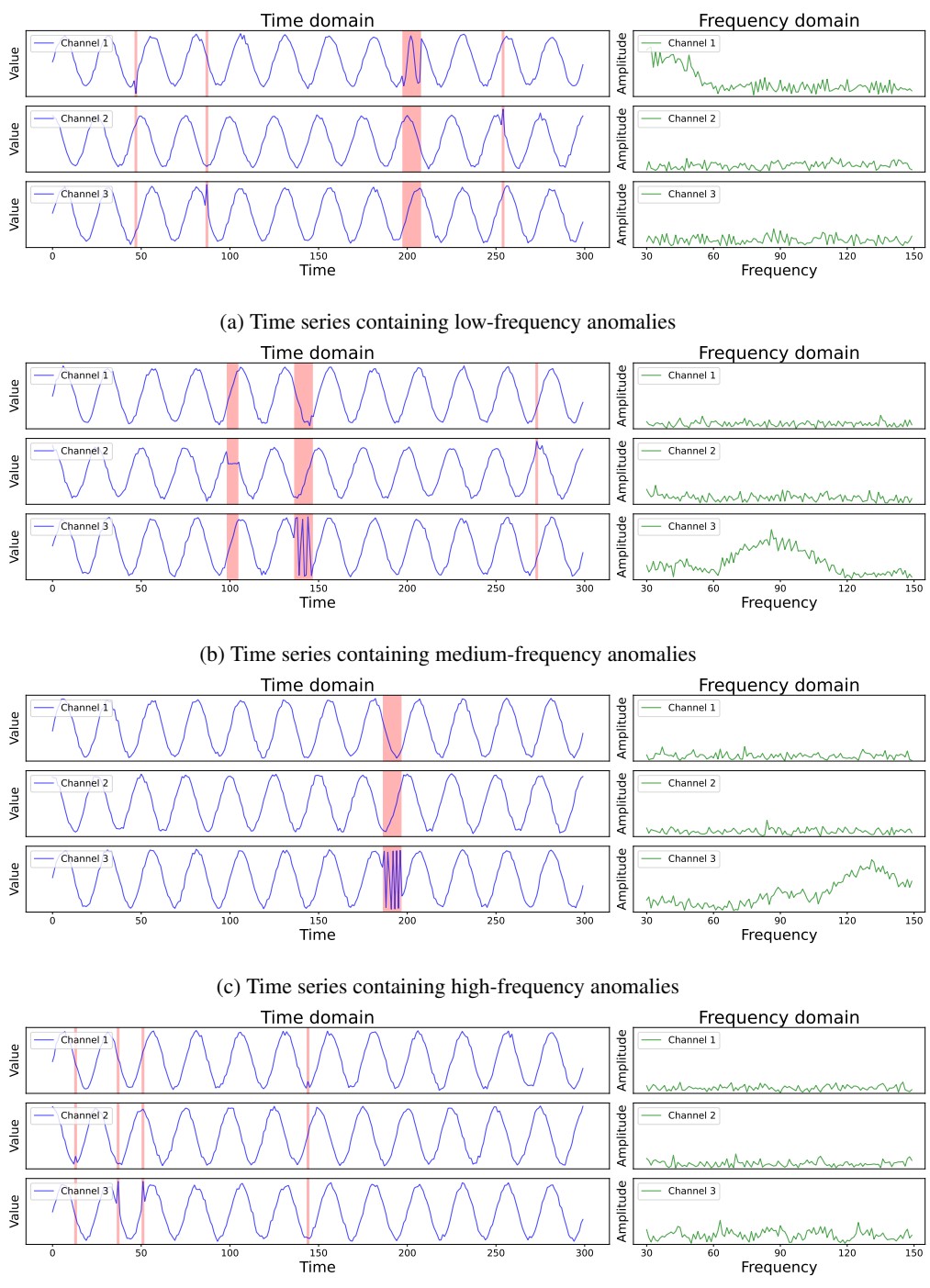

(a) Time series containing low-frequency anomalies

(b) Time series containing medium-frequency anomalies

(c) Time series containing high-frequency anomalies

(d) Anomalies that are not distinctly separated across different frequency bands

Figure 7: Four synthetic time series simulating scenarios with known frequency-specific anomalies.

