# OpenReview forum: "CATCH: Channel-Aware Multivariate Time Series Anomaly Detection via Frequency Patching"
_ICLR.cc/2025/Conference — ICLR 2025 Poster_

### Official Review · Reviewer_LbNG · 2024-10-28

**Soundness:** 3
**Presentation:** 2
**Contribution:** 3
**Rating:** 8
**Confidence:** 4

**Summary:**

This paper centers on temporal anomaly detection and introduces an algorithm capable of capturing fine-grained frequency characteristics and channel interdependencies. The proposed method leverages frequency patches to discern subtle frequency details and employs channel fusion techniques to identify correlations among different channels. The experimental results indicate that the algorithm delivers state-of-the-art performance across various datasets.

**Strengths:**

1. The paper introduces innovative frequency patching techniques and a Channel Fusion Module (CFM) in
the field of multivariate time series anomaly detection, which can more finely capture frequency
characteristics and channel correlations, marking a significant improvement over existing methods.

2. The experimental findings demonstrate that the proposed methodology possesses commendable performance.

**Weaknesses:**

1. The methodological descriptions and accompanying figures are not entirely congruent, which poses challenges for readers. For instance, the structure on the left side of Figure 2 appears disorganized, obscuring the input data and its flow for the reader. Furthermore, the freq-score and time-score mentioned in Figure 3 lack corresponding textual or formulaic explanations in the relevant sections, thereby hindering readers' comprehension of the actual computation of the anomaly scores.

2. While the CFM is an innovative feature, its complexity might be challenging for some readers to grasp.
The paper could provide more intuitive explanations or diagrams to help readers better understand how
the CFM operates.

3. The experiments did not incorporate classic datasets such as ASD or SWaT.

**Questions:**

1. As the paper indicates, frequency characteristics exhibit long-tailed distributions, with the majority of information concentrated in the low-frequency band. Thus, what is the rationale behind the focus of this study on capturing details within the high-frequency band? In reality, as depicted in Figure 1(a), there are already significant differences between normal and anomalous data in the low-frequency band.
2. Was point adjustment employed when assessing the algorithm's performance?
3. Why does Table 2, presenting the multi-metric results, only illustrate performance on three datasets? It might be beneficial to include all results in the appendix for comprehensiveness.

---

> ### Author Response · Authors · 2024-11-20
> **Response to Reviewer LbNG (Part I)**
>
> Thank you for your detailed and constructive feedback.
>
> **W1:** The methodological descriptions and accompanying figures are not entirely congruent, which poses challenges for readers.
>
> Thank you for your valuable feedback, which has played a crucial role in improving the quality of the paper and enhancing the reading experience for the audience. Based on your suggestions, we have made the following revisions:
>
> **Modifications to Figure 2 of the revised paper:**
>
> - Optimization of input arrows: We added input arrows for both the Mask Generator and the Channel-Masked Transformer Layer, using yellow arrows to indicate that they share the same input source, which improves the logical coherence.
> - Redrawing of the CFM: The Channel Fusion Module (CFM) has been redrawn to more intuitively showcase its operation, helping readers better understand its function and role.
>
> **Comprehensive revision and organization of Chapter 3 of the revised paper according to Figure 2 to ensure consistency between text and figures:**
>
> 1. We reorganized Section 3.1, "Architecture Overview," to systematically introduce the various modules of CATCH from a macro perspective.
> 2. Based on the architecture diagram in Figure 2, Sections 3.2 to 3.4 were newly structured and rearranged:
>     - Section 3.2 elaborates on the design and relationships of the Mask Generator, Channel-Masked Transformer Layer, and Channel Correlation Discovery.
>     - Section 3.3 focuses on the Time-Frequency Reconstruction Module.
>     - Section 3.4 explains the implementation logic of the joint bi-level optimization, linking Sections 3.2 and 3.3, and clearly presenting the final loss function.
> 3. Section 3.5 further explains the calculation method for anomaly scores.
>
> These changes aim to optimize the logical flow of the sections, making the content more concise, while enhancing the reader's overall understanding of the CATCH method and the reading experience.
>
> **Optimization of Figure 3 description of the revised paper:** We enhanced the textual description corresponding to Figure 3 $\underline{\text{in Section 3.5 lines 360-376 of the revised paper}}$ and added formulas to ensure the content is clearer and easier to understand.
>
> **W2:** While the CFM is an innovative feature, its complexity might be challenging for some readers to grasp.
>
> - Thank you for your valuable feedback, which has played an important role in improving the quality of our paper. Based on this specific suggestion, we have made two modifications:
> 1. We have redrawn the **Channel Fusion Module** $\underline{\text{in Figure 2 of the revised paper}}$ to help readers better understand how the CFM operates.
> 2. We have reorganized Section 3.2, "CHANNEL FUSION MODULE." Specifically, we have detailed the design and relationships of the Mask Generator, Channel-Masked Transformer Layer, and Channel Correlation Discovering, as shown in the Channel Fusion Module in Figure 2.

---

> > ### Author Response · Authors · 2024-11-20
> > **Response to Reviewer LbNG (Part II)**
> >
> > **W3:** Lack of evaluation of some datasets.
> >
> > We have added evaluations for the ASD and SWaT datasets, and the results show that CATCH still performs excellently on these datasets, maintaining its superior performance.
> >
> > | Dataset  | Method       | Acc          | P            | R            | F1           | R-P          | R-R          | R-F          | Aff-P        | Aff-R        | Aff-F        | A-R          | A-P          | R-A-R        | R-A-P        | V-ROC        | V-PR         |
> > | -------- | ------------ | ------------ | ------------ | ------------ | ------------ | ------------ | ------------ | ------------ | ------------ | ------------ | ------------ | ------------ | ------------ | ------------ | ------------ | ------------ | ------------ |
> > | **SWAT** | DualTF       | 0.635        | $\underline{\text{0.143}}$ | **0.404**    | **0.212**    | 0.122        | **0.220**    | **0.157**    | 0.535        | **0.993**    | 0.695        | **0.567**    | $\underline{\text{0.143}}$ | $\underline{\text{0.611}}$ | $\underline{\text{0.172}}$ | $\underline{\text{0.608}}$ | $\underline{\text{0.171}}$ |
> > |          | iTransformer | 0.794        | 0.088        | $\underline{\text{0.074}}$ | 0.080        | 0.066        | $\underline{\text{0.203}}$ | 0.099        | 0.568        | $\underline{\text{0.976}}$ | 0.718        | 0.242        | 0.084        | 0.353        | 0.122        | 0.344        | 0.118        |
> > |          | Modern       | 0.836        | 0.119        | 0.055        | 0.075        | 0.072        | 0.200        | 0.106        | 0.593        | 0.942        | $\underline{\text{0.728}}$ | 0.244        | 0.093        | 0.357        | 0.138        | 0.348        | 0.132        |
> > |          | TFAD         | **0.878**    | 0.135        | 0.001        | 0.001        | **0.135**    | 0.115        | 0.124        | $\underline{\text{0.613}}$ | 0.777        | 0.686        | 0.500        | 0.123        | 0.509        | 0.165        | 0.508        | 0.164        |
> > |          | CATCH (ours) | $\underline{\text{0.857}}$ | **0.195**    | 0.056        | $\underline{\text{0.087}}$ | $\underline{\text{0.124}}$ | 0.184        | $\underline{\text{0.148}}$ | **0.623**    | 0.956        | **0.755**    | $\underline{\text{0.545}}$ | **0.166**    | **0.673**    | **0.251**    | **0.662**    | **0.241**    |
> > | **ASD**  | DualTF       | 0.851        | 0.114        | 0.185        | 0.109        | 0.165        | 0.146        | 0.099        | 0.551        | 0.760        | 0.605        | 0.579        | 0.095        | 0.624        | 0.103        | 0.611        | 0.103        |
> > |          | iTransformer | 0.912        | 0.220        | 0.198        | 0.152        | $\underline{\text{0.172}}$ | 0.266        | $\underline{\text{0.180}}$ | $\underline{\text{0.669}}$ | 0.954        | $\underline{\text{0.782}}$ | 0.692        | 0.158        | 0.746        | 0.170        | 0.739        | 0.170        |
> > |          | ModernTCN    | 0.898        | $\underline{\text{0.230}}$ | **0.253**    | $\underline{\text{0.184}}$ | 0.165        | $\underline{\text{0.291}}$ | 0.176        | 0.660        | **0.969**    | 0.780        | $\underline{\text{0.759}}$ | $\underline{\text{0.164}}$ | $\underline{\text{0.812}}$ | $\underline{\text{0.197}}$ | $\underline{\text{0.804}}$ | $\underline{\text{0.194}}$ |
> > |          | TFAD         | **0.950**    | 0.077        | 0.010        | 0.017        | 0.077        | 0.062        | 0.066        | 0.532        | 0.785        | 0.630        | 0.502        | 0.052        | 0.506        | 0.089        | 0.506        | 0.091        |
> > |          | CATCH (ours) | $\underline{\text{0.933}}$ | **0.309**    | $\underline{\text{0.225}}$ | **0.219**    | **0.212**    | **0.303**    | **0.219**    | **0.693**    | $\underline{\text{0.966}}$ | **0.804**    | **0.824**    | **0.231**    | **0.861**    | **0.269**    | **0.853**    | **0.261**    |

---

> ### Author Response · Authors · 2024-11-20
> **Response to Reviewer LbNG (Part III)**
>
> **Q1:** What is the rationale behind the focus of this study on capturing details within the high-frequency band?
>
> Although most of the information is concentrated in the low-frequency band, the mixture anomaly frequency domain map in Figure 1a shows that anomalies still affect high-frequency information. Our goal is to address the limitations of previous methods in modeling high-frequency information. By dividing the frequency bands, we aim to allow low-frequency and high-frequency information to be detected more effectively and avoid negative interference between them. As shown in the ablation study results $\underline{\text{in Table 3 of the original paper}}$, "Training w/o Patching" (i.e., the method without frequency band division) performs significantly worse than the frequency band division strategy employed in CATCH. This result indicates that frequency band division helps in more effectively detecting anomalies.
>
> Additionally, we designed a set of experiments to further demonstrate that our method effectively handles time series with high-frequency anomalies. We used the synthetic dataset creation method provided by TODS[1] to generate four synthetic datasets, simulating scenarios with known frequency-specific anomalies: low-frequency anomalies, medium-frequency anomalies, high-frequency anomalies, and anomalies that are not distinctly separated across different frequency bands. We then tested CATCH alongside several state-of-the-art methods on these datasets.
>
> The experimental results $\underline{\text{in Table 12 in lines 1350-1359 of the revised paper}}$ show that CATCH consistently outperforms other algorithms across all four datasets. Notably, most algorithms struggle to handle time series containing high-frequency anomalies, while CATCH performs exceptionally well. Moreover, even on time series where anomalies are uniformly distributed across each frequency band, CATCH still achieves outstanding results. These findings provide strong evidence that CATCH excels in capturing fine-grained frequency characteristics, effectively detecting high-frequency anomalies and significantly enhancing the performance of time series anomaly detection.
>
> [1] Revisiting time series outlier detection: Definitions and benchmarks. NeurIPS 2021
>
> **Q2:** Was point adjustment employed when assessing the algorithm's performance?
>
> In the all 16 evaluation metrics listed $\underline{\text{in Appendix A.2 of the original paper}}$, we did not use the point adjust operation. Consistent with the views of many studies[1, 2], we believe that the point adjust metric does not effectively reflect the actual performance of the algorithm.
>
> [1] TimeSeAD: Benchmarking Deep Multivariate Time-Series  Anomaly Detection. TMLR 2023
>
> [2] Towards a rigorous evaluation of time-series anomaly detection. AAAI 2022
>
> **Q3:** Why does Table 2, presenting the multi-metric results, only illustrate performance on three datasets?
>
> Due to space limitations in the main text, we only presented the performance of the algorithm on multiple evaluation metrics for three datasets. However, we provide a detailed report on the performance of all algorithms across all datasets and evaluation metrics $\underline{\text{in Table 5-9 in the Appendix C of the original paper}}$.

---

> > ### Comment · Reviewer_LbNG · 2024-11-22
> > **I have read the rebuttal**
> >
> > I have read the rebuttal.
> > I hope the author can revise the paper, as mentioned in the response, include new experimental results, and release the source code.

---

> > > ### Author Response · Authors · 2024-11-22
> > >
> > > Thank you for your suggestions. We deeply appreciate the opportunity to enhance the quality of our paper. We will adopt your suggestions to include the experimental results of SWAT and ASD in the main text and present all metrics results in the appendix. However, this involves additional experimental and textual work. We are working hard to provide a modified version as soon as possible and will notify you once we have revised the paper. Best wishes!

---

> > > ### Author Response · Authors · 2024-11-23
> > > **Response to Reviewer LbNG**
> > >
> > > Thank you for your suggestions. We have addressed your suggestions by completing the experimental and textual modifications, and we have updated both our paper and code repository accordingly.
> > >
> > > - The results of CATCH and 15 baselines on the two new datasets, ASD and SWAT, are now presented $\underline{\text{in Table 2 of the main text of the revised paper}}$.
> > > - All metric results can be found $\underline{\text{in the Appendix C Table 6–11 of the revised paper}}$.
> > > - We have updated all scripts required to reproduce these results in the `scripts`  folder of our anonymous code repository.
> > >
> > > We would like to inquire if our response addresses your primary concerns. If it does, we kindly request that you reconsider the score. If you have any additional suggestions, we are more than willing to engage in further discussions and make necessary improvements to the paper.
> > >
> > > Thanks again for dedicating your time to enhancing our paper!

---

> ### Author Response · Authors · 2024-11-25
> **Waiting for response**
>
> Dear Reviewer LbNG,
>
> Since the End of the Rebuttal comes soon, we would like to inquire if our response addresses your primary concerns. If it does, we kindly request that you reconsider the score. If you have any additional suggestions, we are more than willing to engage in further discussions and make necessary improvements to the paper. Thanks again for dedicating your time to enhancing our paper!
>
> Kind Regards, Authors

---

> > ### Author Response · Authors · 2024-11-26
> > **Kindly Request for Reviewer's Feedback**
> >
> > Dear Reviewer LbnG,
> >
> > Since the End of author/reviewer discussions is coming soon, may we know if our response addresses your main concerns? If so, we kindly ask for your reconsideration of the score. If you have any further concerns, please let us know and we will be more than happy to engage in more discussion and paper improvements.
> >
> > Once again, thank you for your suggestion and time!

---

> > > ### Author Response · Authors · 2024-11-29
> > > **Kindly Request for Reviewer's Feedback**
> > >
> > > Dear Reviewer LbnG,
> > >
> > > We really appreciate your efforts during this tight review timeline and the recognition of the strengths of our paper.
> > >
> > > We are really happy and respect your current rating for our work. We would be grateful if you could confirm that you have read our responses and let us know if there's anything we might have overlooked that could improve the score so that we can have the opportunity to showcase this work at the conference. We are eager to discuss any concerns that may remain open after our response. Thank you again for taking the time for this review.

---

> > > > ### Author Response · Authors · 2024-12-02
> > > > **Kindly Request for Reviewer's Feedback**
> > > >
> > > > Dear Reviewer LbnG,
> > > >
> > > > We really appreciate your efforts during this tight review timeline and the recognition of the strengths of our paper.
> > > >
> > > > We are really happy and respect your current rating for our work. Since we have addressed your suggested additions in the revised paper, we would be grateful if you could confirm that you have read our responses and let us know if there's anything we might have overlooked that could improve the score so that we can have the opportunity to showcase this work at the conference. We are eager to discuss any concerns that may remain open after our response. Thank you again for taking the time for this review.

---

> > > > > ### Comment · Reviewer_LbNG · 2024-12-03
> > > > > **I have Updated my score**
> > > > >
> > > > > I have updated my score.

---

> > > > > > ### Author Response · Authors · 2024-12-03
> > > > > >
> > > > > > We are thrilled that our responses have effectively addressed your questions and comments. We would like to express our sincerest gratitude for taking the time to review our paper and provide us with such detailed feedback.

---

### Official Review · Reviewer_aaEz · 2024-10-31

**Soundness:** 3
**Presentation:** 3
**Contribution:** 3
**Rating:** 6
**Confidence:** 4

**Summary:**

This paper introduces CATCH, a Multivariate Time Series Anomaly Detection framework that enables the simultaneous detection of point and subsequence anomalies through frequency patching. By leveraging Fourier Transformation across time and frequency domains, CATCH enhances subsequence anomaly detection and utilizes the Channel Fusion Module (CFM) to adaptively exploit channel correlations. The CFM integrates a channel correlation discovery mechanism and a bi-level multi-objective optimization process to effectively isolate irrelevant channels and cluster relevant ones.

**Strengths:**

1. The paper introduces CATCH, a novel framework that enhances the detection of both point and subsequence anomalies in multivariate time series through frequency domain patching. This approach creatively combines existing ideas with new methodologies, particularly through its Channel Fusion Module (CFM).
2. The paper is clearly organized and well-written.
3. CATCH addresses some limitations of existing methods. Its integration of frequency and time domain analysis holds promise for impactful applications in various fields.

**Weaknesses:**

1.	The paper presents modules in isolation, lacking a cohesive explanation of their interactions, which can confuse readers regarding the overall system functionality. A clearer narrative linking module interactions would enhance comprehension.
2.	Key operations like instance normalization and frequency domain patching are described without adequate context, leaving readers questioning their necessity.
3.	The innovation behind the Channel Fusion Module (CFM) is not sufficiently highlighted, particularly regarding the necessity of the masking mechanism and its impact on detection performance.
4.	The goals and significance of the Time-Frequency Reconstruction Module (TFRM) are not clearly articulated, potentially leading to misunderstandings about its contributions. Clarifying the objectives of TFRM would enhance the reader's understanding of its role.

**Questions:**

1.	In line 53, the term "heterogeneous subsequence anomalies" is mentioned. Could you provide a specific definition for heterogeneous subsequence anomalies? How do they differ from or relate to regular time series anomalies?
2.	In sections 2.2 and 2.3, in addition to discussing the current state of research, could you include some definitional descriptions? For instance, elaborating on the relationships between channels and frequency domain information in the MTSAD problem could enhance the clarity and structure of the article.
3.	In section 3.1, the description of the framework structure provides technical details, but it lacks a clear explanation of the processes involved. The roles and relationships of the various modules don’t seem to connect well, which affects the logical flow. Adding descriptions of their functions and interrelations could improve the overall coherence.
4.	In lines 230 to 245, where the mathematical principles of the framework are introduced, the abundance of symbols can be overwhelming. It would be clearer if each symbol were introduced upon its first occurrence, or if specific definitions were provided before delving into the mathematical principles and formulas.
5.	In line 369, does the description suggest that contextual and global anomalies are considered as point anomalies in this paper? Similarly, does this apply to the subsequence types in parentheses? It seems that in the field of time series anomaly detection, anomalies are usually classified into various categories without clarifying their hierarchical relationships. This may need further investigation and explanation.
6.	In Figure 5, when examining the temporal scores and frequency domain scores separately, both seem to serve as good anomaly detection metrics. Why does the paper choose to combine these two scores? Are there any examples where the temporal score performs poorly but can be compensated by the frequency domain score (or vice versa)? Such results could better illustrate the effectiveness of this anomaly scoring approach.
7.	There are some typographic faults: The reference in line 185 needs to be enclosed in parentheses; in line 161, there is a spelling error, it should be "modules" instead of "Moudles"; in line 320, it would be better to align the text and the image on the same page.

---

> ### Author Response · Authors · 2024-11-20
> **Response to Reviewer aaEz (Part I)**
>
> Thank you for providing your detailed and constructive feedback.
>
> **W1:** Lack a cohesive explanation of modules interactions.
>
> Thank you for your valuable feedback, which has been instrumental in improving the quality of our paper. Based on your suggestions, we have thoroughly rewritten and reorganized $\underline{\text{Chapter 3 in the revised paper}}$ as follows:
>
> 1. Reorganized Section 3.1, "Architecture Overview," to systematically introduce all modules of CATCH from a macro perspective;
> 2. Restructured Sections 3.2 to 3.4 based on the architecture diagram $\underline{\text{Figure 2 in the revised paper}}$:
>     - Section 3.2 provides an in-depth explanation of the Mask Generator, Channel-Masked Transformer Layer, and Channel Correlation Discovering, along with their interconnections;
>     - Section 3.3 focuses on the Time-Frequency Reconstruction Module;
>     - Section 3.4 elaborates on the implementation logic of joint bi-level optimization, connecting Sections 3.2 and 3.3, and clearly presents the final loss function formulation;
> 3. Section 3.5 explains how anomaly scores are calculated.
>
> **W2:** Key operations like instance normalization and frequency domain patching are described without adequate context, leaving readers questioning their necessity.
>
> Thank you for your valuable feedback! Instance normalization, as a commonly used module in time series modeling, functions to stabilize the statistical characteristics of time series during training. This helps mitigate the adverse effects caused by distribution shifts between the training and testing data, thereby enhancing the model's generalization capability. Additionally, in the Introduction section, we have elaborated on the need to segment frequency bands to capture finer-grained inter-channel relationships, with frequency domain patching being a technique to achieve this segmentation. Following your suggestion, we have supplemented sufficient context $\underline{\text{in Section 3.1 of the revised paper}}$ to explain the motivation and rationale behind instance normalization and frequency domain patching.
>
> **W3:** The innovation behind the Channel Fusion Module (CFM) is not sufficiently highlighted.
>
> Thank you for your feedback. We would like to clarify the core innovation of our proposed CFM. The primary novelty of CFM lies in its ability to discover appropriate inter-channel relationships for each fine-grained frequency band, thereby improving the reconstruction of multivariate time series and enhancing anomaly detection capabilities.
>
> Fixed channel relationships, such as CI (channel-independent) and CD (channel-dependent) face inherent limitations. They fail to balance capacity and robustness, indicating that neither CI nor CD alone represents the optimal approach to modeling channel relationships. Our proposed CFM, however, dynamically identifies the most suitable relationship between CI and CD for each frequency band. Leveraging a specially designed optimization strategy, CFM iteratively refines these relationships.
>
> Regarding the implementation, the masking mechanism you mentioned acts as a CI-based **Mask Generator**, which dynamically perceives intra-band channel correlations to generate a mask matrix. This matrix reflects absolute relationships, determining which channels should be considered for interaction and which should not. For uncorrelated channels, the masking mechanism effectively eliminates potential adverse effects, addressing the robustness issues inherent in CD. Conversely, for correlated channels, relationships are computed through the **Channel-masked Transformer Layer**, which calculates attention scores.
>
> In essence, while the Mask Generator works in a CI manner, it enables the Channel-masked Transformer Layer to avoid the drawbacks of CD. Furthermore, in our **Channel Correlation Discovering** mechanism, we have designed optimization objectives that adaptively refine the discovered channel relationships.
>
> $\underline{\text{Our ablation studies in Table 3 of the original paper}}$ demonstrate that applying this channel-discovery mechanism within CFM consistently outperforms using CI or CD alone in terms of anomaly detection effectiveness.

---

> ### Author Response · Authors · 2024-11-20
> **Response to Reviewer aaEz (Part II)**
>
> **W4:** The goals and significance of the Time-Frequency Reconstruction Module (TFRM) are not clearly articulated.
>
> Thank you for your valuable feedback! The design purpose and significance of the Time-Frequency Reconstruction Module (TFRM) are as follows:
>
> - TFRM is designed to generate time-domain reconstruction loss and frequency-domain reconstruction loss. These two losses, combined with the contrastive loss and regularization loss in the **Channel Fusion Module (CFM)**, together constitute the final loss for backpropagation in this study.
> - In anomaly detection tasks, calculating the anomaly score of a time series is a critical post-processing step. Through this process, we can determine whether each time point is an anomaly and estimate its probability. Similar to other reconstruction-based anomaly detection methods, CATCH uses reconstruction errors as anomaly scores to identify anomalies in time series. TFRM comprises time-domain and frequency-domain reconstruction components, aiming to ensure the model performs well in both domains. With this design, we combine the reconstruction errors from both domains in the subsequent $\underline{\text{Section 3.4 of the original paper}}$ to produce the final anomaly scores. When calculating anomaly scores, the time-domain reconstruction loss is better suited for detecting point anomalies, while the frequency-domain reconstruction loss is more effective for handling subsequence anomalies. By integrating reconstruction errors from both domains, CATCH can effectively address both point and subsequence anomalies simultaneously.
>
> **Q1:** What's the definition for heterogeneous subsequence anomalies?
>
> We sincerely apologize for not providing a detailed explanation of "heterogeneous subsequence anomalies" in the previous version. In fact, the term "heterogeneous" emphasizes diversity, referring to the rich and varied types of subsequence anomalies. As described in [1,2], subsequence anomalies can be further categorized into seasonal, shapelet, and trend anomalies. Therefore, the types of subsequence anomalies exhibit considerable diversity.
>
> [1] Revisiting time series outlier detection: Definitions and benchmarks. NeurIPS 2021
>
> [2] Deep learning for time series anomaly detection: A survey.  ACM Computing Surveys 2024
>
> **Q2:** In sections 2.2 and 2.3, in addition to discussing the current state of research, could you include some definitional descriptions?
>
> Thank you for your valuable feedback! Based on your suggestions, we have added definitional descriptions at the beginning of $\underline{\text{Sections 2.2 and 2.3 in the revised paper}}$ to further clarify the concepts of "channel" and "channel strategy" and to explain the necessity of considering channel correlations and modeling from a frequency domain perspective. The specific content is as follows:
>
> A channel refers to a variable in multivariate time series anomaly detection, while a channel strategy refers to how these inter-variable correlations are effectively considered during the modeling process. This paper employs a reconstruction-based anomaly detection algorithm, using reconstruction error as the anomaly score, making reconstruction quality crucial for anomaly detection accuracy. Since the channels in multivariate time series often exhibit complex dependencies, explicitly modeling these correlations enables a more comprehensive capture of global features, thereby improving reconstruction capabilities and anomaly detection performance.
>
> Frequency domain analysis can uncover subsequence anomalies that are challenging to detect in the time domain, such as anomalies in periodic fluctuations or oscillation patterns, significantly enhancing detection accuracy. Consequently, frequency-based time series anomaly detection models have garnered widespread attention in recent years.
>
> **Q3:** In section 3.1, the description of the framework structure provides technical details, but it lacks a clear explanation of the processes involved.
>
> Thank you for your valuable feedback, which has played a significant role in improving the quality of our paper. Based on your suggestions, we have reorganized $\underline{\text{Section 3.1 of the revised paper}}$, "Architecture Overview," to systematically introduce the various modules of CATCH from a macro perspective.

---

> ### Author Response · Authors · 2024-11-20
> **Response to Reviewer aaEz (Part III)**
>
> **Q4:** In lines 230 to 245, where the mathematical principles of the framework are introduced, the abundance of symbols can be overwhelming.
>
> Thank you for your valuable feedback, which has been instrumental in improving the quality of our paper. Based on this specific suggestion, we provide precise definitions of the variables before delving into the mathematical principles and formulas $\underline{\text{in lines 243-261 of the revised paper}}$:
>
> **Q5:** In line 369, does the description suggest that contextual and global anomalies are considered as point anomalies in this paper? Similarly, does this apply to the subsequence types in parentheses?
>
> Consistent with the perspectives of [1, 2, 3] and other studies, we also categorize time series anomalies into point anomalies and subsequence anomalies. Point anomalies are further divided into global point anomalies and contextual point anomalies, while subsequence anomalies can be classified into trend anomalies, seasonal anomalies, and shapelet anomalies.
>
> We apologize for not providing an explanation in the original version of the paper. Thank you for your suggestion; we have now included a detailed explanation $\underline{\text{in line 38 of the revised paper}}$.
>
>
>
> [1] Revisiting time series outlier detection: Definitions and benchmarks. NeurIPS 2021
>
> [2] Deep learning for time series anomaly detection: A survey.  ACM Computing Surveys 2024
>
> [3] DCdetector: Dual Attention Contrastive Representation Learning for Time Series Anomaly Detection. KDD 2023
>
> **Q6:** In Figure 5, when examining the temporal scores and frequency domain scores separately, both seem to serve as good anomaly detection metrics. Why does the paper choose to combine these two scores?
>
> $\underline{\text{From Figure 5 of the original paper}}$, it can be observed that for point anomalies (first and second columns), temporal anomaly scores increase sharply at the actual anomaly locations, whereas frequency anomaly scores tend to identify a broader range of intervals as anomalies. For pattern anomalies (third, fourth, and fifth columns), frequency anomaly scores remain high throughout the entire anomalous interval, while temporal anomaly scores only detect a small portion of the anomalous interval. Thus, each domain plays a distinct role in anomaly detection.
>
> To better address real-world time series that may contain both point anomalies and subsequence anomalies, we chose to combine the two anomaly scores using a weighted approach. Compared to temporal anomaly scores, which are better at detecting point anomalies, or frequency anomaly scores, which are more effective for detecting subsequence anomalies, our approach allows for more accurate detection of both point and subsequence anomalies, offering a more comprehensive anomaly detection capability.
>
> **Q7:** There are some typographic faults.
>
> Thank you for your valuable feedback. We have thoroughly revised the entire manuscript $\underline{\text{in the revised paper}}$.

---

> ### Author Response · Authors · 2024-11-25
> **Waiting for response**
>
> Dear Reviewer aaEz,
>
> Since the End of the Rebuttal comes soon, we would like to inquire if our response addresses your primary concerns. If it does, we kindly request that you reconsider the score. If you have any additional suggestions, we are more than willing to engage in further discussions and make necessary improvements to the paper. Thanks again for dedicating your time to enhancing our paper!
>
> Kind Regards, Authors

---

> ### Author Response · Authors · 2024-11-29
> **Kindly Request for Reviewer's Feedback**
>
> Dear Reviewer aaEz,
>
> We sincerely appreciate the effort you have put into reviewing our paper during this busy period, as well as the recognition of the strengths of our work. We are truly grateful for the increased score. We fully respect your current evaluation of our work. However, based on the scoring standards of past ICLRs, we find that our current score of 5.75 is at the borderline level. Therefore, we kindly ask if you could consider raising the score once again, so that we would have the opportunity to present our work at the conference. We would be deeply grateful. Thank you again for taking the time to review and comment on our paper.

---

> > ### Author Response · Authors · 2024-12-02
> > **Kindly Request for Reviewer's Feedback**
> >
> > Dear Reviewer aaEz,
> >
> > Sorry to bother you again. We sincerely appreciate the effort you have put into reviewing our paper during this busy period, as well as the recognition of the strengths of our work. We are truly grateful for the increased score. We fully respect your current evaluation of our work.
> >
> > The rebuttal is ending soon. However, based on the scoring standards of past ICLRs, we find that our current score of 5.75 is at the borderline level. Therefore, we kindly ask if you could consider raising the score once again, so that we would have the opportunity to present our work at the conference. We would be deeply grateful. Thank you again for taking the time to review and comment on our paper.

---

### Official Review · Reviewer_wHnp · 2024-11-03

**Soundness:** 2
**Presentation:** 3
**Contribution:** 2
**Rating:** 5
**Confidence:** 4

**Summary:**

This paper presents a new multivariate time series anomaly detection method that operates in the frequency domain. The proposed method introduces two main components: (1) a frequency patching mechanism that segments the frequency domain into bands for capturing fine-grained frequency characteristics, and (2) a channel fusion module that leverages patch-wise masking and masked attention to find relevant channel correlations. The approach is optimized using a bi-level multi-objective algorithm.

**Strengths:**

1.	The paper exhibits good presentation standards. Specifically, the quality of figures can reach the bar of ICLR. The visualizations effectively communicate complex concepts and experimental results.
2.	The idea of frequency patch learning is novel in the field of time series anomaly detection, while I am not sure whether this idea is proposed in the domain of time series modeling.
3.	The authors released an anonymous GitHub repository, which improves reproducibility.

**Weaknesses:**

1. The foundational motivation of the paper needs stronger articulation. The authors primarily focus on the limitations of reconstruction-based methods, specifically their tendency to overlook details in high-frequency bands. However, several studies have already explored leveraging frequency information in MTSAD. The introduction section could benefit from a detailed comparison between current frequency-based methods and the proposed approach. A structured comparison that highlights the specific gaps addressed by CATCH would strengthen the paper’s positioning and clarify its contributions to the field. Consider including a table or a structured comparison in the introduction that clearly outlines how CATCH addresses specific gaps in existing frequency-based MTSAD methods, such as handling high-frequency information, computational efficiency, and the ability to capture channel correlations.
2. The experimental evaluation would benefit from a broader comparison with relevant frequency-based methods. Although several approaches (e.g., SR-CNN, PFT, TFAD, and Dual-TF) are mentioned, they are notably absent from the experimental comparisons. To enhance this aspect, it would be constructive to include these specific methods in the experimental comparisons and to explain why comparing against these particular approaches is valuable for demonstrating CATCH’s contributions.
3. The paper’s central claim regarding superior detection of frequency-specific anomalies requires more rigorous theoretical or empirical validation. While the method aims to better capture fine-grained frequency characteristics, there is no dedicated experimental design to verify this capability. To address this, consider adding specific experiments or analyses that can demonstrate superior detection of frequency-specific anomalies. For instance, you could create synthetic datasets with known frequency-specific anomalies or analyze performance across different frequency bands.
4. The manuscript requires careful editorial revision to address. There are some typos like “results” in the abstract. Also, some citations should be updated to published versions instead of preprints.

**Questions:**

Could you elaborate on how your frequency patching approach differs from existing frequency-domain analysis methods?

---

> ### Author Response · Authors · 2024-11-20
> **Response to Reviewer wHnp (Part I)**
>
> Thank you for your thoughtful and constructive comments.
>
> **W1:** Outlines how CATCH addresses specific gaps in existing frequency-based TSAD methods.
>
> Thank you for your suggestions. $\underline{\text{In the revised  paper}}$, we have included a table in the Related Work section to highlight how CATCH addresses specific gaps in existing frequency-based multivariate time series anomaly detection (MTSAD) methods. Additionally, we have reorganized and rewritten Section 2.3, Frequency Domain Analysis for TSAD. The specific modifications are as follows:
>
> SR-CNN[1], the first method to leverage the frequency domain for TSAD, employs a frequency-based approach to generate saliency maps for identifying anomalies, and PFT[2] built on this foundation by introducing partial Fourier transform to achieve substantial acceleration. However, both methods are confined to univariate time series and fail to address the complexities of multivariate scenarios. TFAD[3] emerges as the first approach to integrate time-domain and frequency-domain analyses for MTSAD, yet it lacks time-frequency granularity alignment. Dual-TF[4], the most recent algorithm for MTSAD using time-frequency analysis, partially addresses the time-frequency granularity alignment issue but lacks a tailored backbone, instead relying directly on the Anomaly Transformer. While TFAD and Dual-TF represent progress in this area, they still exhibit the following limitations: i) Frequency-domain modeling methods have inherent biases, often overlooking high-frequency information; 2) Insufficient exploration and utilization of channel correlations in multivariate time series.
>
>
> | **Property** | **Mmultivariate time serie anomaly detection** | **Time-frequency granularity alignment** | **Handle high-frequency information** | **Capture channel correlations** |
> | :---: | :---: | :---: | :---: | :---: |
> | SR-CNN | ✗ | ✗ | ✗ | ✗ |
> | PFT | ✗ | ✗ | ✗ | ✗ |
> | TFAD | ✓ | ✗ | ✗ | ✗ |
> | Dual-TF | ✓ | ✓ | ✗ | ✗ |
> | CATCH | ✓ | ✓ | ✓ | ✓ |
>
> [1]  Time-series anomaly detection service at microsoft. KDD 2019
>
> [2] Fast and accurate partial fourier transform for time series data. KDD 2022
>
> [3]  TFAD: A decomposition time series anomaly detection architecture with time-frequency analysis. CIKM 2022
>
> [4] Breaking the time-frequency granularity discrepancy in time-series anomaly detection. WWW 2024
>
>
> **W2:** Lack of evaluation of some baselines.
>
> Thank you for your valuable feedback! Please note that $\underline{\text{in our original paper}}$, we compared CATCH with the latest 2024 work, Dual-TF. According to the $\underline{\text{results in Table 1 of the original paper}}$, CATCH significantly outperforms Dual-TF. The reasons for not selecting other frequency domain analysis methods as baselines are as follows:
>
> - SR-CNN and PFT were published earlier and are both designed for univariate time series anomaly detection tasks; hence, they were not included in the comparison.
> - TFAD was published in 2022, while Dual-TF was published in 2024. Moreover, the Dual-TF paper has demonstrated its superiority over TFAD, which is why we only compared CATCH with the state-of-the-art Dual-TF in the original paper.
>
> Following your suggestion, we have added a comparison with TFAD. The experimental results show that CATCH still demonstrates superior performance.

---

> > ### Author Response · Authors · 2024-11-20
> > **Response to Reviewer wHnp (Part II): Comparison with new baseline TFAD**
> >
> > **We have added a comparison with TFAD. The experimental results show that CATCH still demonstrates superior performance.**
> >
> > Average A-R (AUC-ROC) and Aff-F (Affiliated-F1) accuracy measures for 12 real-world datasets and 6 synthetic datasets of different types of anomalies. The best results are highlighted in bold, and the second-best results are underlined.
> >
> > | Dataset        | Metric | CATCH        | Modern       | iTrans       | DualTF       | TFAD         |
> > | -------------- | ------ | ------------ | ------------ | ------------ | ------------ | ------------ |
> > | **CICIDS**     | Aff-F  | **0.787**    | 0.654        | $\underline{\text{0.708}}$ | 0.692        | 0.579        |
> > |                | A-R    | **0.795**    | $\underline{\text{0.697}}$ | 0.692        | 0.603        | 0.504        |
> > | **CalIt2**     | Aff-F  | **0.835**    | 0.780        | $\underline{\text{0.812}}$ | 0.751        | 0.744        |
> > |                | A-R    | **0.838**    | 0.676        | $\underline{\text{0.791}}$ | 0.574        | 0.504        |
> > | **Credit**     | Aff-F  | **0.750**    | $\underline{\text{0.744}}$ | 0.713        | 0.663        | 0.660        |
> > |                | A-R    | **0.958**    | $\underline{\text{0.957}}$ | 0.934        | 0.703        | 0.500        |
> > | **GECCO**      | Aff-F  | **0.908**    | $\underline{\text{0.893}}$ | 0.839        | 0.701        | 0.627        |
> > |                | A-R    | **0.970**    | $\underline{\text{0.952}}$ | 0.795        | 0.714        | 0.499        |
> > | **Genesis**    | Aff-F  | **0.896**    | 0.833        | $\underline{\text{0.891}}$ | 0.810        | 0.535        |
> > |                | A-R    | **0.974**    | 0.676        | 0.690        | $\underline{\text{0.937}}$ | 0.497        |
> > | **MSL**        | Aff-F  | **0.740**    | $\underline{\text{0.726}}$ | 0.710        | 0.588        | 0.665        |
> > |                | A-R    | **0.664**    | $\underline{\text{0.633}}$ | 0.611        | 0.576        | 0.500        |
> > | **NYC**        | Aff-F  | **0.994**    | $\underline{\text{0.769}}$ | 0.684        | 0.708        | 0.689        |
> > |                | A-R    | **0.816**    | 0.466        | $\underline{\text{0.640}}$ | 0.633        | 0.502        |
> > | **PSM**        | Aff-F  | **0.859**    | 0.825        | $\underline{\text{0.854}}$ | 0.725        | 0.628        |
> > |                | A-R    | **0.652**    | 0.593        | 0.592        | $\underline{\text{0.600}}$ | 0.500        |
> > | **SMD**        | Aff-F  | **0.847**    | $\underline{\text{0.840}}$ | 0.827        | 0.679        | 0.660        |
> > |                | A-R    | **0.811**    | 0.722        | $\underline{\text{0.745}}$ | 0.631        | 0.500        |
> > | **SMAP**       | Aff-F  | **0.699**    | 0.635        | 0.587        | 0.674        | $\underline{\text{0.675}}$ |
> > |                | A-R    | **0.504**    | 0.455        | 0.409        | 0.478        | $\underline{\text{0.500}}$ |
> > | **SWAT**       | Aff-F  | **0.755**    | $\underline{\text{0.728}}$ | 0.718        | 0.695        | 0.686        |
> > |                | A-R    | $\underline{\text{0.545}}$ | 0.244        | 0.242        | **0.567**    | 0.500        |
> > | **ASD**        | Aff-F  | **0.804**    | $\underline{\text{0.782}}$ | 0.780        | 0.605        | 0.630        |
> > |                | A-R    | **0.824**    | 0.692        | $\underline{\text{0.759}}$ | 0.579        | 0.502        |
> > | **Contextual** | Aff-F  | **0.823**    | 0.619        | $\underline{\text{0.802}}$ | 0.635        | 0.569        |
> > |                | A-R    | **0.910**    | 0.562        | $\underline{\text{0.905}}$ | 0.598        | 0.504        |
> > | **Global**     | Aff-F  | **0.949**    | 0.748        | $\underline{\text{0.922}}$ | 0.649        | 0.566        |
> > |                | A-R    | **0.997**    | 0.873        | $\underline{\text{0.976}}$ | 0.595        | 0.500        |
> > | **Seasonal**   | Aff-F  | **0.997**    | 0.681        | $\underline{\text{0.992}}$ | 0.776        | 0.686        |
> > |                | A-R    | **0.998**    | 0.512        | $\underline{\text{0.946}}$ | 0.701        | 0.502        |
> > | **Shapelet**   | Aff-F  | **0.985**    | 0.675        | $\underline{\text{0.961}}$ | 0.692        | 0.684        |
> > |                | A-R    | **0.970**    | 0.522        | $\underline{\text{0.864}}$ | 0.573        | 0.503        |
> > | **Trend**      | Aff-F  | **0.916**    | 0.734        | $\underline{\text{0.901}}$ | 0.677        | 0.642        |
> > |                | A-R    | **0.892**    | 0.612        | $\underline{\text{0.847}}$ | 0.524        | 0.502        |
> > | **Mixture**    | Aff-F  | **0.892**    | 0.856        | $\underline{\text{0.862}}$ | 0.652        | 0.710        |
> > |                | A-R    | **0.931**    | 0.763        | $\underline{\text{0.854}}$ | 0.570        | 0.501        |

---

> ### Author Response · Authors · 2024-11-20
> **Response to Reviewer wHnp (Part III)**
>
> **W3:** Lack of dedicated experimental designs to verify the ability to effectively capture fine-grained frequency characteristics.
>
> Thank you for your valuable feedback! Based on your suggestions, we designed a set of experiments to validate the advantages of our method in capturing fine-grained frequency characteristics. We used the synthetic dataset creation method provided by TODS[1] to generate four synthetic datasets, simulating scenarios with known frequency-specific anomalies: low-frequency anomalies, medium-frequency anomalies, high-frequency anomalies, and anomalies that are not distinctly separated across different frequency bands. We then tested CATCH and several state-of-the-art methods on these datasets.
>
> The experimental results $\underline{\text{in Table 12 in lines 1350-1359 of  the revised paper}}$ show that CATCH consistently outperforms other algorithms across all four datasets. Notably, most algorithms perform poorly when dealing with time series containing high-frequency anomalies, while CATCH demonstrates outstanding performance. Moreover, even on time series where anomalies are uniformly distributed across each frequency band, CATCH still achieves excellent results. These findings provide strong evidence that CATCH excels in capturing fine-grained frequency characteristics, significantly enhancing the performance of time series anomaly detection.
>
> [1] Revisiting time series outlier detection: Definitions and benchmarks. NeurIPS 2021
>
> **W4:** There are some typos like “results” in the abstract. Also, some citations should be updated to published versions instead of preprints.
>
> Thank you for your valuable feedback. We have carefully addressed these issues $\underline{\text{in the revised paper}}$. Specifically:
>
> - We meticulously check and correct any spelling errors throughout the manuscript.
> - We update the citations to reflect the final published versions instead of preprints, ensuring all references are accurate and up-to-date.
>
> **Q1:** Could you elaborate on how your frequency patching approach differs from existing frequency-domain analysis methods?
>
> Please see weakness 1.  Specifically, compared to SR-CNN, PFT, TFAD, and Dual-TF, our algorithm supports both univariate and multivariate tasks. It achieves time-frequency granularity alignment, enabling fine-grained modeling within each frequency band to accurately reconstruct normal patterns and detect heterogeneous subsequence anomalies. Additionally, it can flexibly adapt to distinct channel interrelationships across different frequency bands.

---

> ### Author Response · Authors · 2024-11-25
> **Waiting for response**
>
> Dear Reviewer wHnp,
>
> Since the End of the Rebuttal comes soon, we would like to inquire if our response addresses your primary concerns. If it does, we kindly request that you reconsider the score. If you have any additional suggestions, we are more than willing to engage in further discussions and make necessary improvements to the paper. Thanks again for dedicating your time to enhancing our paper!
>
> Kind Regards, Authors

---

> > ### Author Response · Authors · 2024-11-26
> > **Waiting for response**
> >
> > Dear Reviewer wHnp,
> >
> > We would like to inquire if our response addresses your primary concerns. If it does, we kindly request that you reconsider the score. If you have any additional suggestions, we are more than willing to engage in further discussions and make necessary improvements to the paper. Thanks again for dedicating your time to enhancing our paper!
> >
> > Kind Regards, Authors

---

> > > ### Author Response · Authors · 2024-11-26
> > > **Kindly Request for Reviewer's Feedback**
> > >
> > > Dear Reviewer wHnp,
> > >
> > > Since the End of author/reviewer discussions is coming soon, may we know if our response addresses your main concerns? If so, we kindly ask for your reconsideration of the score. If you have any further concerns, please let us know and we will be more than happy to engage in more discussion and paper improvements.
> > >
> > > Once again, thank you for your suggestion and time!

---

> > > > ### Author Response · Authors · 2024-11-29
> > > > **Kindly Request for Reviewer's Feedback**
> > > >
> > > > Dear Reviewer wHnp
> > > >
> > > > We really appreciate your efforts during this tight review timeline and the recognition of the strengths of our paper.
> > > >
> > > > We are really happy and respect your current rating for our work.  We would be grateful if you could confirm that you have read our responses and let us know if there's anything we might have overlooked that could improve the score so that we can have the opportunity to showcase this work at the conference. We are eager to discuss any concerns that may remain open after our response. Thank you again for taking the time for this review.

---

> > > > > ### Author Response · Authors · 2024-12-02
> > > > > **Kindly Request for Reviewer's Feedback**
> > > > >
> > > > > Dear Reviewer wHnp
> > > > >
> > > > > We really appreciate your efforts during this tight review timeline and the recognition of the strengths of our paper.
> > > > >
> > > > > We are really happy and respect your current rating for our work. We would be grateful if you could confirm that you have read our responses and let us know if there's anything we might have overlooked that could improve the score so that we can have the opportunity to showcase this work at the conference. Since the rebuttal is ending soon, we are eager to discuss any concerns that may remain open after our response. Thank you again for taking the time for this review.

---

> > > > > > ### Author Response · Authors · 2024-12-03
> > > > > > **Kindly Request for Reviewer's Feedback**
> > > > > >
> > > > > > Dear Reviewer wHnp,
> > > > > >
> > > > > > We greatly appreciate the effort you have put into reviewing our paper during the review phase, as well as your recognition of the strengths of our work. We have responded to each of your constructive comments point by point. We would be very grateful if you could confirm that you have read our responses and let us know if there is anything we might have overlooked that could further improve the quality of the paper.
> > > > > >
> > > > > > We have already addressed the concerns of the other reviewers, and have received higher ratings for those responses. If our replies have satisfied you, we would also be very grateful if you could reconsider the score, which would give us a greater opportunity to present this work at the conference. As the rebuttal phase is ending soon, we would like to discuss any remaining concerns that might still exist after our responses. Once again, thank you for taking the time to review this work.
> > > > > >
> > > > > > Best regards!

---

> > > > > > > ### Author Response · Authors · 2024-12-04
> > > > > > > **Kindly Request for Reviewer's Feedback**
> > > > > > >
> > > > > > > Dear Reviewer wHnp,
> > > > > > >
> > > > > > > We greatly appreciate the effort you have put into reviewing our paper during the review phase and we have responded to each of your constructive comments point by point. Since the author/reviewer discussion phase concluded, we hope you could reconsider the score to give us a greater opportunity to present this work at the conference.
> > > > > > >
> > > > > > > Best regards!

---

### Official Review · Reviewer_AEcB · 2024-11-03

**Soundness:** 3
**Presentation:** 3
**Contribution:** 3
**Rating:** 6
**Confidence:** 4

**Summary:**

The paper proposed a method called CATCH
 for Multivariate Time Series Anomaly Detection.

**Strengths:**

1. The paper is well structured.

2. The architecture of CATCH is clearly articulated.

3. Nice figures help to understand the proposed CATCH

4. The proposed CATCH has been verified by extensive experiments.

5. The code of the proposed method has been open, thus it has good reproducibility.

**Weaknesses:**

1. Figures are not self-explanatory sufficiently such as Fig. 1(b).
2. Lack of parameter settings, resulting in weak reproducibility.
3. Table 1 shows that the proposed method outperforms all baselines for all tested datasets. Does this mean that the proposed method can win in any dataset/application?
4. The tested datasets are not the ones most commonly tested by the baselines. Why? Please test some datasets tested by your baselines.
5. No guidance on choosing parameters for your proposed method?

**Questions:**

Please see above.

---

> ### Author Response · Authors · 2024-11-20
> **Response to Reviewer AEcB (Part I)**
>
> We would like to sincerely thank Reviewer AEcB for providing a detailed review and insightful comments. We have revised our paper accordingly.
>
> **W1:**  Fig. 1(b) is not self-explanatory sufficiently.
>
> Thank you for your valuable feedback! In the description of Figure 1(b) in the original paper, we briefly introduced the content of the image, while its meaning was elaborated in detail in lines 54-60 of the main text. To more clearly convey the meaning of the figure, we have redrawn Figure 1(b) and revised its caption $\underline{\text{in lines 80-86 of the revised paper}}$. The new description is as follows:
>
> "(b) Shows the frequency bands of a multivariate time series with three channels. These channels exhibit varying correlations across different frequency bands, where Channels 1 and 2 exhibit similar behavior in the third frequency band and are therefore marked in red, while Channel 3 exhibits distinct characteristics and is marked in yellow. In the fourth frequency band, all channels behave similarly and are marked in green, while in the fifth band, they exhibit distinct characteristics and are marked in different colors."
>
> **W2:** Lack of parameter settings, resulting in weak reproducibility.
>
> Thank you for your valuable feedback on our work! Your suggestions have played a significant role in improving the quality of our paper.
>
> Our paper is highly reproducible. $\underline{\text{In line 27 of the abstract in the original paper}}$, we provided a link to an anonymous code repository. Within the repository's `scripts` folder, we provided parameter configuration scripts for CATCH and all baseline methods that achieve optimal performance on all datasets. Following the tutorial in the README file, readers can reproduce all the experimental results.
>
> However, we acknowledge that the descriptions of baseline method parameter settings in the original paper were not sufficiently detailed. To address this, we made improvements in the revised version, adding detailed explanations of parameter settings for all baseline methods $\underline{\text{in lines 814-837 of the revised paper}}$.
>
> The specific changes are as follows:
>
> For each baseline method, we strictly followed the hyperparameter configurations recommended in their original papers. Additionally, we conducted hyperparameter searches on multiple sets $\underline{\text{see Appendix A.4 of the revised paper}}$ and selected the optimal configurations based on these evaluations to ensure a comprehensive and fair assessment of each method's performance.
>
> **W3:** Table 1 shows that the proposed method outperforms all baselines for all tested datasets. Does this mean that the proposed method can win in any dataset/application?
>
> Thank you for your question! Our algorithm does not always outperform others across all datasets or applications. Due to space limitations, $\underline{\text{Table 1 in the original paper}}$ only reports results under two widely accepted evaluation metrics: a label-based metric *Affiliated-F1-score* (*Aff-F*) and a score-based metric *Area under the Receiver Operating Characteristics Curve* (*ROC*). These two metrics have been widely used in several studies to evaluate algorithm performance [1,2,3], which is why we focus on them in the main text.
>
> Under these two metrics, our algorithm performs exceptionally well. However, real-world applications are often more complex, and as the ``No-Free-Lunch Theorem"  suggests, it is impossible to consistently outperform all other algorithms across all datasets. Furthermore, as shown $\underline{\text{in Table 2 and Tables 5–9 in the appendix of the original paper}}$, our algorithm leads in most cases under other evaluation metrics but does not achieve consistent superiority across all of them.
>
> [1] TSB-UAD: An End-to-End Benchmark Suite for Univariate Time-Series Anomaly Detection. PVLDB 2022
>
> [2] Anomaly Detection in Time Series: A Comprehensive Evaluation. PVLDB 2022
>
> [3] Drift doesn’t Matter: Dynamic Decomposition with Diffusion Reconstruction for Unstable Multivariate Time Series Anomaly Detection. NeurIPS 2023

---

> ### Author Response · Authors · 2024-11-20
> **Response to Reviewer AEcB (Part II)**
>
> **W4:** Lack of evaluation of some datasets.
>
> Thank you for your valuable feedback. We apologize for not including some commonly used benchmark datasets, such as SWaT and SMAP, in our experiments. The reason is that [1, 2] pointed out significant issues with these datasets, making them unsuitable for anomaly detection.
>
> Additionally, since there is currently no widely recognized high-quality multivariate anomaly detection dataset, we have supplemented our evaluation by including SWaT and SMAP. The results show that our method CATCH still performs exceptionally well on these datasets, achieving superior performance.
>
> [1] TimeSeAD: Benchmarking Deep Multivariate Time-Series  Anomaly Detection. TMLR 2023
>
> [2] Position: Quo Vadis, Unsupervised Time Series Anomaly Detection?  ICML 2024
>
> | Dataset  | Method       | Acc          | P            | R            | F1           | R-P          | R-R          | R-F          | Aff-P        | Aff-R        | Aff-F        | A-R          | A-P          | R-A-R        | R-A-P        | V-ROC        | V-PR         |
> | -------- | ------------ | ------------ | ------------ | ------------ | ------------ | ------------ | ------------ | ------------ | ------------ | ------------ | ------------ | ------------ | --------------- | ---------------- | --------------- | ------------ | ------------ |
> | **SAMP** | DualTF       | 0.663        | 0.119        | **0.256**    | **0.163**    | 0.110        | $\underline{\text{0.226}}$ | $\underline{\text{0.148}}$ | 0.509        | **0.997**    | 0.674        | 0.478        | 0.122        | $\underline{\text{0.516}}$ | $\underline{\text{0.141}}$ | $\underline{\text{0.515}}$ | $\underline{\text{0.141}}$ |
> |          | iTransformer | 0.685        | 0.086        | 0.151        | 0.110        | 0.088        | 0.211        | 0.124        | 0.439        | 0.885        | 0.587        | 0.409        | 0.114        | 0.441        | 0.126        | 0.439        | 0.126        |
> |          | Modern       | 0.689        | 0.096        | $\underline{\text{0.171}}$ | $\underline{\text{0.123}}$ | 0.087        | **0.276**    | 0.133        | 0.472        | 0.971        | 0.635        | 0.455        | 0.114        | 0.491        | 0.129        | 0.489        | 0.129        |
> |          | TFAD         | **0.868**    | $\underline{\text{0.125}}$ | 0.005        | 0.010        | $\underline{\text{0.125}}$ | 0.118        | 0.122        | $\underline{\text{0.515}}$ | $\underline{\text{0.979}}$ | $\underline{\text{0.675}}$ | $\underline{\text{0.500}}$ | $\underline{\text{0.128}}$ | 0.504        | 0.134        | 0.504        | 0.133        |
> |          | CATCH (ours) | $\underline{\text{0.862}}$ | **0.236**    | 0.035        | 0.061        | **0.164**    | 0.182        | **0.173**    | **0.574**    | 0.895        | **0.699**    | **0.504**    | **0.131**    | **0.546**    | **0.155**    | **0.543**    | **0.155**    |
> | **SWAT** | DualTF       | 0.635        | $\underline{\text{0.143}}$ | **0.404**    | **0.212**    | 0.122        | **0.220**    | **0.157**    | 0.535        | **0.993**    | 0.695        | **0.567**    | $\underline{\text{0.143}}$ | $\underline{\text{0.611}}$ | $\underline{\text{0.172}}$ | $\underline{\text{0.608}}$ | $\underline{\text{0.171}}$ |
> |          | iTransformer | 0.794        | 0.088        | $\underline{\text{0.074}}$ | 0.080        | 0.066        | $\underline{\text{0.203}}$ | 0.099        | 0.568        | $\underline{\text{0.976}}$ | 0.718        | 0.242        | 0.084        | 0.353        | 0.122        | 0.344        | 0.118        |
> |          | Modern       | 0.836        | 0.119        | 0.055        | 0.075        | 0.072        | 0.200        | 0.106        | 0.593        | 0.942        | $\underline{\text{0.728}}$ | 0.244        | 0.093        | 0.357        | 0.138        | 0.348        | 0.132        |
> |          | TFAD         | **0.878**    | 0.135        | 0.001        | 0.001        | **0.135**    | 0.115        | 0.124        | $\underline{\text{0.613}}$ | 0.777        | 0.686        | 0.500        | 0.123        | 0.509        | 0.165        | 0.508        | 0.164        |
> |          | CATCH (ours) | $\underline{\text{0.857}}$ | **0.195**    | 0.056        | $\underline{\text{0.087}}$ | $\underline{\text{0.124}}$ | 0.184        | $\underline{\text{0.148}}$ | **0.623**    | 0.956        | **0.755**    | $\underline{\text{0.545}}$ | **0.166**    | **0.673**    | **0.251**    | **0.662**    | **0.241**    |
>
> **W5:** Lack guidance on choosing parameters for the proposed method.
>
> Since our anomaly detection method is based on reconstruction and operates in an unsupervised manner, we avoid using labeled data when selecting algorithm hyperparameters. To address this, we adopted the unsupervised technique from [1] for parameter selection.
>
> [1] Unsupervised Time Series Outlier Detection with Diversity-Driven Convolutional Ensembles. PVLDB 2022

---

> > ### Comment · Reviewer_AEcB · 2024-11-21
> > **Thanks for the response**
> >
> > I have improved scores. Good luck.

---

> > > ### Author Response · Authors · 2024-11-22
> > >
> > > Thank you very much for appreciating our work. Best wishes!

---

> > > ### Author Response · Authors · 2024-11-29
> > > **Kindly Request for Reviewer's Feedback**
> > >
> > > Dear Reviewer AEcB,
> > >
> > > We sincerely appreciate the effort you have put into reviewing our paper during this busy period, as well as the recognition of the strengths of our work. We are truly grateful for the increased score. We fully respect your current evaluation of our work. However, based on the scoring standards of past ICLRs, we find that our current score of 5.75 is at the borderline level. Therefore, we kindly ask if you could consider raising the score once again, so that we would have the opportunity to present our work at the conference. We would be deeply grateful. Thank you again for taking the time to review and comment on our paper.

---

> > > > ### Author Response · Authors · 2024-12-02
> > > > **Kindly Request for Reviewer's Feedback**
> > > >
> > > > Dear Reviewer AEcB,
> > > >
> > > > Sorry to bother you again. We sincerely appreciate the effort you have put into reviewing our paper during this busy period, as well as the recognition of the strengths of our work. We are truly grateful for the increased score. We fully respect your current evaluation of our work.
> > > >
> > > > The rebuttal is ending soon. However, based on the scoring standards of past ICLRs, we find that our current score of 5.75 is at the borderline level. Therefore, we kindly ask if you could consider raising the score once again, so that we would have the opportunity to present our work at the conference. We would be deeply grateful. Thank you again for taking the time to review and comment on our paper.

---

### Author Response · Authors · 2024-11-20
**Summary of changes**

# Summary of changes:
We sincerely thank all the reviewers for their hard work! We have received many constructive comments, on basis of which we have made corresponding revisions to the paper. In our revised paper, all changes are highlighted in blue, and the main updates are summarized as follows.
- In the Introduction section, we clarify the categorization of point anomalies and subsequence anomalies.
- We redraw Figure 1(b) and revise its caption to make it self-explanatory and more informative.
- In Sections 2.2 and 2.3 of the Related Work section, we add definitional descriptions and include a comparison table to clearly demonstrate the advantages of CATCH over other frequency-domain analysis methods.
- We redraw Figure 2, which illustrates the framework of CATCH, to further clarify the interactions between modules and the details of their inputs.
- We reorganize the content of Chapter 3, dividing it into five subsections instead of the original four. This reorganization enhances the correspondence between the text and Figure 2 and improves the explanation of notations to ensure smoother readability.
- In Appendix D, we include empirical results to demonstrate that CATCH can perform fine-grained modeling for each frequency band.
- We correct some grammatical and spelling errors.
- We update the citations to reflect the final published versions instead of preprints, ensuring all references are accurate and up-to-date.

---

### Meta-Review · Area_Chair_PfyH · 2024-12-21

**Metareview:**

The paper tackles a multivariate time series anomaly detection (MTSAD) problem and introduces a channel-dependent approach to detect diverse point and subsequence anomalies in MTSAD. The method is evaluated on 11 MTSAD datasets.

The strengths of the work can be summarized as follows:
- Well written paper [AEcB, wHnp, aaEz]
- The proposed method is well articulated [AEcB]
- The method is novel in MTSAD [wHnp, aaEz, LbNG]
- The effectiveness of the method is verified on a set of MTSAD datasets [AEcB, aaEz, LbNG]
- Code has been released [AEcB, wHnp]

Its weaknesses include:
- Lack of clarity/details in some key modules, implementation and experiment setup [AEcB, LbNG]
- Results on most widely used datasets are missing [AEcB, LbNG]
- Lack of analysis of parameter settings for the proposed method [AEcB]
- The work is not positioned/motivated well w.r.t. existing frequency-based methods [wHnp]
- Lack of empirical comparison to frequency-based methods [wHnp]
- The claim on detecting frequency-specific anomalies requires more dedicated experimental justification [wHnp]
- Lack of insights about how the modules synergize in addressing the challenges mentioned [aaEz]
- Role and significance of key modules of the method are not articulated well [aaEz]

**Additional Comments On Reviewer Discussion:**

Three out of four reviewers engaged in the discussion. The authors did a great job on their rebuttal and addressed the main concerns from all three reviewers. This helps improve their ratings to two weak accepts and one accept.

Reviewer wHnp provided constructive comments but did not participate in the discussion. I checked the replies from the authors on behalf of Reviewer wHnp and believe that the revised paper addresses most of the concerns, including the below major ones:
- The work is not positioned/motivated well w.r.t. existing frequency-based methods [wHnp]
- Lack of empirical comparison to frequency-based methods [wHnp]
- The claim on detecting frequency-specific anomalies requires more dedicated experimental justification [wHnp]

Therefore, although there are still some cons in the paper, the technical novelty of the proposed method and its justification are considered above the ICLR bar.

---

### Decision · Program_Chairs · 2025-01-22

Accept (Poster)